EMBO
Molecular Medicine

# De novo pyrimidine synthesis is a collateral metabolic vulnerability in *NF2*-deficient mesothelioma

Duo Xu [1,7 ✉], Yanyun Gao[2,3,7], Shengchen Liu[4,7], Shiyuan Yin[1], Tong Hu[1], Haibin Deng [2,3], Tuo Zhang[2,3], Balazs Hegedüs[5], Thomas M Marti[2,3], Patrick Dorn[2,3], Shun-Qing Liang [6], Ralph A Schmid [2,3], Ren-Wang Peng [2,3 ✉] & Yongqian Shu [1 ✉]

## Abstract

**Pleural mesothelioma (PM) is one of the deadliest cancers, with limited therapeutic options due to its therapeutically intractable genome, which is characterized by the functional inactivation of tumor suppressor genes (TSGs) and high tumor heterogeneity, including diverse metabolic adaptations. However, the molecular mechanisms underlying these metabolic alterations remain poorly understood, particularly how TSG inactivation rewires tumor metabolism to drive tumorigenesis and create metabolic dependencies. Through integrated multi-omics analysis, we identify for the first time that *NF2* loss of function defines a distinct PM subtype characterized by enhanced de novo pyrimidine synthesis, which *NF2*-deficient PM cells are critically dependent on for sustained proliferation in vitro and in vivo. Mechanistically, *NF2* loss activates YAP, a downstream proto-oncogenic transcriptional coactivator in the Hippo signalling pathway, which in turn upregulates *CAD* and *DHODH*, key enzymes in the de novo pyrimidine biosynthesis pathway. Our findings provide novel insights into metabolic reprogramming in PM, revealing de novo pyrimidine synthesis as a synthetic lethal vulnerability in *NF2*-deficient tumors. This work highlights a potential therapeutic strategy for targeting *NF2*-deficient mesothelioma through metabolic intervention.**

**Keywords** Pleural Mesothelioma (PM); Metabolic Diversity; Neurofibromin 2 (NF2); De Novo Pyrimidine Synthesis; Synthetic Lethality
**Subject Categories** Cancer; Metabolism; Respiratory System

See also: J Jia and GT Stathopoulos

## Introduction

Pleural mesothelioma (PM) is a deadly tumor originating from pleural mesothelial cells and is etiologically associated with asbestos exposure (Carbone et al, 2019). The prognosis for PM is extremely poor, with a median overall survival of less than 1 year and a 5-year survival rate of only 10%, largely due to limited medical therapies (Janes et al, 2021). Despite extensive research aimed at finding therapeutic targets for PM over the years, advancements in its clinical management have been difficult. Although precision oncology has advanced considerably in the treatment of numerous cancers, its application to mesothelioma remains in the nascent phase (Dulloo et al, 2021), indicating that genomic and molecular patient stratification should be explored to enhance current treatment approaches.

Genomic analyses have shown that the development of PM is primarily driven by the functional inactivation of tumor suppressor genes (TSGs) such as cyclin-dependent kinase 2A/2B (*CDKN2A/2B*), BRCA1-associated protein-1 (*BAP1*), neurofibromin 2 (*NF2*) and large tumor suppressor kinase 2 (*LATS2*) (Bueno et al, 2016; Hmeljak et al, 2018; Mangiante et al, 2023). Unlike directly inhibiting oncogenes, creating targeted therapies that restore the function of TSGs is more complex (Bedard et al, 2020). Research has indicated that indirectly targeting the downstream activation signals resulting from TSG inactivation can selectively kill cancer cells, a concept known as "synthetic lethality." For instance, the sensitivity of BRCA-mutated cancer cells to PARP inhibitors exemplifies this therapeutic strategy (Huang et al, 2020). Therefore, gaining insights into the regulation of non-oncogene dependencies may lead to new treatment options that enhance clinical outcomes for PM patients (Nagel et al, 2016).

Metabolic reprogramming is a fundamental characteristic of tumor initiation and progression, allowing cancer cells to adapt to the challenging tumor microenvironment (Hanahan, 2022). While the approach of targeting specific metabolic vulnerabilities in

[1]Department of Oncology, The First Affiliated Hospital of Nanjing Medical University, Nanjing, China. [2]Department of General Thoracic Surgery, Inselspital, Bern University Hospital, University of Bern, Bern, Switzerland. [3]Department for BioMedical Research (DBMR), Inselspital, Bern University Hospital, University of Bern, Bern, Switzerland. [4]Department of Cardiothoracic Surgery, Nanjing First Hospital, Nanjing Medical University, Nanjing, China. [5]Department of Thoracic Surgery, University Medicine Essen - Ruhrlandklinik, West German Cancer Center, University Hospital Essen, University Duisburg-Essen, Tüschener Weg 40, 45239 Essen, Germany. [6]Department of Medicine, University of Minnesota Twin Cities, 516 Delaware Street SE, Minneapolis, MN, USA. [7]These authors contributed equally: Duo Xu, Yanyun Gao, Shengchen Liu. ✉E-mail: xuduo@jsph.org.cn; Renwang.Peng@insel.ch; shuyongqian@csco.org.cn

tumors offers a promising direction in precision oncology, the inherent heterogeneity of tumors presents considerable obstacles to the effective implementation of this strategy (Stine et al, 2022). Considerable efforts have been made in recent years to analyze PM heterogeneity at various levels, including histological, genetic, and molecular (Blum et al, 2019; Mangiante et al, 2023; Meiller et al, 2021; Quetel et al, 2020). However, there has been a lack of studies investigating the molecular basis of this heterogeneity from a metabolic diversity perspective. In addition, a deeper understanding of the interplay between cancer genetics and tumor metabolism is urgently required to refine precision oncology strategies.

In this study, through an extensive multi-omics analysis, we reveal for the first time that upregulated pyrimidine metabolism defines a distinct metabolic subtype of PM, which is closely associated with the genetic inactivation of the tumor suppressor NF2. We further demonstrated that NF2-YAP signaling promotes PM malignancy by transcriptionally upregulating critical enzymes involved in de novo pyrimidine biosynthesis, including carbamoyl-phosphate synthetase 2, aspartate transcarbamoylase, dihydroorotase (CAD), and dihydroorotate dehydrogenase (DHODH). Importantly, at the therapeutic level, our findings establish that targeting de novo pyrimidine synthesis elicits synthetic lethality in NF2-deficient PM, effectively suppressing tumor growth across multiple preclinical models. Overall, our study highlights how genetic alterations shape the metabolic landscape of PM, uncovering novel molecular adaptations that can serve as promising therapeutic targets for personalized treatment strategies against this daunting disease.

## Results

### Metabolic heterogeneity and subtypes in PM

To elucidate the metabolic changes in PM, we performed differential gene expression analysis on a large-scale transcriptomic dataset comprising 65 normal tissues and 328 tumor samples. Our results revealed a distinct pattern of metabolic gene alterations in PM, with 16 genes exhibiting increased expression and 30 genes showing decreased expression relative to normal tissues. Beyond the canonical molecular changes associated with PM, the tumors displayed enrichment in various metabolic processes (Appendix Fig. S1; Datasets EV1 and 2). Gene set enrichment analysis (GSEA) further identified 6 metabolic pathways that were enriched in tumors, alongside 13 pathways enriched in normal tissues. Notably, pyrimidine metabolism emerged as the top-ranked enriched metabolic pathway in PM, while fatty acid degradation was more prevalent in normal tissues (Appendix Fig. S2A; Dataset EV3). High enrichment scores of pyrimidine metabolism were linked to poor clinical outcomes in PM patients (Appendix Fig. S2B,C). Furthermore, pan-cancer analysis showed an increased enrichment scores of pyrimidine metabolism in tumors compared to normal tissues (Appendix Fig. S3), underscoring the recognized role of pyrimidine biosynthesis dysfunction in tumorigenesis (Wang et al, 2021). While the targeting of cancer-specific metabolic vulnerabilities presents considerable promise in the field of precision oncology, the inherent metabolic diversity observed among tumors represents a substantial obstacle (Kim and DeBerardinis, 2019;

Stine et al, 2022). Our analysis of global metabolic gene expression shifts revealed substantial variability in metabolic dysregulation among PM tumors, as evidenced by a greater distance in metabolic gene expression within tumors compared to normal tissues (Appendix Fig. S4A).

To further explore the metabolic diversity in PM, we employed consensus clustering based on GSVA enrichment scores from 84 metabolic pathways. This approach identified three distinct metabolic subtypes, exhibiting a consistent distribution pattern across datasets, thereby affirming the reliability of this method (Fig. 1A; Appendix Fig. S4B; Dataset EV4). Notably, nearly half of the PM tumors were stratified into Cluster 1, characterized by an apparent enrichment in one carbon pool by folate and pyrimidine metabolism (Fig. 1A; Appendix Fig. S4C,D; Dataset EV5). We further stratified mesothelioma cell lines from the Cancer Cell Line Encyclopaedia (CCLE) (Jiang et al, 2023) into three metabolic subgroups using the nearest shrunken centroids method (Gong et al, 2021; Tibshirani et al, 2002). In principle, cell-line clustering is expected to mirror tumor-derived subtypes; however, it is well recognized that long-term in vitro culture can introduce metabolic shifts, causing some lines to diverge from their original profiles(Chernova et al, 2016). To ensure biological relevance and consistency, we therefore restricted downstream analyses to those CCLE lines whose pathway-based classification aligned precisely with the corresponding tumor clusters. Our analysis revealed that metabolites differentially expressed in PM cell lines associated with Cluster 1—corroborated by publicly available metabolomics datasets—were highly enriched in arginine and proline metabolism, as well as pyrimidine metabolism (Appendix Fig. S4E,F; Datasets EV6 and 7). Clinically, a higher proportion of non-epithelioid cases, known for their aggressive nature, were found in Cluster 1 (Fig. 1B). Patients in this cluster exhibited worse prognoses compared to the other two subgroups (Fig. 1C).

In summary, our multi-omics analysis indicates that metabolic heterogeneity is a defining characteristic of PM and may offer insights into tumor biology and therapeutic vulnerabilities. Notably, the dysregulation of pyrimidine metabolism emerges as a clinical factor with potential implications for precision therapy.

### NF2 deficiency defines a distinct PM with elevated pyrimidine synthesis

To explore the interplay between cancer genetics and tumor metabolism, we analyzed genetic alteration frequencies among the three metabolic subtypes using mutation and copy number variant (CNV) data from TCGA-MESO cohort. Our findings indicated that NF2 mutations/deletions were prevalent in Cluster 1, with the highest frequency of genetic alterations; SLC12A1, was frequently altered in Cluster 2, and BAP1 mutations were most common in Cluster 3 (Fig. 1D,E; Appendix Fig. S4G; Dataset EV8). NF2 is recognized as one of the most mutated TSGs in PM, affecting 40–50% of patients (Bueno et al, 2016; Cheng et al, 1999; Hmeljak et al, 2018; Mangiante et al, 2023). Prior research has shown that NF2 loss contributes to mesothelioma development (Fleury-Feith et al, 2003; Jongsma et al, 2008; Sekido and Sato, 2023). In line with that, our CRISPR-mediated knockout experiments confirmed that NF2 depletion enhanced tumor aggressiveness in vitro and in vivo (Fig. EV1A–G). Analysis of TCGA series indicated that PM patients with genetic alterations of NF2 had poorer overall survival,

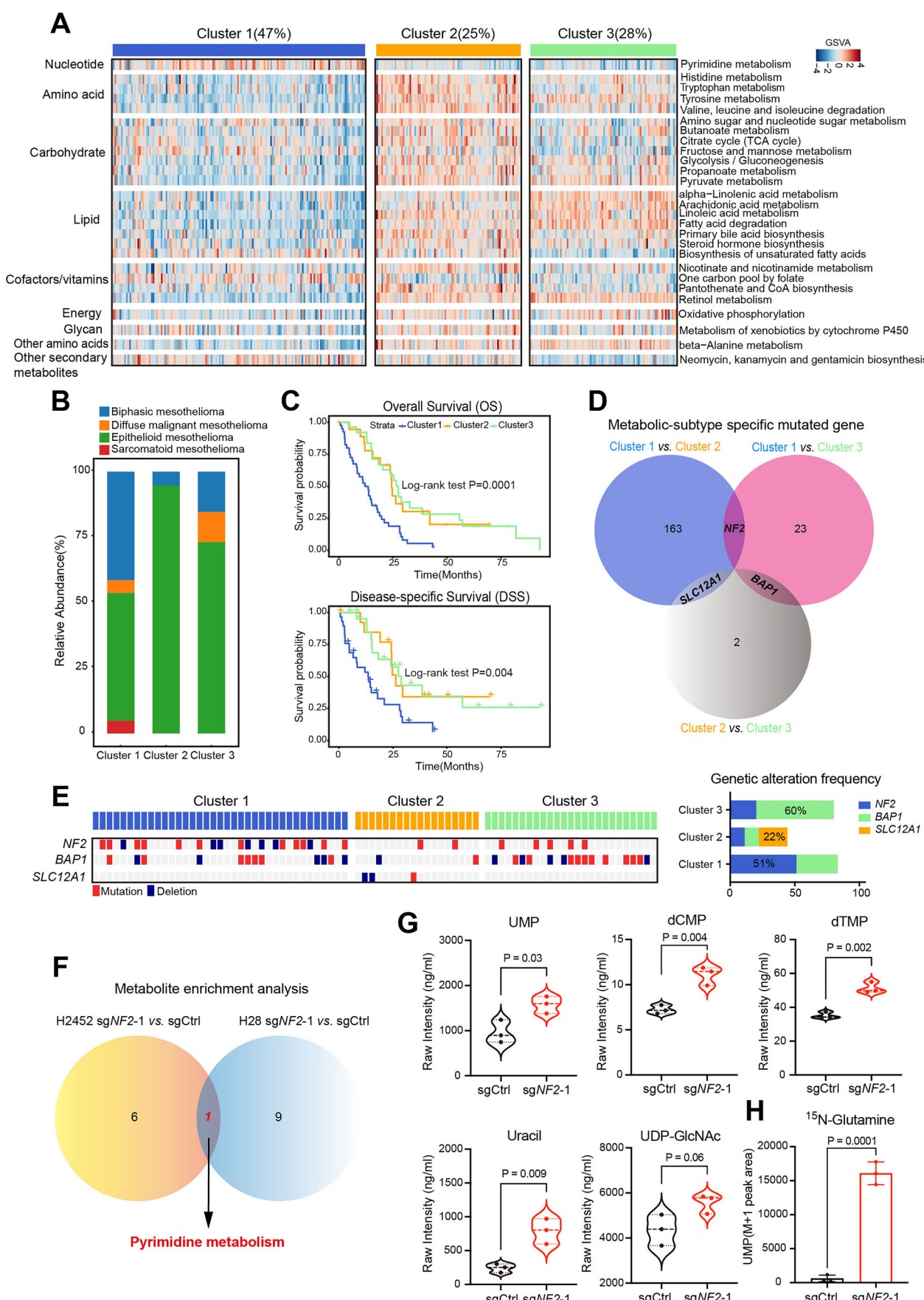

**Figure 1. Clinical and genomic characterization of metabolic subgroups in PM.**

(A) Heatmap showing the gene set variation analysis (GSVA) enrichment scores for the three metabolic pathway-based clustering subgroups, derived from the transcriptomic data of PM tumors ($n = 328$). Each bar represents an individual patient sample. A comparison of GSVA enrichment scores among the metabolic subgroups was conducted using the Wilcoxon test, and the metabolic pathways with a $P$ value < 0.05 are presented ($n = 27$). (B) Bar plots illustrating the distribution of histological classifications among the three metabolic subgroups. Clinical information was extracted from the TCGA-MESO cohort ($n = 87$). (C) Kaplan–Meier univariate survival analyses of PM across the three metabolic subtypes. Overall survival (top) and disease-specific survival (bottom) are presented. Clinical information was extracted from the TCGA-MESO cohort ($n = 87$). The $P$ value was calculated using the log-rank test in R. (D) Venn diagram illustrating the comparison of significantly mutated genes across each pair of metabolic subtypes (adjusted $P$ value < 0.05). Among these, the mutated genes specific to each metabolic subgroup are highlighted: Cluster 1-*NF2*, Cluster 2-*SLC2LA11*, and Cluster 3-*BAP1*. Genetic information was extracted from the TCGA-MESO cohort ($n = 87$). (E) The genetic status of *NF2*, *BAP1*, and *SLC12A1* among different metabolic subgroups in the TCGA-MESO cohort ($n = 87$). Detailed information regarding the frequencies of genetic alterations across the three subgroups is shown in the right panel. The genetic mutation data were downloaded from the UCSC Xena online data portal. (F) Venn diagram illustrating the shared metabolite enrichment pathways in two independent *NF2*-deficient PM cell lines. The metabolite set enrichment analysis (MSEA) was conducted using differentially expressed metabolites identified between *NF2*-knockout (sg*NF2*-1) group and the scrambled control (sgCtrl) group, with a $P$ value < 0.05 and a variable importance in projection (VIP) > 1. Then, the significantly enriched metabolic pathways with $P$ value < 0.05 were identified. The analysis included six biological replicates for H2452, and three biological replicates for H28. (G) Absolute intensity levels of differentially expressed metabolites involved in pyrimidine biosynthesis between the H2452 sg*NF2*-1 and sgCtrl groups. The data are presented as the mean ± SD ($n = 3$; three biological repeats). A two-tailed unpaired $t$ test was used for statistical analysis. (H) The differential abundance of $^{15}$N-glutamine-labeled UMP between the H2452 sg*NF2*-1 and sgCtrl groups. The data are presented as the mean ± SD ($n = 3$; three biological repeats). A two-tailed unpaired $t$ test was used for statistical analysis. Source data are available online for this figure.

although this did not reach statistical significance (Fig. EV1H). Notably, survival analysis conducted on our internal PM cohort validated that low protein levels of NF2 were significantly linked to poor prognosis (Fig. EV1I), reinforcing the clinical relevance of *NF2*.

Given the oncogenic role of *NF2* loss in PM, the prevalence of *NF2* alterations in Cluster 1, and the poor prognosis associated with this metabolic subtype, we further investigated the role of *NF2* in tumor metabolism. Our untargeted metabolomic profiling identified pyrimidine metabolism as the only consistently altered metabolic process in two independent *NF2*-deficient PM cell lines (Figs. 1F and EV2A,B; Table EV1). This observation was further supported by global metabolomic profiling of pan-cancer cell lines, where pyrimidine metabolites were enriched in tumors with low NF2 protein levels (Fig. EV2C,D; Dataset EV9). Furthermore, targeted metabolomic analysis confirmed that *NF2* depletion led to a robust accumulation of key metabolites involved in de novo pyrimidine synthesis (Figs. 1G and EV2E; Table EV2). Notably, $^{15}$N-glutamine isotype tracing experiments validated a significant increase in $^{15}$N-labeled UMP levels in *NF2*-knockout cells compared to controls, supporting an enhanced activity of pyrimidine nucleotide synthesis (Fig. 1H). Overall, our multi-faceted molecular and metabolic analyses establish that *NF2* deficiency defines a distinct subtype of PM characterized by elevated de novo pyrimidine synthesis. These results provide new insights into *NF2*-driven metabolic reprogramming and highlight pyrimidine biosynthesis as a potential therapeutic vulnerability in *NF2*-deficient PM.

### *NF2* loss stimulates de novo pyrimidine synthesis in PM

To elucidate the molecular mechanism by which *NF2* loss drives de novo pyrimidine synthesis, we conducted unbiased RNA sequencing and proteomic profiling on wild-type and *NF2*-deficient PM cells (Fig. 2A). Given that *NF2* loss can induce tumor-specific dependencies on other hyperactive genes and proteins, which may serve as synthetic lethal therapeutic targets (Savage et al, 2024; Yang et al, 2020). In the subsequent integrated analysis of the transcriptome and proteome, we focused on the common candidates that exhibited upregulation following *NF2* deletion.

The top candidates identified in the transcriptome were *PCK2, MME*, and *MGLL*, while *PM20D2, MME*, and *WWC1* were most prominent in the proteome (Figs. 2B and EV3A; Datasets EV10 and 11).

Notably, the candidates regulated by *NF2* exhibit a notable enrichment in several metabolic processes, such as amino acid biosynthesis, one carbon pool by folate, alanine/aspartate/glutamate metabolism, and pyrimidine metabolism (Fig. 2C). Importantly, CAD and DHODH, the critical enzymes for de novo pyrimidine biosynthesis (Wang et al, 2019), were significantly increased at both transcript and protein levels in *NF2*-deficient PM cells (Figs. 2D and EV3B,C). This finding was further confirmed by qRT-PCR and western blot (Figs. 2E,F and EV3D). Moreover, the enzymatic activity of DHODH was markedly elevated in *NF2*-deficient cells compared to wild-type counterparts (Fig. EV3E). While *UMPS* mRNA levels remained stable, its protein expression showed an increase following *NF2* deletion (Fig. EV3B–D), supporting the idea that *NF2* loss promotes the de novo pyrimidine synthesis pathway.

Indeed, we observed that H2052 mesothelioma cells carrying *NF2* mutations, exhibited higher levels of CAD and DHODH compared to normal cells and other PM cell lines with wild-type *NF2* (Fig. 2G). IHC analysis of mesothelioma tissue microarray consistently showed that CAD and DHODH protein levels were generally higher in tumors than in matched normal tissues (Figs. 2H and EV3F). Notably, PM patients with high expression of CAD and DHODH had a poorer prognosis (Fig. EV3G). However, no significant correlation was identified between NF2 and CAD expression in the Nanjing cohort (Fig. 2I,J), while in the Bern cohort, patients with low NF2 levels (NF2_low) showed an increase in CAD expression (Fig. 2L,M). Meanwhile, DHODH expression was notably elevated in NF2_low group across both cohorts (Fig. 2K,N), reinforcing the link between *NF2* loss and enhanced pyrimidine biosynthesis.

These results indicate that *NF2* deletion modulates de novo pyrimidine metabolism by enhancing the expression of pyrimidine biosynthetic enzymes. Given the importance of pyrimidine synthesis in malignant transformation and the fact that this process is typically non-essential for most mammalian tissues' homeostasis, de novo pathway enzymes present logical targets for

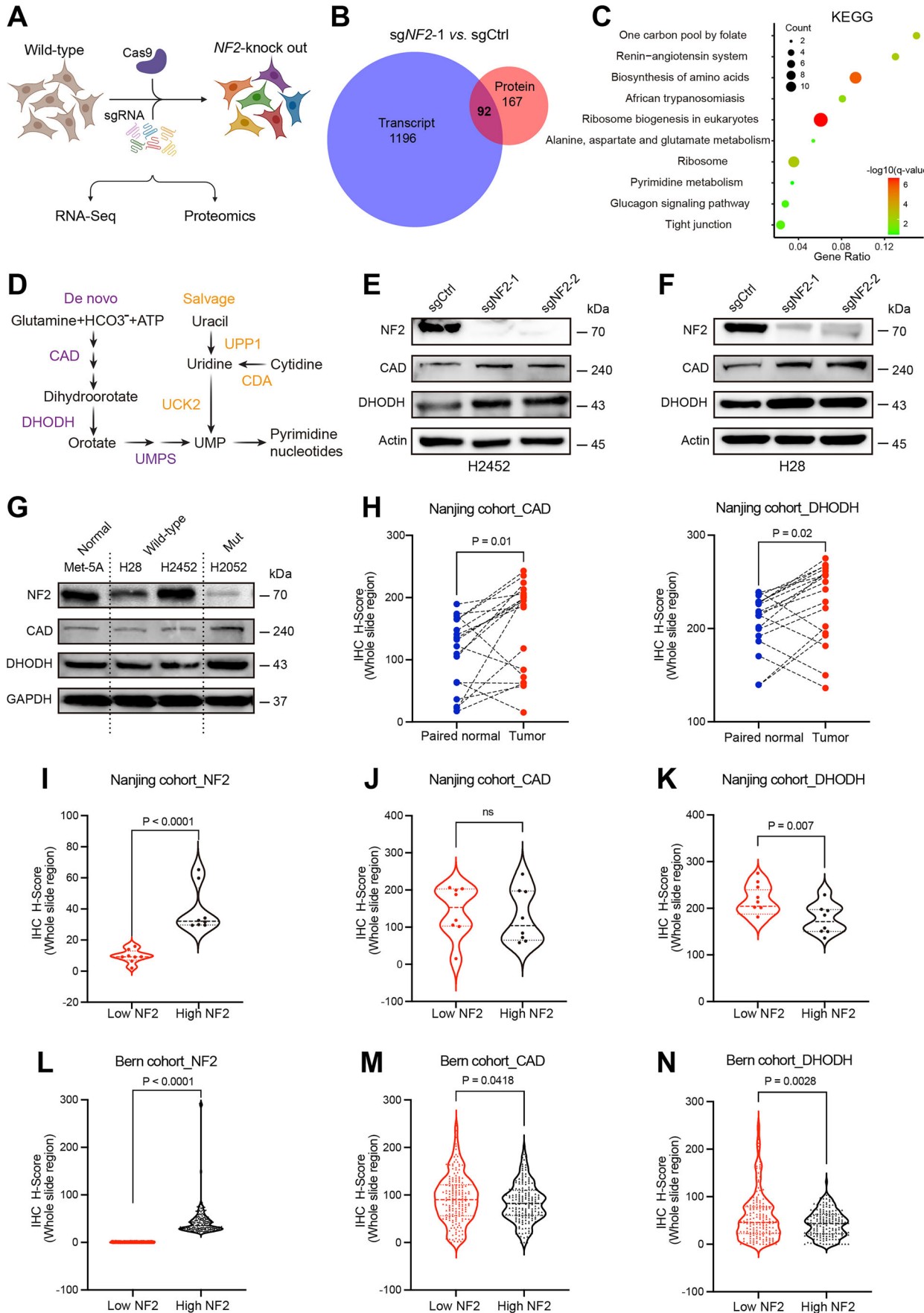

◀  **Figure 2.   Deletion of *NF2* rewires the de novo pyrimidine synthesis pathway.**

(A) Schematic model of the experimental design is presented. After the H2452 PM cell line was transfected with a scrambled control (sgCtrl) or an *NF2*-targeting sgRNA (sg*NF2*-1), RNA sequencing and label-free proteomics analyses were conducted on both populations. (B) Venn diagram illustrating the overlap between the differentially upregulated genes (*P* value < 0.05 and fold change >1.4) and proteins (*P* value < 0.05 and fold change>1.3) in the H2452 sg*NF2*-1 group compared to the sgCtrl group. (C) Kyoto Encyclopedia of Genes and Genomes (KEGG) enrichment analysis was conducted on 92 common candidates shared by the differentially upregulated genes and proteins in the H2452 sg*NF2*-1 group. The gene ratio represents the number of dysregulated candidates relative to the total number of annotated genes within the specified pathway. (D) Schematic model of the de novo and salvage pathways for pyrimidine biosynthesis. (E, F) Immunoblots of the indicated proteins in the H2452 (E) and H28 (F) pleural mesothelioma (PM) cell lines transfected with a scrambled control (sgCtrl) or *NF2*-targeting sgRNAs (sg*NF2*-1, sg*NF2*-2). Representative images from three independent experiments are presented. The quantification data are displayed in Appendix Fig. S5A,B. (G) Immunoblots of the indicated proteins in normal mesothelial cells (Met-5A) and PM cell lines (*NF2* wild-type: H28 and H2452; *NF2* mutant: H2052). Representative images from three independent experiments are presented. The quantification data are displayed in Appendix Fig. S5D. (H) The differences in the protein levels of CAD (left) and DHODH (right) between paired normal tissue and tumor samples from the internal mesothelioma tissue microarray (Nanjing cohort; *n* = 16). A two-tailed paired *t* test was used for statistical analysis. (I–K) The differences in the protein levels of NF2 (I), CAD (J), and DHODH (K) between the low- and high-NF2 subgroups, stratified according to the top and bottom quartiles of the IHC-H score in the internal mesothelioma tissue microarray (Nanjing cohort; *n* = 17; 34 punches). A two-tailed unpaired *t* test was used for statistical analysis. (L–N) The differences in the protein levels of NF2 (L), CAD (M), and DHODH (N) between the low- and high-NF2 subgroups, stratified according to the top and bottom quartiles of the IHC-H score in the internal mesothelioma tissue microarray (Bern cohort; *n* = 98; 368 punches). A nonparametric Kolmogorov–Smirnov test was performed for statistical analysis. Source data are available online for this figure.

cancer therapy (Mullen and Singh, 2023). Thus, we next sought to investigate whether there is a specific dependency on pyrimidine biosynthesis in *NF2*-deficient PM.

## De novo pyrimidine synthesis as a targetable vulnerability in *NF2*-deficient PM

Genetic silencing of either *CAD* or *DHODH* impaired the viability of PM cells, notably resulting in apparent growth inhibition in *NF2*-deletion group (Figs. 3A,B and EV4A–D). Supporting this observation, data from DepMap(Tsherniak et al, 2017) indicated that PM cells with low NF2 levels showed increased sensitivity to genetic knockout of *DHODH* (Fig. EV4E). In addition, pharmacological inhibition of DHODH exerted a robust antiproliferative effect on *NF2*-deletion PM cells, as evidenced by reduced cell viability, colony formation, and tumor spheroid formation (Fig. 3C–G). And rescue experiments utilizing sgRNA-resistant NF2 successfully suppressed CAD/DHODH expression and restored cell viability in the context of DHODH inhibition (Fig. EV4F–I). Moreover, both commercial mesothelioma cell lines and primary patient-derived cells (PDCLs) with inherent NF2 deficiency exhibited greater sensitivity to DHODH inhibition compared to NF2-intact counterparts (Figs. 3H,I and EV4J–M). Further analysis of the drug sensitivity of Leflunomide, an FDA-approved agent targeting DHODH, revealed a positive correlation between NF2 protein levels and drug sensitivity typically existed in pleural mesothelioma across pan-cancer cell lines, indicating that this association might be tissue context-specific (Fig. EV4N,O; Table EV3).

To verify that the antitumor effects of DHODH inhibition are specifically due to the blockade of de novo pyrimidine synthesis, we conducted exogenous uridine rescue experiments. The addition of uridine at reasonable concentrations fully rescued DHODH inhibitor-induced cell viability suppression (Fig. 4A). Moreover, we confirmed that DHODH inhibition led to obvious S-phase cell cycle arrest and DNA damage, in both commercial and primary cells with defective NF2 (Fig. 4B–E). We also noted an increase in apoptotic cell death among commercial *NF2*-deficient cell lines treated with DHODH inhibitors, with PDCLs exhibiting an even more pronounced level of apoptosis (Fig. 4F,G). This observation suggests that DHODH inhibition may exert cytostatic

effects in commercial cell lines, while exhibiting cytotoxic effects in PDCLs.

Our research further revealed that *NF2*-deficient PM cells exhibited increased resistance to the inhibition of inosine monophosphate dehydrogenase (IMPDH), an enzyme involved in purine biosynthesis (Fig. EV5A). Moreover, targeting pyrimidine salvage pathway with a UCK2 (uridine-cytidine kinase 2) inhibitor fails to discriminate *NF2*-deficient from wild-type cells under both normal and glucose-deprivation stress conditions, a state known to force cells to rely on the salvage pathway (Skinner et al, 2023) (Fig. EV5B–E). Given that mesothelioma is poorly vascularized, resulting in limited glucose availability in situ (Ohta et al, 1999; Ostergaard et al, 2013), we further assessed the sensitivity to DHODH inhibitors under glucose-limited conditions. Glucose restriction did not alter drug sensitivity, indicating that *NF2*-deficient PM cells maintain their reliance on pyrimidine biosynthesis even under metabolic stress (Fig. EV5F–H). Although *NF2*-deletion cells displayed increased uridine phosphorylase 1 (UPP1) expression, another mediator of pyrimidine salvage pathway, the knockdown of *UPP1* did not suppress the enhanced proliferation driven by *NF2* loss (Fig. EV5I–K). These findings align with the notion that highly proliferative cancer cells require de novo pyrimidine biosynthesis, while resting or differentiated cells rely on the salvage pathway (Wang et al, 2021). Thus, our results indicate that de novo pyrimidine biosynthesis is a cancer-dependent process in *NF2*-deficient PM.

## NF2-YAP axis contributes to the rewiring of pyrimidine metabolism

We next asked how *NF2* deficiency leads to the upregulation of CAD and DHODH. Given the increase in CAD and DHODH at both the transcript and protein levels upon loss of *NF2* (Figs. 2 and EV3), we reasoned that *NF2* deletion may induce their transcriptional upregulation. As expected, *NF2* deficiency impaired Hippo signaling (Fig. 5A), a canonical NF2-regulated pathway that is often disrupted in PM. *NF2* negatively regulates the Hippo pathway by promoting LATS1/2-dependent phosphorylation of Yes-associated protein (YAP), which is a target of proteasomal degradation, thereby blocking its role as a transcriptional coactivator (Dey et al, 2020; Sekido and Sato, 2023). Our prior study indicated that PM

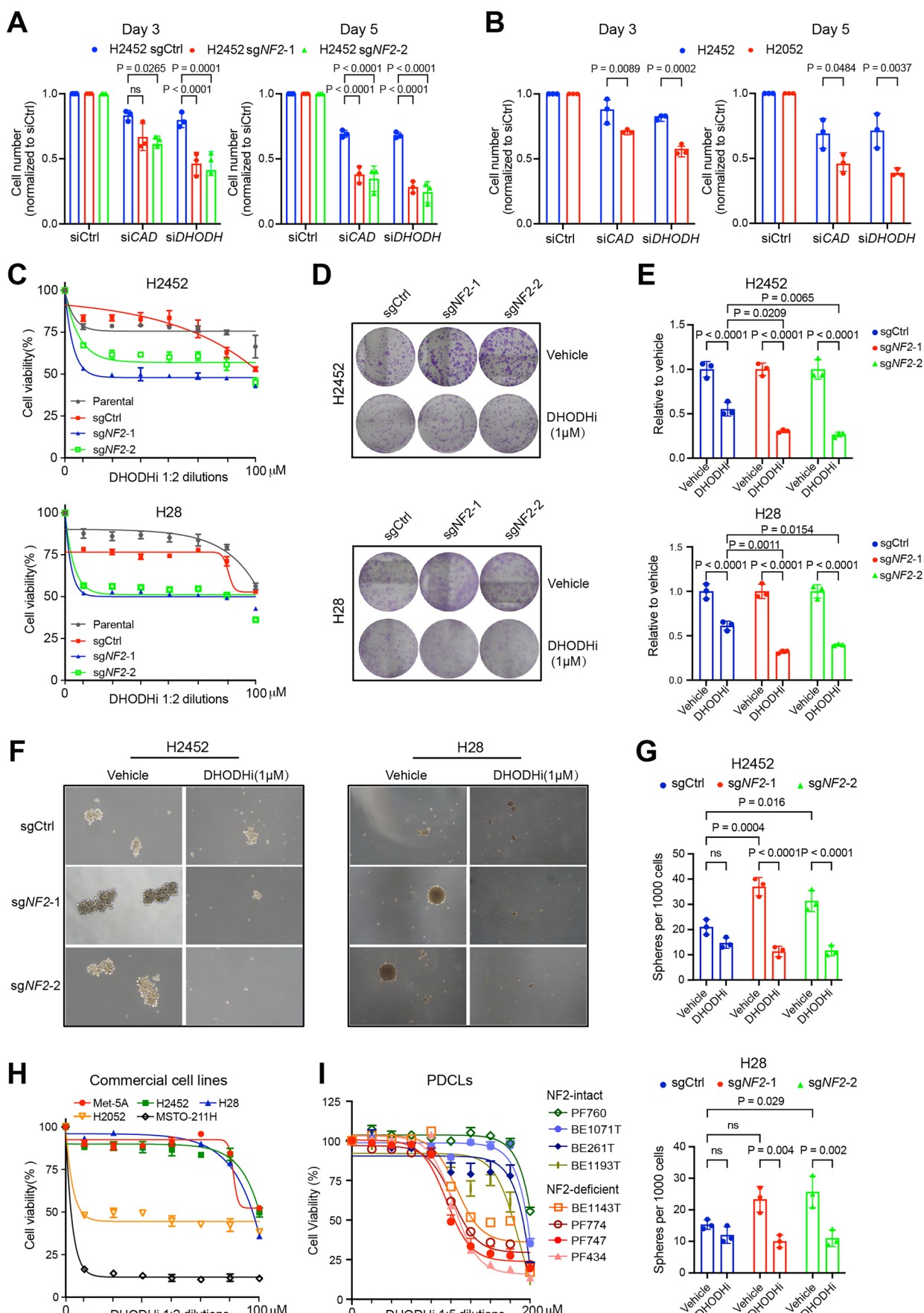

**Figure 3. A specific dependence on the de novo pyrimidine metabolic process in *NF2*-deficient PM.**

(A) Cell viability of the indicated cell populations transfected with small interfering RNA (siRNA) targeting *CAD* or *DHODH*. The cell numbers were calculated after 3 days (left) and 5 days (right) of transfection and were normalized to the negative control (siCtrl). The data are presented as the mean ± SD ($n = 3$; three biological repeats). Two-way ANOVA with multiple comparisons was used for statistical analysis. (B) Cell viability of H2452 (*NF2* wild-type) and H2052 (*NF2* mutant) PM cells transfected with siRNA targeting *CAD* or *DHODH*. The cell numbers were calculated after 3 days (left) and 5 days (right) of transfection and normalized to siCtrl. The data are presented as the mean ± SD ($n = 3$; three biological repeats). Two-way ANOVA with multiple comparisons was used for statistical analysis. (C) Parental, sgCtrl, and *NF2*-knockout (sg*NF2*-1, sg*NF2*-2) H2452 (top) and H28 (bottom) PM cells were treated with increasing doses of the DHODH inhibitor Brequinar. Cell viability was measured 120 h after treatment. The data are presented as the mean ± SD. Representative results from three independent experiments are shown. (D, E) Clonogenic assay of parental, sgCtrl, and *NF2*-knockout (sg*NF2*-1, sg*NF2*-2) H2452 (top) and H28 (bottom) PM cells treated with vehicle or the DHODH inhibitor (1 μM) for 120 h. After a 14-day culture period, viable cells were stained with crystal violet dye. Representative images (D) and quantification (E) from three independent experiments are shown. The data are presented as the mean ± SD. Two-way ANOVA with multiple comparisons was used for statistical analysis. (F, G) Sphere formation assay of sgCtrl, *NF2*-knockout (sg*NF2*-1, sg*NF2*-2), H2452 (left), and H28 (right) PM cells treated with vehicle or the DHODH inhibitor (1 μM) for 96 h, followed by continuous culture for 14 days. Representative images (F) at the end of the experiment, along with the quantification (G). The data are presented as the mean ± SD ($n = 3$; three biological repeats). Two-way ANOVA with multiple comparisons was used for statistical analysis. (H) Cell viability of normal mesothelial cells (Met-5A) and commercial PM cell lines (*NF2* wild-type: H2452, H28, and MSTO-211H; *NF2* mutant: H2052) treated with increasing doses of the DHODH inhibitor Brequinar for 120 h. The data are presented as the mean ± SD. Representative results from three independent experiments are shown. (I) Cell viability assay of primary-derived cell lines (PDCLs) following treatment with various doses of the DHODH inhibitor Brequinar for 120 h. The data are presented as the mean ± SEM from three or four independent experiments. Source data are available online for this figure.

cells with mutations in *NF2/LATS1/2* showed low levels of YAP_pS127 and increased activity of oncoprotein YAP (Xu et al, 2021). Indeed, *NF2* deficiency strongly decreased YAP_S127 in PM cells, while restoring NF2 expression reversed these changes (Figs. 5B,C and EV6A,B).

Supporting this model, mesothelioma patients with low levels of NF2 showed a decreased YAP_pS127 expression across two independent cohorts (Fig. 5D,E). Moreover, low levels of YAP_pS127 were associated with poor clinical outcomes (Fig. EV6C). Notably, an analysis of the TCGA cohort of PM patients revealed a positive correlation between YAP mRNA levels and CAD, while showing a negative correlation with enzymes involved in the pyrimidine salvage pathway (UPP1, CDA, and UCK2) (Fig. EV6D). This suggests that tumors with active YAP1 preferentially utilize the de novo synthesis pathway over the salvage pathway. However, no significant correlation was found between YAP1 and the mRNA levels of DHODH or UMPS, likely due to post-transcriptional regulation, protein stability, or cell state-specific factors, which are known limitations of transcriptomic analyses.

Building on these observations, we proceeded to investigate whether YAP exerts a direct influence on the expression of CAD and DHODH. Both genetic and pharmacological inhibition of YAP resulted in a substantial reduction in the transcript and protein levels of these markers in *NF2*-deletion PM cells (Fig. 5F–H). This inhibition markedly diminished the sensitivity of these *NF2*-deficient cells to the DHODH inhibitor (Fig. 5I,J), indicating that YAP is both essential and sufficient for the reliance of *NF2*-deficient PM cells on de novo pyrimidine biosynthesis. Analysis of ChIP-Seq data from mesothelioma cell lines (Zheng et al, 2019) indicated that both CAD and DHODH are potential targets of the YAP-TEAD1 transcriptional complex (Li et al, 2019) (Fig. EV6E,F). This finding was corroborated by our ChIP-qPCR experiments, which revealed the presence of YAP-binding regions within the CAD and DHODH genes (Figs. 5K and EV6G,H). Reinforcing ChIP-qPCR results, a dual-luciferase reporter assay provided additional evidence that PM cells with *NF2* mutations, correlating with elevated YAP activity, exhibited robust increases in signal intensity from both the CAD and DHODH predicted binding regions, while no such increase were detected in *NF2* wild-type cells characterized by low YAP

activity (Fig. 5L). These results clarify that the loss of *NF2* activates YAP, which directly regulates the transcription of CAD and DHODH, thereby promoting de novo pyrimidine metabolism in *NF2*-deficient PM cells. This YAP-driven metabolic rewiring provides a potentially targetable vulnerability in *NF2*-deficient mesothelioma.

## Targeting de novo pyrimidine metabolism suppresses *NF2*-deficient tumor growth

Consistent with the tumor-suppressive role of *NF2*, genetic knockout of *NF2* dramatically increased the tumor burden in both subcutaneous and orthotopically transplanted PM mouse models (Fig. 6A–C). Metabolomic profiling and IHC analysis of residual tumors indicated a substantial enrichment of pyrimidine metabolism within *NF2*-deletion group, with concomitant decreased YAP_pS127 and increased CAD and DHODH (Fig. EV7A–C; Dataset EV12). As expected, DHODH inhibition significantly suppressed tumor growth and reduced Ki-67-positive cells in *NF2*-deficient tumors, while having minimal effects on wild-type tumors. This intervention resulted in a prolonged survival benefit in mice bearing orthotopically transplanted *NF2*-deficient PM tumors (Figs. 6A–C and EV7D–F). Similarly, in mesothelioma cells with inherent *NF2* deletions, DHODH inhibition led to a notable reduction in tumor growth (Fig. 6D). Importantly, primary-derived xenograft (PDX) mesothelioma models with inherent NF2 deficiency exhibited a stronger response to DHODH inhibition compared to NF2-intact group, further underscoring the therapeutic vulnerability of NF2-deficient PM (Figs. 6E–G and EV7G,H).

Targeting DHODH has been shown to improve sensitivity to cisplatin (Jiang et al, 2023). Our research revealed that the combination of cisplatin and DHODH inhibition exerted a synergistic effect in PM cell line with *NF2* mutations, yielding a mean synergy score over 10, while no effect was achieved in the wild-type group (Fig. EV8A–D). Similar synergistic results were observed in PM cells with genetic modifications of *NF2* (Fig. EV8E,F). In vivo experiments further validated that the combined treatment of cisplatin and DHODH inhibitor ta reduced tumor growth in *NF2*-deficient group, without inducing notable

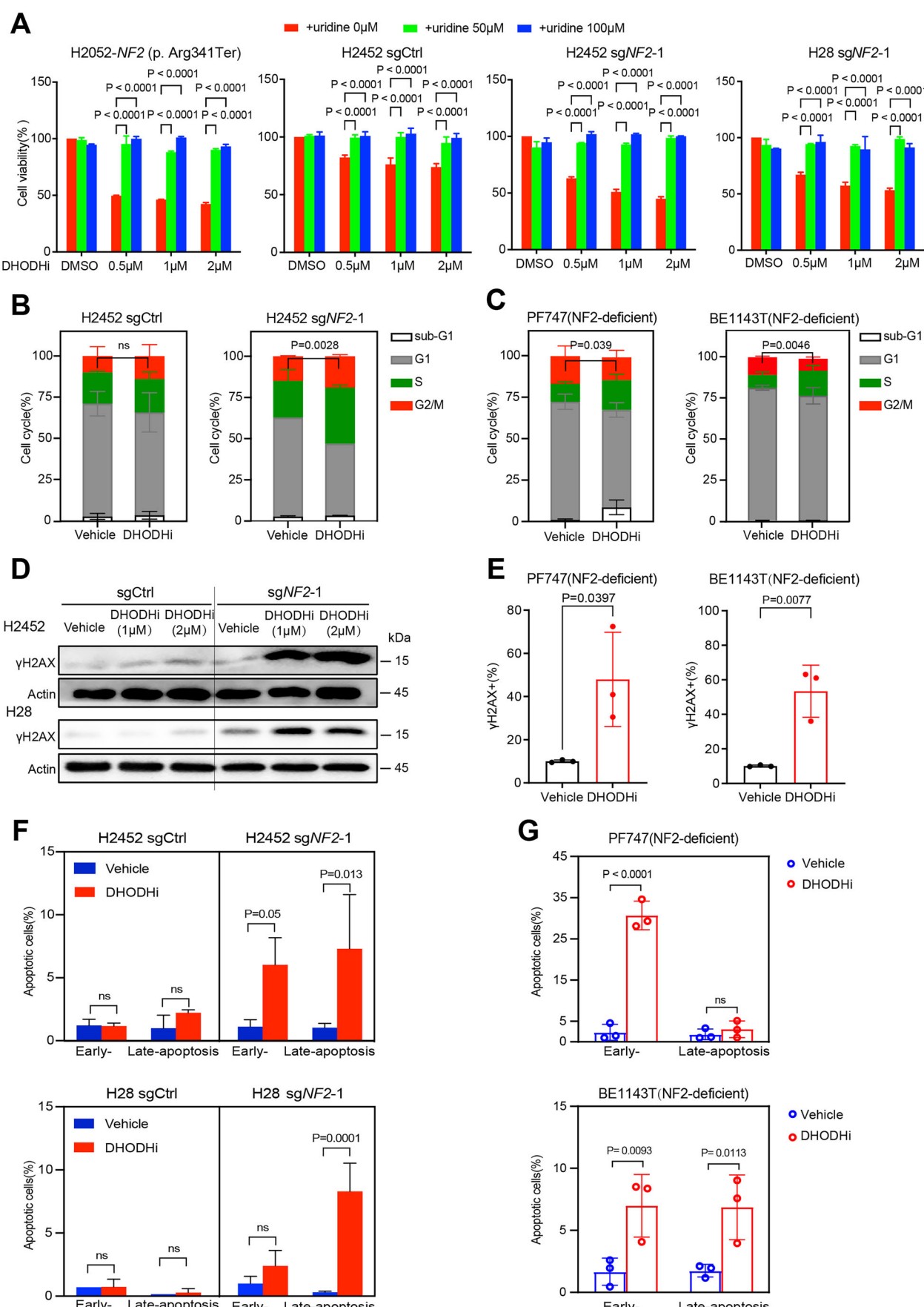

◄  **Figure 4.   Blocking de novo pyrimidine synthesis triggers cell cycle arrest, DNA Damage, and apoptotic cell death in *NF2*-deficient PM.**

(A) Cell viability of the indicated cell populations treated with the indicated doses of the DHODH inhibitor Brequinar, with or without uridine supplementation (50 μM or 100 μM) for 72 h. The data are presented as the mean ± SD (n = 3; three biological repeats). Two-way ANOVA with multiple comparisons was used for statistical analysis. (B) Cell cycle analysis of H2452 sgCtrl and *NF2*-knockout (sg*NF2*-1) PM cells treated with 1 μM DHODH inhibitor Brequinar for 24 h. The data are presented as the mean ± SD (n = 3; three biological repeats). Two-way ANOVA with multiple comparisons was used for statistical analysis. (C) Cell cycle analysis of PDCLs with inherent NF2 deficiency (PF747 and BE1143T) following 24 h of treatment with the DHODH inhibitor Brequinar. PF747 was treated with 2 μM, and BE1143T was treated with 10 μM DHODH inhibitor. The data are presented as the mean ± SD (n = 3; three biological repeats). Two-way ANOVA with multiple comparisons was used for statistical analysis. (D) Immunoblots of extracts from H2452 (top) and H28 (bottom) sgCtrl and *NF2*-knockout (sg*NF2*-1) PM cells treated with the indicated doses of the DHODH inhibitor Brequinar for 48 h. Representative images from three independent experiments are shown. The quantification data are displayed in Appendix Fig. S5F. (E) The percentage of γH2AX-positive cells in two NF2-deficient PDCLs PF747 and BE1143T, after 24 h of treatment with the DHODH inhibitor Brequinar. PF747 was treated with 2 μM, and BE1143T was treated with 10 μM DHODH inhibitor. The gate for the vehicle groups was set at approximately 10% to represent the basal level of DNA damage. The data are presented as the mean ± SD (n = 3; three biological repeats). A two-tailed unpaired t test was used for statistical analysis. (F) Flow cytometry-based apoptotic assay in H2452 (top)&H28 (bottom) sgCtrl and *NF2*-knockout (sg*NF2*-1) PM cell lines treated with the DHODH inhibitor Brequinar for 72 h. The bar graph shows the percentage of apoptotic cells that underwent early or late apoptosis induced by 1 μM Brequinar. The data are presented as the mean ± SD (n = 3; three biological repeats). Two-way ANOVA with multiple comparisons was used for statistical analysis. (G) The percentages of apoptotic cell death in two NF2-deficient PDCLs PF747 (top) and BE1143T (bottom), after 72 h of treatment with the DHODH inhibitor Brequinar. PF747 was treated with 2 μM, and BE1143T was treated with 10 μM DHODH inhibitor. The data are presented as the mean ± SD (n = 3; three biological repeats). Two-way ANOVA with multiple comparisons was used for statistical analysis. Source data are available online for this figure.

toxicity (Figs. 6H and EV8G). Given that platinum-based chemotherapy is a vital treatment option for PM patients, this approach may represent an adjunctive strategy to improve drug response and merits further investigation.

Our preclinical results, encompassing both commercial and primary mesothelioma models, indicate that the de novo pyrimidine biosynthesis pathway is a promising synthetic lethal target in *NF2*-deficient PM.

## Discussion

In this study, we report that PM features highly heterogeneous metabolic profiles, among which upregulated pyrimidine metabolism was identified as a distinct metabolic subtype linked to unfavorable clinical outcomes. Further genomic analysis revealed that *NF2* deficiency is a metabolic subtype-specific genetic alteration, characterized by elevated de novo pyrimidine biosynthesis. Notably, we validated that *NF2*-deficient PM cells exhibit a heightened reliance on de novo pyrimidine metabolism for rapid cell proliferation, rendering them particularly vulnerable to DHODH inhibition in both in vitro and preclinical mouse models. Collectively, these findings enrich the understanding of the metabolic diversity within PM and open new avenues for precision oncology.

Despite numerous investigations into cancer metabolism reprogramming in PM (Urso et al, 2020), effective treatments for this cancer have yet to be developed, underscoring the urgent need for a comprehensive understanding of PM's metabolic diversity. Through a systematic analysis of metabolic heterogeneity, we identified pyrimidine metabolism as a prominent metabolic subtype, affecting approximately half of PM patients and correlating with frequent genetic alterations in *NF2*. Moreover, our results corroborate previous findings indicating that *BAP1*-mutant PM is susceptible to lipid metabolism inhibition (Pandey et al, 2023), as we observed that PM patients in Cluster 3, closely associated with *BAP1* loss, exhibited increased linoleic acid, arachidonic acid, and ether lipid metabolism. Interestingly, genetic alterations in *SLC12A1*, a recently identified oncogene (Teng et al, 2016), were frequently detected in Cluster 2, which is characterized by a mixed

type of dysregulation across major metabolic categories, potentially complicating targeted interventions. Until recently, the role of *SLC12A1* in tumor metabolism has been poorly understood, warranting further investigation into its coordination of tumorigenesis in PM. Although *TP53* loss is known to have profound metabolic consequences and occurs in up to 20% of mesothelioma cases (Hmeljak et al, 2018; Liu et al, 2019), no direct association with metabolic subgroups was identified in our study.

Pyrimidine biosynthesis is essential for tumor growth as it supports the production of substantial amounts of cellular building blocks required for cell proliferation (Bajzikova et al, 2019; Wang et al, 2021; Yang et al, 2023). Intriguingly, a recent study revealed that DHODH-mediated pyrimidine synthesis plays a role in suppressing mitochondrial lipid peroxidation and ferroptosis, suggesting it may contribute to tumor survival through an additional defense mechanism (Mao et al, 2021). Our findings demonstrate that de novo pyrimidine biosynthesis represents a collateral metabolic vulnerability induced by *NF2* deficiency in PM, reinforcing the notion that specific oncogenic mutations can rewire the canonical pyrimidine metabolism pathway and may thus be targeted therapeutically. We previously demonstrated that cancer cells with low NF2 display increased sensitivity to pyrimidine analogs, including cytarabine and 5-fluorouracil (Yang et al, 2021). Expanding on this, we now show that blocking de novo pyrimidine synthesis in *NF2*-deficient PM cells leads to persistent cell cycle arrest, DNA damage, and apoptotic cell death, explaining the preferential antitumor effects of DHODH inhibitors in *NF2*-deficient PM. Notably, we observed differential antitumor responses between commercial cell lines and PDCLs. This discrepancy highlights a potential divergence in metabolic dependencies between established cell lines and PDCLs (Chernova et al, 2016), the underlying molecular differences warrant further investigation.

Although we observed an increased abundance of purine nucleotides, including guanosine and inosine, following the deletion of *NF2*. Our results indicate that *NF2*-deficient PM cells exhibit a phenotype resistant to purine biosynthesis blockade, suggesting that the observed increase in purine nucleotides may be a secondary effect resulting from enhanced pyrimidine biosynthesis. Furthermore, we confirmed that glucose deficiency, a stress

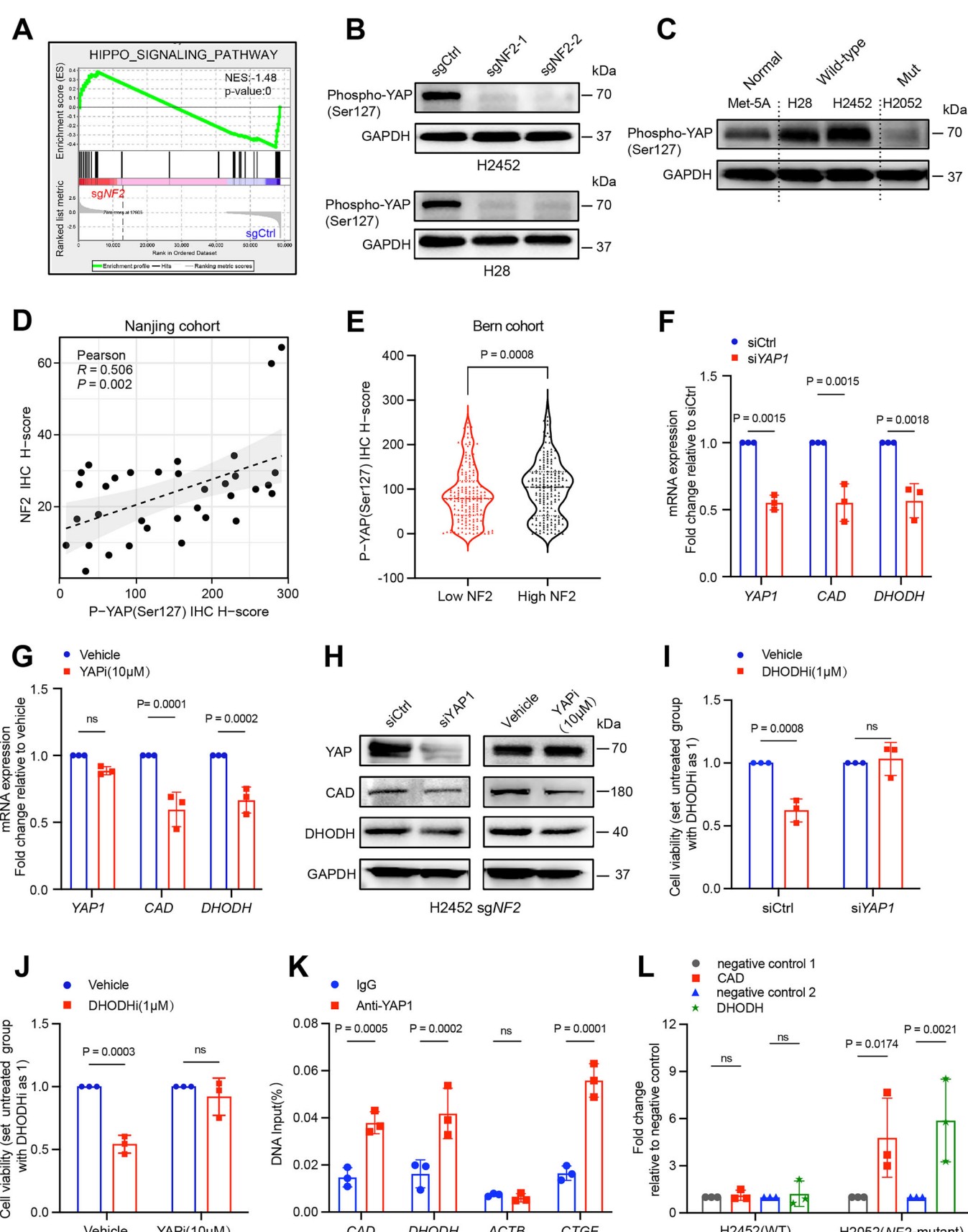

**Figure 5. The canonical NF2-YAP axis facilitates the rewiring of de novo pyrimidine synthesis by the transcriptional regulation of CAD and DHODH.**

(A) Gene Set Enrichment Analysis (GSEA) plot showing the dysregulation of the Hippo signaling pathway in H2452 sg*NF2*-1 group compared to the sgCtrl group. The *P* value is calculated using a permutation test. (B) Immunoblots of the indicated proteins in H2452 (top) and H28 (bottom) PM cell lines transfected with a scrambled control (sgCtrl) or *NF2*-targeting sgRNAs (sg*NF2*-1, sg*NF2*-2). Representative images from three independent experiments are shown. The quantification data are displayed in Appendix Fig. S5C. (C) Immunoblots of the indicated proteins in normal mesothelial cells (Met-5A) and commercial PM cell lines (*NF2* wild-type: H28 and H2452; *NF2* mutant: H2052). Representative images from three independent experiments are shown. The quantification data are displayed in Appendix Fig. S5E. (D) Scatter plot illustrating the correlation analysis between the IHC H-score of NF2 and phosphorylated YAP (Ser127) based on the tissue microarray of the internal mesothelioma patient cohort (Nanjing cohort; $n = 17$; 34 punches). (E) The difference in the protein levels of phosphorylated YAP (Ser127) between the low- and high-NF2 subgroups, stratified according to the top and bottom quartiles of the IHC-H score in the internal mesothelioma tissue microarray (Bern cohort; $n = 98$; 368 punches). A nonparametric Kolmogorov–Smirnov test was performed for statistical analysis. (F, G) The mRNA levels of YAP1, CAD, and DHODH in H2452 sg*NF2*-1 cells transfected with the indicated siRNAs (F) or treated with the specific YAP-TEAD inhibitor 1 (Peptide 17) for 48 h (G). The data are presented as the mean ± SD ($n = 3$; three biological repeats). Two-way ANOVA with multiple comparisons was used for statistical analysis. (H) Immunoblots of the indicated proteins in H2452 sg*NF2*-1 cells transfected with the specified siRNAs (left) or treated with the specific YAP-TEAD inhibitor 1 (Peptide 17; right) for 48 h. Representative images from three independent experiments are shown. The quantification data are displayed in Appendix Fig. S5G. (I, J) Cell viability of H2452 sg*NF2*-1 cells transfected with the indicated siRNAs (I) or treated with the specific YAP-TEAD inhibitor 1 (Peptide 17; J) followed by treatment with the DHODH inhibitor (1 μM) for 48 h. The data are presented as the mean ± SD ($n = 3$; three biological repeats). Two-way ANOVA with multiple comparisons was used for statistical analysis. (K) ChIP-qPCR analysis of YAP1-binding regions in the CAD promoter and candidate DHODH enhancer in H2052 cells, utilizing control immunoglobulin G (IgG) or an anti-YAP1 antibody. The values represent percentages of the input. The primers used were designed based on the YAP1/TEAD1 binding peak regions identified in the ChIP-seq datasets retrieved from the Cistrome Data Browser. *ACTB* was used as a negative control, and *CTGF* served as a positive control. The data are presented as the mean ± SD ($n = 3$; three biological repeats). Two-way ANOVA with multiple comparisons was used for statistical analysis. (L) Dual-luciferase reporter assay of YAP1-binding regions in the CAD promoter and candidate DHODH enhancer in H2452 (*NF2* wild-type) and H2052 (*NF2* mutant) PM cells. The data are presented as the mean ± SD ($n = 3$; three biological repeats). Two-way ANOVA with multiple comparisons was used for statistical analysis. Source data are available online for this figure.

condition forcing cells to rely on the salvage pathway (Skinner et al, 2023), did not abrogate the elevated proliferation in *NF2*-deficient PM cells. In line with that, targeting UPP1, an enzyme that contributes to the salvage pathway by cleaving uridine into uracil, did not reduce the *NF2*-driven hyperproliferation in PM cells. Collectively, our research provides compelling evidence that *NF2*-driven PM tumorigenesis relies explicitly on de novo pyrimidine biosynthesis for cell survival.

Metabolic enzymes are rarely genetically modified and instead serve as downstream effectors of oncogenic signals (e.g., *KRAS* mutation or *PTEN* deletions) (Dey et al, 2021). A recent study indicated that *KRAS* alterations may promote tumorigenesis in a subset of mesothelioma patients and displayed a repulsive tendency with *NF2* mutations (Marazioti et al, 2022). Our analysis revealed that *KRAS* and *NF2* mutations can coexist in a small group of patients in Cluster 1, representing 26% of those with *NF2* mutations. Importantly, we observed no significant differences in pyrimidine metabolism between patients with *NF2* deficiencies who had *KRAS* mutations and those with wild-type *KRAS*. This aligns with our finding that *KRAS* knockdown does not affect the *NF2*-driven increase in CAD and DHODH expression, suggesting the alterations in pyrimidine metabolism associated with *NF2* deficiency occur independently of KRAS signaling in mesothelioma (Appendix Fig. S6). Given that NF2 is a well-established inhibitor of RAS, and that RAS signaling has been shown to promote pyrimidine synthesis (Wang et al, 2021), future studies are warranted to determine whether the interplay between *NF2* and *KRAS* in regulating pyrimidine metabolism is context-dependent. Interestingly, our study confirms that YAP, a critical downstream oncogenic mediator of NF2-Hippo signaling, facilitates *NF2* deficiency-driven de novo pyrimidine metabolism by transcriptionally upregulating CAD and DHODH. In addition, we observed that MSTO-211H cells with *LATS2* mutations, a core component of the NF2-Hippo pathway, also display strong sensitivity to DHODH inhibition, providing valuable insights into how alterations in various components of the NF2-Hippo signaling may be susceptible

to inhibition of de novo pyrimidine synthesis and could serve as a useful molecular stratification tool for guiding clinical decisions. Notably, while TAZ and YAP are equivalently placed as downstream effectors in the Hippo pathway (Hayashi et al, 2015), our findings reveal that TAZ expression is not affected by *NF2* loss or direct YAP manipulations (Appendix Fig. S7), indicating that YAP and TAZ may have distinct functional roles in *NF2*-driven tumorigenesis.

Glutamine serves as a vital substrate for fueling nucleotide metabolism (Fu et al, 2019). A recent study has revealed that mesothelioma is a glutamine-addicted tumor, suggesting that glutamine/glutamate depletion may be a viable treatment strategy (Adhikary et al, 2023). Our comprehensive transcriptomic and proteomic analysis has revealed that *NF2*-deficient PM cells are highly enriched in alanine/aspartate/glutamate metabolism, which indirectly supports the de novo pathway flux (Mullen and Singh, 2023). Moreover, research has shown that YAP can reprogram glutamine metabolism to increase de novo nucleotide biosynthesis by inducing the transcription of glutamine synthetase (GLUL) in liver cancer (Cox et al, 2016). Intriguingly, YAP has also been found to promote glycolysis in *NF2*-mutant kidney tumors (White et al, 2019). This finding contrasts with a prior study that the absence of *NF2* in mouse embryonic fibroblasts (MEFs) and Schwann cells leads to a decreased glycolytic metabolism (Stepanova et al, 2017). This discrepancy suggests the existence of a context-specific metabolic rewiring system that warrants further investigation. Thus, defining the cooperative mechanism that links key upstream nutrients/substrates, e.g., glucose and amino acids, to de novo pyrimidine synthesis in the context of *NF2*-driven tumorigenesis remains an intriguing area for future research.

The recent approval of combined immunotherapy that targets PD-1 and CTLA-4 has altered the clinical management of mesothelioma (Fennell et al, 2022). However, a considerable number of cases exhibiting nonresponse pose a big challenge. Research indicates that targeting nucleotide metabolism, particularly pyrimidine synthesis, may enhance the antitumor response to

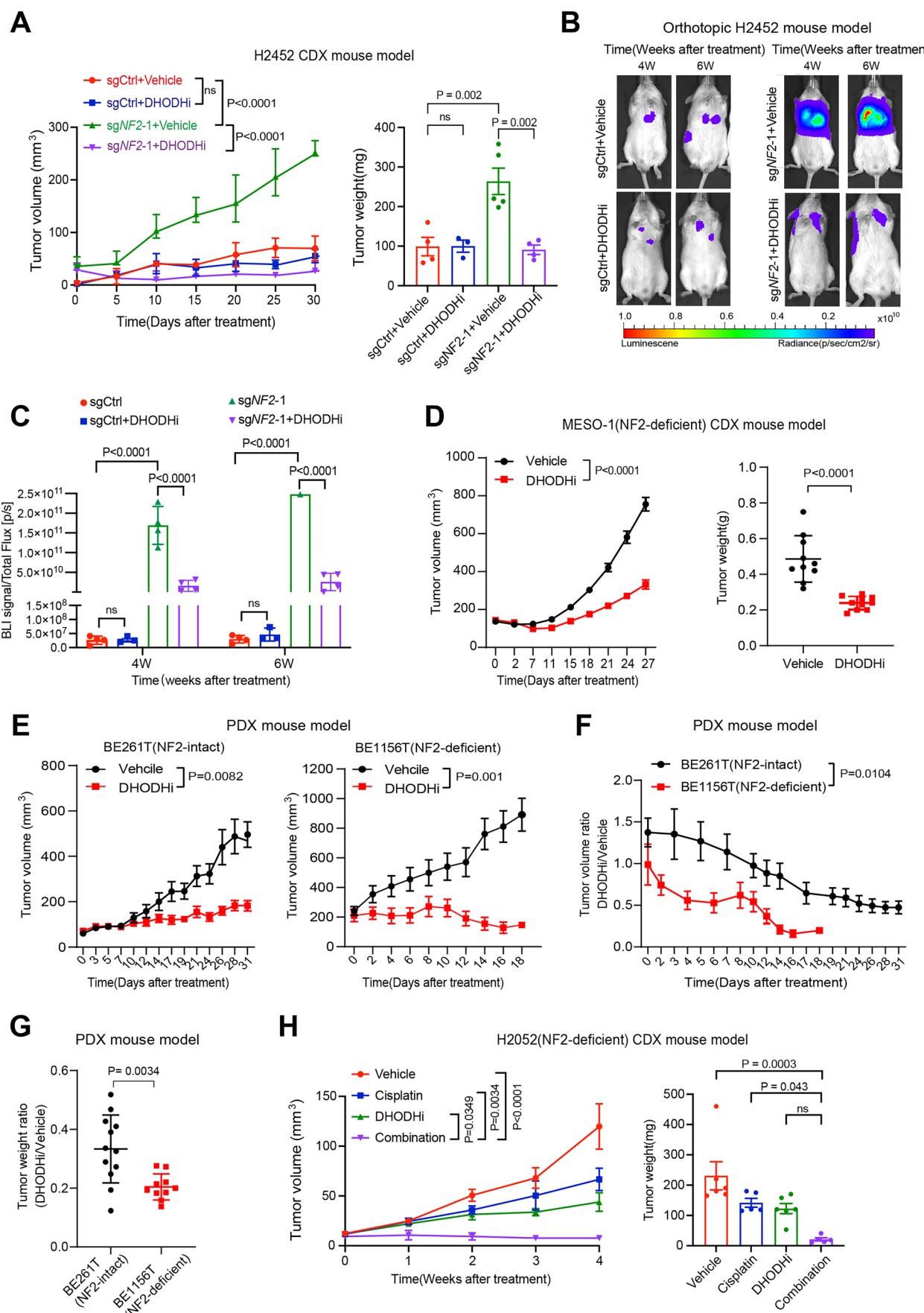

**Figure 6.　Inhibition of de novo pyrimidine synthesis suppresses tumor growth in *NF2*-deficient PM in preclinical mouse models.**

(A) H2452 cells expressing sgCtrl or sg*NF2*-1 were injected subcutaneously as indicated, and the mice were subsequently treated with or without the DHODH inhibitor (Brequinar; 30 mg/kg, three times weekly; $n = 5$ per group). The dynamic development of xenograft tumors was monitored over a period of 30 days (left). The weights of the resected tumors are shown (right). The data are presented as the mean ± SEM. Two-way ANOVA was used to compare tumor growth over time, and one-way ANOVA was used for the comparison of tumor weights. (B) Representative bioluminescence imaging (BLI) showing the tumor growth in the indicated groups within an orthotopic mesothelioma model using NCG (NOD/ShiLtJGpt Prkdc$^{em26Cd52}$Il2rg$^{em26Cd22}$/Gpt) mice. (C) Quantification of the BLI signals (photons/s) in the specified groups of NCG mice within an orthotopic mesothelioma model ($n = 5$ per group). The data are presented as the mean ± SEM. ns not significant. Two-way ANOVA with multiple comparisons was used for statistical analysis. (D) MESO-1 cells were injected subcutaneously as indicated, and the mice were subsequently treated with or without the DHODH inhibitor (Brequinar; 30 mg/kg, three times weekly; $n = 5$ per group). The dynamic development of xenograft tumors was monitored over a period of 27 days (left). The weights of the resected tumors are shown (right). The data are presented as the mean ± SEM. Two-way ANOVA was used to compared tumor growth over time, and a two-tailed unpaired *t* test was used for comparison of tumor weight. (E) The tumor growth of patient-derived xenograft (PDX) BE261T (NF2-intact; $n = 5$ per group) and BE1156T (NF2-deficient; $n = 6$ per group) with and without DHODHi treatment (Brequinar, 30 mg/kg, three times weekly). The data are presented as the mean ± SEM. Two-way ANOVA was used for statistical analysis. (F) The efficacy of the DHODH inhibitor was compared between the NF2-intact (BE261T) and NF2-deficient (BE1156T) PDX models. The tumor volume ratio was calculated by dividing the tumor sizes in the DHODH inhibitor group by those in the vehicle group at various time points. The ratios were plotted on a single graph for comparative analysis. The data are presented as the mean ± SEM. Two-way ANOVA was used for statistical analysis. (G) The efficacy of the DHODH inhibitor was compared by calculating the ratio of tumor weight between the NF2-intact (BE261T) and NF2-deficient (BE1156T) PDX models. The data are presented as the mean ± SD. A two-tailed unpaired *t* test was used for statistical analysis. (H) Mice were injected subcutaneously with *NF2*-mutant (H2052) cells and treated with the indicated drugs for 4 weeks. The dynamic development of xenograft tumors is shown on the left. The weights of resected tumors are presented on the right. The data are presented as the mean ± SEM. Two-way ANOVA was used to compare tumor growth over time, and one-way ANOVA was used for the comparison of tumor weights. Source data are available online for this figure.

immunotherapy (Wu et al, 2022). Paradoxically, studies have shown that altered nucleotide metabolism may enable tumor immune evasion by inducing nucleotide deprivation in immune effector cells (Mullen and Singh, 2023). Therefore, it is imperative to further investigate the metabolic interactions between cancer cells and immune cells, as well as the implications of this crosstalk on antitumor immunity in mesothelioma.

To the best of our knowledge, this research represents the first study employing comprehensive multi-omics analyses to decipher the metabolic diversity of PM. Notably, our study provides a molecular foundation and insights into metabolic vulnerability that can inform personalized therapy for *NF2*-deficient PM patients. The inclusion of PDCLs in our study enhances the clinical relevance of our findings. Moreover, given that several DHODH inhibitors are clinically approved or being evaluated in clinical trials, our results could be readily translated into clinical applications for patients. More broadly, this study highlights the need for further research into *NF2*-driven metabolic dependencies in other cancer types, which could potentially broaden therapeutic strategies beyond mesothelioma.

## Limitations of the study

This study utilized immunocompromised xenograft models to investigate tumor-intrinsic mechanisms; however, we acknowledge the inherent limitations of these models, particularly in relation to inflammation-driven cancers and their responses to immunotherapy. Future research should incorporate syngeneic or humanized mouse models to more accurately evaluate the interactions between tumor-intrinsic alterations and the immune microenvironment. In addition, although our study demonstrated a synergistic effect between cisplatin and DHODH inhibitors in preclinical PM mouse models, the specific molecular mechanisms that underlie this synergy are not yet fully understood. Lastly, while we have provided evidence indicating that glucose limitation does not induce resistance to DHODH inhibitors, other physiological factors, including circulating uridine or hypoxia, may contribute to intrinsic resistance. These potential metabolic resistance mechanisms warrant further exploration.

## Methods

### Reagents and tools table

| Reagent/resource | Reference or source | Identifier or catalog number |
|---|---|---|
| **Experimental models** | | |
| Met-5A | Xu et al, 2020; Xu et al, 2021 | N/A |
| H2452 | | N/A |
| H28 | | N/A |
| MSTO-211H | | N/A |
| H2052 | | N/A |
| MESO-1 | | N/A |
| BE261T | Liang et al, 2015 | N/A |
| BE1071T | This study | N/A |
| BE1193T | This study | N/A |
| BE1143T | This study | N/A |
| PF760 | Hegedüs et al, 2023 | N/A |
| PF434 | Hegedüs et al, 2023 | N/A |
| PF747 | Hegedüs et al, 2023 | N/A |
| PF774 | Hegedüs et al, 2023 | N/A |
| BE1156T | This study | N/A |
| Nude mice | GemPharmatech | Athymic nu/nu |
| NSG mice | In-house breeding facilities in Bern | NOD-SCID IL2Rγnull |
| NCG mice | GemPharmatech | (NOD/ShiLtJGpt Prkdcem26Cd52Il2rgem26Cd22/Gpt) |
| **Recombinant DNA** | | |
| Lenti-CAS9-sgRNA system | GeneChem | U6-sgRNA-SFFV-Cas9-FLAG-P2A-mCherry |
| Lentiviral expression vector | GeneChem | pRRLSIN-cPPT-SFFV-MCS-3FLAG-E2A-EGFP-SV40-puromycin |
| HBLV-luc-BSD | Hanbio Biotechnology | Luciferase reporter vector |
| **Antibodies** | | |
| Rabbit anti-NF2 | Proteintech | 21686-1-AP |
| Rabbit anti-NF2 | CST | 6995 |

| Reagent/resource | Reference or source | Identifier or catalog number |
|---|---|---|
| Rabbit anti-CAD | Proteintech | 16617-1-AP |
| Rabbit anti-DHODH | Proteintech | 14877-1-AP |
| Rabbit anti-YAP | CST | 14074S |
| Rabbit anti-P-YAP(Ser127) | CST | 13008S |
| Rabbit anti-P-Histone H2AX(Ser139) | CST | 9718 T |
| Rabbit anti-Alexa Fluor® 647 anti-H2A.X-Phosphorylated (Ser139) | BioLegend | 613408 |
| Rabbit anti-Ki-67 | Proteintech | 27309-1-AP |
| Rabbit anti-β-actin | CST | 2203T |
| Rabbit anti-GAPDH | CST | 5174T |
| anti-mouse HRP-conjugated antibody | CST | 91196S |
| anti-rabbit HRP-conjugated antibody | CST | 7074S |
| IRDye 800CW Donkey anti-Rabbit | Li-Cor Biosciences | 92632213 |
| IRDye 680RD conjugated anti-mouse IgG antibody | Li-Cor Biosciences | 92668072 |
| ChIP-grade YAP antibody | CST | 14074S |
| **Oligonucleotides and other sequence-based reagents** | | |
| sgCtrl | GeneChem | Table EV7 |
| sgNF2-1 | GeneChem | Table EV7 |
| sgNF2-2 | GeneChem | Table EV7 |
| YAP1 siRNA | GenePharma | Table EV7 |
| CAD siRNA | GenePharma | Table EV7 |
| DHODH siRNA | GenePharma | Table EV7 |
| UPP1 siRNA | GenePharma | Table EV7 |
| KRAS siRNA | GenePharma | Table EV7 |
| YAP1 PCR Primers | This study | Table EV8 |
| CAD PCR Primers | This study | Table EV8 |
| DHODH PCR Primers | This study | Table EV8 |
| GAPDH PCR Primers | This study | Table EV8 |
| UMPS PCR Primers | This study | Table EV8 |
| UPP1 PCR Primers | This study | Table EV8 |
| KRAS PCR Primers | This study | Table EV8 |
| CTGF ChIP Primers | This study | Table EV8 |
| ACTB ChIP Primers | This study | Table EV8 |
| CAD ChIP Primers | This study | Table EV8 |
| DHODH ChIP Primers | This study | Table EV8 |
| Sequences of the promoter fragments for CAD | GeneChem | Table EV8 |

| Reagent/resource | Reference or source | Identifier or catalog number |
|---|---|---|
| Sequences of the promoter fragments for DHODH | GeneChem | Table EV8 |
| **Chemicals, enzymes, and other reagents** | | |
| UCK2 inhibitor | MedChemExpress | HY-148394 |
| $^{15}$N-glutamine | MedChemExpress | HY-N0390S |
| DCPIP (2,6-dichlorophenol-indophenol) | MedChemExpress | HY-D0018 |
| L-Dihydroorotic acid (DHO) | MedChemExpress | HY-W015495 |
| Rotenone | MedChemExpress | HY-B1756 |
| Antimycin A | Shanghai Maokang Biotechnology | MS0070 |
| ADDA 5 hydrochloride | MedChemExpress | HY-U00448 |
| Brequinar (DUP785) | Selleck chem | S6626 |
| YAP-TEAD inhibitor 1(Peptide 17) | Selleck chem | S8164 |
| uridine | Selleck chem | S2029 |
| Cisplatin | Selleck chem | S1166 |
| IMPDH inhibitor | Selleck chem | S2487 |
| Bovine serum albumin (BSA) | MedChemExpress | HY-D0842 |
| d-luciferin | GOLDBIO | LUCK 100 |
| DAPI | Thermo Fisher Scientific | R37606 |
| Cell Counting Kit-8 | Dojindo, Kumamoto | CK18 |
| Lipofectamine 3000 | Invitrogen | L3000001 |
| Odyssey blocking buffer | Li-COR Biosciences | 927-60001 |
| TBS buffer | Sigma | 94158 |
| Clarity Western ECL Substrate | Bio-Rad | 1705060 |
| EZ-Magna ChIP™ A/G Magnetic Beads | Merck Millipore | 17-10086 |
| RPMI-1640 medium | Sigma-Aldrich | 8758 |
| RPMI-1640 medium without L-glutamine | Sigma-Aldrich | R0883 |
| RPMI-1640 medium without glucose | Sigma-Aldrich | R1383 |
| DMEM medium (high glucose) | Sigma-Aldrich | D5796 |
| Fetal bovine serum | Life Technologies | 10270-106 |
| penicillin/streptomycin | Sigma-Aldrich | P0781 |
| Phosphate-buffered saline | Vivacell | C3580-0500 |
| MammoCult™ Human Medium | Stemcell Technology | 05620 |
| B.D. Matrigel Basement Membrane Matrix | Corning | 356231 |

| Reagent/resource | Reference or source | Identifier or catalog number |
|---|---|---|
| **Software** | | |
| Cancer Dependency Map Data Portal | Tsherniak et al, 2017 | N/A |
| MetaboAnalyst | https://www.metaboanalyst.ca/docs/Format.xhtml | N/A |
| UCSC Xena online data portal | https://xenabrowser.net/datapages/ | N/A |
| web-based tool to perform hypergeometric test | https://systems.crump.ucla.edu/hypergeometric/index.php | N/A |
| GraphPad Prism 10 | GraphPad Software, Inc. | N/A |
| Living Image software | PerkinElmer | N/A |
| Case Viewer 2.4 | 3DHISTEC Ltd. | N/A |
| SynergyFinder web tool | Ianevski et al, 2017 | N/A |
| R software (version 3.4.3) | https://www.r-project.org/ | N/A |
| Analyst® TF 1.7.1 software | AB Sciex | N/A |
| **Other** | | |
| Ultralow attachment plates | Corning | 3471 |
| Automated Cell Counter | Shanghai RuiYu Biotech | Countstar IC1000 |
| Multi-Scan FC microplate reader | Thermo Fisher Scientific | N/A |
| Annexin V Apoptosis Detection Kit-FITC | Thermo Fisher Scientific | 88-8005 |
| SDS-PAGE | Bio-Rad | 4561033 |
| SDS-PAGE | GenScript | M00655 |
| PVDF membranes | Bio-Rad | 1620177 |
| Nitrocellulose transfer packs | Bio-Rad | 1704158 |
| Flow cytometry | Beckman | Moflo Astros |
| UHPLC system | Thermo Fisher Scientific | Vanquish |
| Q Exactive HFX mass spectrometer | Thermo Fisher Scientific | Orbitrap MS |
| QIAquick PCR purification kit | Qiagen | 28104 |
| Bond Polymer Refine Detection Kit | Leica Biosystems | DS9800 |
| Dual-Luciferase Reporter Assay System | Promega | E1910 |
| BioTek Synergy H1 Microplate Reader | Agilent Technologies | N/A |
| RNeasy Mini Kit | Qiagen | 74106 |
| NanoDrop MicroUV–visible spectrophotometer | Thermo Fisher Scientific | N/A |
| High-capacity cDNA reverse transcription kit | Applied Biosystems | 4368814 |
| QuantStudio 7 Real-Time PCR Detection System | Thermo Fisher Scientific | N/A |

| Reagent/resource | Reference or source | Identifier or catalog number |
|---|---|---|
| QIAwave RNA Mini Kit | QIAGEN | 74536 |
| Bioanalyzer 2100 system | Agilent Technologies | N/A |
| NEBNext® Ultra™ RNA Library Prep Kit for Illumina® | NEB | N/A |
| BCA protein assay kit | Beyotime | P0012 |
| IVIS Spectrum | Caliper Life Sciences | N/A |
| Automated BOND RX system | Leica Biosystems | N/A |

## Data retrieval and preprocessing

Transcriptomic data from patients with PM, featuring high-coverage gene profiles (over 20,000 genes), were sourced from The Cancer Genome Atlas (TCGA) MESO dataset, the Gene Expression Omnibus (GEO; GSE12345, GSE163720, GSE42977, and GSE51024) and the European Bioinformatics Institute (EMBL-EBI; E-MTAB-1109) (Data ref: Crispi et al, 2009; De Rienzo et al, 2021; De Rienzo et al, 2013; Roe et al, 2012; Suraokar et al, 2014). To mitigate batch effects, we utilized the "ComBat" function from the "sva" package, employing an empirical Bayesian framework to adjust and eliminate batch-related biases prior to merging the transcriptomic datasets (Leek et al, 2012). Further details about these public datasets can be found in Table EV4.

## Identification of differentially expressed genes (DEGs) and functional enrichment analysis

The empirical Bayesian methodology incorporated in the "limma" R package was employed to identify DEGs from the gene expression profiles across various groups. The threshold for defining DEGs was set at an adjusted $P$ value $< 0.05$. Common DEGs were determined using the "VennDiagram" R package. In addition, Gene Ontology (GO) analysis, Kyoto Encyclopedia of Genes and Genomes (KEGG) pathway analysis, and gene set enrichment analysis (GSEA) were performed utilizing the "ClusterProfiler" R package.

## Metabolic pathway-based clustering

To deconstruct the metabolic diversity present among mesothelioma patients, a total of 1638 metabolism-related genes were retrieved from the KEGG database. These genes were organized into eighty-four metabolic pathways, which were classified into ten principal groups in accordance with KEGG classifications (Dataset EV13). The Gene Set Variation Analysis (GSVA) algorithm, a widely recognized and advanced method for gene set projection that reflects the activity levels of biological processes(Hanzelmann et al, 2013), was employed to compute the gene set enrichment score for each metabolic pathway based on transcriptomic data. This score serves to indicate whether the genes within a specific set are synergistically upregulated or downregulated in the samples, thereby providing insights into the activation or suppression of pathways.

A correlation-based distance metric was utilized to estimate the global variations between pairs of metabolic gene expression profiles (Gong et al, 2021). Then, consensus clustering was conducted using the "Consensus Cluster Plus" package in R, with 1000 iterations and 80% resampling, to ascertain the optimal number of stable metabolism-related subpopulations among the patients in the dataset based on the expression of metabolism-related genes. Ultimately, the patients were categorized into three distinct metabolic-related subpopulations: Cluster 1, comprising 47% of all samples, was characterized by a notable upregulation of nucleotide metabolism, particularly in the pyrimidine synthesis pathway; Cluster 2, representing 25% of all samples, was identified as a mixed type exhibiting combined dysregulation across all major categories; and Cluster 3, accounting for 28% of all tumors, was distinguished by a relative upregulation of linoleic acid metabolism, arachidonic acid metabolism, and ether lipid metabolism.

Furthermore, we extended the metabolic pathway-based subtypes to mesothelioma cell lines utilizing the R package "pamr." The expression data from the tumor samples ($n = 328$) were integrated with the mesothelioma cell line data from the Cancer Cell Line Encyclopedia (CCLE; $n = 19$) and subsequently normalized within each dataset. Prior to the implementation of the "pamr.train" and "pamr.predict" functions, we computed the enrichment score for each metabolic pathway in each sample using the GSVA method. Subsequently, the cell lines were categorized into the metabolic pathway-based subtypes of the tumor samples based on the outcomes of the "pamr" predictive analysis. Transcriptomic (Expression Public 23Q4) and metabolomic (Metabolomics) datasets for the PM cell lines used in this research were obtained from the Cancer Dependency Map Data Portal (Tsherniak et al, 2017).

## Comparison of genetic alteration frequencies among metabolic subtypes

Genetic alteration data, encompassing copy number variations and mutations, were sourced from the TCGA-MESO series via the UCSC Xena online data portal (https://xenabrowser.net/datapages/). Initially, we computed the proportion of samples exhibiting gene mutations for each metabolic subtype and conducted Fisher's exact test to evaluate mutation frequency differences among the subtypes utilizing the TCGA-MESO dataset (Dataset EV8). In addition, the hypergeometric test was employed to assess the significance of NF2, with the $P$ value calculated using a web-based tool (https://systems.crump.ucla.edu/hypergeometric/index.php).

## Cell culture and reagents

Normal mesothelial cells (Met-5A) and commercial PM cell lines (H28, H2052, MSTO-211H, H2452, and MESO-1) were previously described (Xu et al, 2020; Xu et al, 2021). Primary patient-derived cell lines (PDCLs) such as BE261T, BE1071T, BE1193T, and BE1143T were established from surgically resected PM tumors following the protocol outlined in Switzerland (Liang et al, 2015; Xu et al, 2018); In addition, cell lines PF760, PF434, PF747, and PF774 were derived from pleural effusions of mesothelioma patients, as previously described in Germany (Hegedüs et al, 2023). Both commercial cell lines and BE primary populations were cultured in RPMI-1640 medium (Catalog #8758; Sigma-Aldrich),

supplemented with 10% fetal bovine serum (Catalog #10270-106; Life Technologies) and 1% penicillin/streptomycin solution (Catalog #P0781; Sigma-Aldrich). PF cells were maintained in DMEM medium (high glucose; Catalog #D5796; Sigma-Aldrich), also supplemented with 10% fetal bovine serum and 1% penicillin/streptomycin solution. Commercial cell lines and primary cells sourced from Germany underwent authentication through short tandem repeat (STR) profiling, and all cells were routinely tested to ensure they were free of mycoplasma. Details regarding the cell lines utilized in this study can be found in Table EV5.

RPMI-1640 medium without L-glutamine (Catalog #R0883) or glucose (Catalog #R1383) was obtained from Sigma-Aldrich (St. Louis, MO, USA). The small-molecule inhibitor Brequinar (DUP785; Catalog #S6626), YAP-TEAD inhibitor 1 (Peptide 17; Catalog #S8164), uridine (Catalog #S2029), cisplatin (Catalog #S1166) and IMPDH inhibitor (mycophenolic; Catalog #S2487) were purchased from Selleck chem (Houston, TX, USA). A UCK2 inhibitor (Catalog #HY-148394) and $^{15}$N-glutamine (Catalog #HY-N0390S) were obtained from MedChemExpress (South Brunswick, NJ, USA).

## 3D sphere formation assay

A three-dimensional (3D) culture of tumor spheres was performed as previously described (Xu et al, 2018). In brief, single cells were suspended in MammoCult™ Human Medium (Catalog #05620; Stemcell Technology, Canada) and plated at a density of 1000 cells/ml in ultralow attachment plates (Catalog #3471; Corning Incorporated, Corning, NY, USA). Following a 24-h incubation period, cells were treated with pharmacological agents for a duration of 96 h, after which they were cultured without any medications for a period of 10–14 days, with the cell culture medium being refreshed every 3 days. Each experimental group was established in triplicate, and all experiments were conducted three times.

## Cell viability and clonogenic assay

Cells were cultured in 96-well plates at a density of 2500 cells per well and subsequently treated with pharmacological agents 24 h post-seeding for a duration of 96 h, unless specified otherwise. Cell viability was evaluated utilizing a Cell Counting Kit-8 (CCK-8, Cat. #CK18; Dojindo, Kumamoto, Japan) or the Alkaline Phosphatase (APH) assay, in accordance with the manufacturer's guidelines. The impact of the treatments on cell proliferation was normalized against the untreated control group. Each data point was derived from triplicate measurements, and each experiment was conducted three times. Unless indicated otherwise, a representative result is presented. The best-fit curve was generated using GraphPad Prism [log(inhibitor) vs. response-variable slope (four parameters)]. In addition, to evaluate cell proliferation, PM cells transfected with either control or CAD/DHODH siRNAs were seeded in triplicate in 24-well plates at a density of $1 \times 10^4$ cells per well. After the indicated time points, the cells were trypsinized and counted using an automated cell counter (Countstar IC1000, Shanghai RuiYu Biotech Co. Ltd.).

For the clonogenic assay, exponentially growing cells were seeded in six-well plates at a density of 1000 cells per well and treated with drugs for 3 days. The cells were then cultured without

further treatment for a period of 10–14 days, depending on the growth rate. Colonies were subsequently stained with crystal violet. Quantification was performed using crystal violet staining in conjunction with 10% acetic acid, and the absorbance was measured at 590 nm using a Multi-Scan FC microplate reader (Thermo Fisher Scientific, USA).

## Flow cytometry-based assays

PM cells were treated as specified in the figure legends. Following treatment, the cells present in the supernatant were collected, washed with phosphate-buffered saline (PBS; Catalog #C3580-0500; Vivacell), pooled, and resuspended in 200 μl of binding buffer. The cells were then stained using the Annexin V Apoptosis Detection Kit-FITC (Catalog #88-8005; Thermo Fisher Scientific) in accordance with the manufacturer's guidelines and subsequently analyzed using flow cytometry. To evaluate the level of apoptosis in cell lines stably transfected with mCherry, propidium iodide (PI) was substituted with 4',6-diamidino-2-phenylindole (DAPI).

For cell cycle and DNA damage analysis, the cells were trypsinized, collected by centrifugation, washed, and resuspended in 0.1% bovine serum albumin (BSA) (Catalog #HY-D0842; MedChemExpress) in PBS. They were then fixed in 75% (v/v) ethanol at a density of $1 \times 10^6$ cells/ml overnight. The cells were also stained with anti-γH2AX(Ser139) antibody (Alexa Fluor 647, Catalog 613408, BioLegend) overnight on a rotator at 4 °C. Following the addition of 1000 units of RNase A, the cells were stained with 0.5 μg/ml DAPI (Catalog #R37606; Thermo Fisher Scientific) for an hour and subsequently analyzed using the BD FACS SORP LSR II flow cytometer.

## Immunoblotting

Cell lysates were prepared, and Western blot analysis was conducted in accordance with the methodology outlined by (Xu et al, 2018). In brief, equal amounts of protein lysates (10-25 μg per lane) were separated using SDS-PAGE (Catalog #4561033; Bio-Rad Laboratories, Catalog #M00655, GenScript) and subsequently transferred onto 0.2-μm PVDF membranes (Catalog #1620177; Bio-Rad) or nitrocellulose transfer packs (Catalog #1704158; Bio-Rad). The membranes were blocked for one hour at room temperature with 5% skim milk or for 2 h with Odyssey blocking buffer (Catalog #927-60001; Li-COR Biosciences), followed by overnight incubation at 4 °C with the appropriate primary antibodies (Table EV6) in the blocking buffer. After three washes with TBS buffer (Catalog #94158; Sigma) containing 0.2% Tween 20, the membranes were incubated at room temperature for an hour with either an anti-mouse HRP-conjugated antibody (Catalog #91196S; Cell Signaling Technology), an anti-rabbit HRP-conjugated antibody (Catalog #7074S; Cell Signaling Technology), IRDye 800CW Donkey anti-Rabbit (Catalog #92632213; Li-Cor Biosciences), or IRDye 680RD conjugated anti-mouse IgG antibody (Catalog #92668072; Li-Cor Biosciences). Membranes were then subjected to chemiluminescence using Clarity Western ECL Substrate (Catalog #1705060; Bio-Rad), or those stained with fluorophore-conjugated secondary antibodies were scanned using the Licor Odyssey Infrared Imaging instrument. Representative blots from a minimum of three independent experiments are presented. ImageJ/Image Studio software was employed for the quantification of western blot data.

## Lentivirus transfection

The Lenti-CAS9-sgRNA system (U6-sgRNA-SFFV-Cas9-FLAG-P2A-mCherry), specifically designed to knockout NF2, was developed by GeneChem (Shanghai, China). The targeted sequences are listed in Table EV7. Stable genetic knockout cell lines were established following the methodology (Xu et al, 2020). In brief, cells were cultured overnight to reach 50–70% confluence, then infected with the appropriate lentiviral vector at a multiplicity of infection (MOI) of ~10, along with 8 μg/ml polybrene. After a 3-day infection period, mCherry-positive cells were isolated via flow cytometry (Moflo Astros; Beckman) for subsequent analyses. A relevant scrambled sgRNA served as a negative control. The knockout efficiency was assessed using western blot.

To generate sgRNA-resistant NF2 cDNA, a lentiviral expression vector (pRRLSIN-cPPT-SFFV-MCS-3FLAG-E2A-EGFP-SV40-puromycin) was constructed by GeneChem (Shanghai, China). This vector was specifically engineered to be resistant to sgRNA sequences targeting NF2, as indicated in Table EV7. In summary, cell lines with a stable NF2 knockout were transduced with sgRNA-resistant NF2 cDNA at an MOI of approximately 10, along with 8 μg/ml polybrene. Three days post-infection, mCherry/GFP-positive cells were sorted using flow cytometry (Moflo Astros; Beckman) for further experimentation. The restoration of NF2 was evaluated through western blot.

## Small interfering RNA (siRNA) knockdown

Knockdown of YAP1, CAD, DHODH, KRAS, and UPP1 was achieved via specific duplex siRNAs obtained from GenePharma (Shanghai, China), with a control siRNA duplex employed as the negative control. The specific sequences can be found in Table EV7. In brief, cells that reached 70–80% confluence were transfected with siRNAs (10 nM each) and Lipofectamine 3000 (Catalog #L3000001; Invitrogen) following the manufacturer's guidelines.

## Identification of YAP/TEAD-binding sites

The ChIP-Seq data for YAP1/TEAD1 in the mesothelioma cell lines H2052 and MSTO-211H were retrieved from the Cistrome Data Browser (CistromeDB) (Zheng et al, 2019). In addition, the H3K27ac ChIP-Seq dataset for MSTO-211H was downloaded to identify active enhancers and promoters. The genome sites for the TEAD motif were predicted via the HOMER tool. The data in bigwig and bed formats were visualized using the Integrative Genomics Viewer (IGV). Regions that exhibited both motifs and ChIP-Seq peaks were deemed credible binding sites. In instances where no binding sites were identified within the promoters, we conducted further investigations to identify potential enhancer regions within the same topologically associated domain (TAD) for YAP/TEAD binding. The coordinates for chromatin loops and TADs were sourced from a 3D genome browser (Wang et al, 2018). Subsequently, the regions interacting with the target promoters were analyzed to ascertain the presence of any motifs or peaks. Given the challenges associated with loop calling from high-throughput chromosome conformation capture (Hi-C) data, we also examined active enhancer sites within the 500 kb flanking regions surrounding the target promoters within the same TAD for motifs and peaks.

## Chromatin immunoprecipitation (ChIP) assay

The cells were subjected to crosslinking in a 1% paraformaldehyde solution for a duration of 10 min, followed by the addition of glycine to achieve a final concentration of 125 mM for an additional 5 min. Subsequently, the cells were washed twice with cold PBS and collected in PBS. They were then sonicated using an ultrasonic homogenizer for 10 min at 30% power on ice to fragment the DNA to an average size ranging from 200 to 1000 base pairs (bp). A volume of fifty microliters from each sonicated sample was retained for the assessment of DNA concentration and fragment size. The cell lysates were incubated overnight at 4 °C with 20 μl of EZ-Magna ChIP™ A/G Magnetic Beads (Catalog #17-10086; Merck Millipore, Darmstadt, Germany) and 5 μg of ChIP-grade YAP antibody (Catalog #14074S; Cell Signaling Technology, Beverly, MA, United States). Following this, the beads were collected, washed, and treated with proteinase K for 2 h at 62 °C, as well as with RNase for 1 h at 37 °C. DNA purification was performed using a QIAquick PCR purification kit (Catalog #28104; Qiagen, Germany). The resulting DNA fragments were analyzed through quantitative PCR with reverse transcription, utilizing the primer sequences provided in Table EV8. The samples were normalized against the input DNA.

## Dual-luciferase functional assay

For the luciferase assay, cells were initially seeded in a 24-well culture plate and incubated overnight. The firefly plasmids were co-transfected with the Renilla construct at a concentration of 0.04 μg. After 48 h, the transfected cells were washed twice with PBS, and total protein extraction was performed using Dual-Luciferase Reporter Assay System (Catalog # E1910; Promega) via passive lysis buffer at a volume of 100 μl per well. Firefly and Renilla luciferase activities were measured in the lysate using a BioTek Synergy H1 Microplate Reader (Agilent Technologies, CA, USA). Firefly luciferase values were normalized to Renilla luciferase values. The specific sequences of the promoter fragments utilized in these experiments can be found in Table EV8.

## RNA extraction and quantitative real-time PCR (qRT-PCR)

Total RNA was isolated and purified using the RNeasy Mini Kit (Catalog #74106; Qiagen, Germany). RNA integrity was assessed via a NanoDrop MicroUV–Visible spectrophotometer (Thermo Fisher Scientific). Complementary DNA (cDNA) was synthesized with a high-capacity cDNA reverse transcription kit (Catalog # 4368814; Applied Biosystems, Foster City, CA, USA) according to the manufacturer's instructions. Quantitative reverse transcription polymerase chain reaction was performed with SYBR Green Supermix on a QuantStudio 7 Real-Time PCR Detection System (Thermo Fisher Scientific). Gene-specific primers were obtained from Generay (Shanghai, China) and are detailed in Table EV8. The expression levels of each target gene were normalized to that of *GAPDH* and compared among the different groups via the $^{\Delta\Delta}$CT method. The baseline and threshold for Ct calculation were automatically set with QuantStudio Real-Time PCR Software v1.3.

## Illumina RNA sequencing

Total RNA was extracted from sgCtrl or *NF2*-KO PM cells via the QIAwave RNA Mini Kit (Catalog #74536; QIAGEN) according to the manufacturer's protocol. The integrity of the RNA was evaluated using the RNA Nano 6000 Assay Kit in conjunction with the Bioanalyzer 2100 system (Agilent Technologies, CA, USA). An input of 1 μg of RNA per sample was employed for the RNA sample preparations. In alignment with the manufacturer's guidelines, sequencing libraries were constructed using the NEBNext® Ultra™ RNA Library Prep Kit for Illumina® (NEB, USA), with index codes incorporated to facilitate the assignment of sequences to each sample. Following cluster generation, the library preparations were sequenced on an Illumina NovaSeq platform, yielding 150 bp paired end reads. Differential expression analysis was conducted using the DESeq2 R package (version 1.20.0). The resulting *P* values were adjusted using Benjamini–Hochberg's method to control the false discovery rate (FDR). A *P* value less than 0.05 and log2-fold change threshold of 0.5 (which equates to a fold change greater than 1.4) were set to determine significant changes in gene expression (Dotsenko et al, 2024; Schinke et al, 2022).

## Proteome-wide label-free quantification

SDT buffer (4% SDS, 100 mM Tris-HCl, pH 7.6) was introduced to the cell lysates. Following centrifugation at $14,000 \times g$ for 15 min, the protein concentration in the supernatant was determined using a BCA protein assay kit (Catalog #P0012; Beyotime). Subsequently, 20 μg of protein from each sample was combined with 6X loading buffer and subjected to boiling for 5 min. The proteins were then separated using 12% SDS-PAGE, and the protein bands were visualized through Coomassie Blue R-250 staining. After conducting filter-aided sample preparation (FASP digestion), the initial data obtained from liquid chromatography-tandem mass spectrometry (LC-MS/MS) were analyzed using MaxQuant software version 1.6.17.0 and were cross-referenced against the database. The threshold for the global false discovery rate (FDR) for peptide and protein identification was set at 0.01. Protein abundance was assessed based on the normalized spectral protein intensity (LFQ intensity). A *P* value less than 0.05 and fold change greater than 1.3 were used as a valuable threshold for proteomics analysis (Lanfredi et al, 2021; Zhang et al, 2018).

## Liquid chromatography-mass spectrometry (LC-MS/MS) metabolomics analysis

The cells ($10^7$ cells/sample) were washed with ice-cold saline and then snap-frozen in liquid nitrogen. Extract solution (acetonitrile:methanol:water = 2:2:1) containing isotopically labeled internal standard mixture was added to the sample. After 30 s vortex, the samples underwent three freeze–thaw cycles using liquid nitrogen. Then the samples were sonicated for 10 min in ice-water bath. Then the samples were incubated at −40 °C for 1 h and centrifuged at 12,000 rpm for 15 min at 4 °C. The resulting supernatant was transferred to a fresh glass vial for LC/MS analysis. LC-MS/MS analyses were performed using an UHPLC system (Vanquish, Thermo Fisher Scientific) with an UPLC BEH Amide column (2.1 mm × 100 mm, 1.7 μm) coupled to Q Exactive HFX mass spectrometer (Orbitrap MS, Thermo). The mobile phase consisted

of 25 mmol/L ammonium acetate and 25 ammonia hydroxide in water pH = 9.75 (A) and acetonitrile (B). The autosampler temperature was 4 °C, and the injection volume was 2 μL. The QE HFX mass spectrometer was used for its ability to acquire MS/MS spectra on information-dependent acquisition (IDA) mode in the control of the acquisition software (Xcalibur, Thermo). The ESI source conditions were set as following: sheath gas flow rate as 30 Arb, Aux gas flow rate as 25 Arb, capillary temperature 350 °C, full MS resolution as 120,000, MS/MS resolution as 7500, collision energy as 10/30/60 in NCE mode, spray Voltage as 3.6 kV (positive) or −3.2 kV (negative), respectively. The raw data were converted to the mzXML format using ProteoWizard and processed with an in-house program for peak detection, extraction, alignment, and integration. In-house MS2 database was applied in metabolite annotation. The cut-off for annotation was set at 0.3. For the targeted metabolomic analysis, a specific set of multiple reaction monitoring (MRM) transitions corresponding to the designated metabolites was monitored simultaneously for further optimization.

The metabolites that showed significant differences between *NF2*-knockout (sg*NF2*-1) and scrambled control (sgCtrl), with a *P* value less than 0.05 and a variable importance in projection (VIP) score greater than 1, were submitted to MetaboAnalyst, an online platform (https://www.metaboanalyst.ca/docs/Format.xhtml) designed for metabolite set enrichment analysis (MSEA). This analysis aimed to identify the biologically relevant metabolic pathways affected by the genetic knockout of *NF2*.

## $^{15}$N-glutamine isotype tracing

$^{15}$N-glutamine tracing experiments were performed on H2452 sgCtrl and sg*NF2* cell populations. In brief, the cells were cultured in glutamine-free media supplemented with 2 mM $^{15}$N-glutamine for a duration of 18 h. Subsequently, metabolites were extracted using ice-cold methanol containing phenylhydrazine. The samples were incubated at 1500 rpm for 30 min at 4 °C, followed by a one-hour incubation at −20 °C (Zimmermann et al, 2014). The samples were then centrifuged for 15 min at 12,000 rpm at 4 °C, and the supernatant was collected and dried using a SpeedVac in H2O mode. The dried extract was reconstituted in a solution of 5% acetonitrile in water prior to LC-MS analysis on an Agilent 1290 II UPLC coupled to Sciex 5600+ quadrupole-TOF MS. Hydrophilic interaction liquid chromatography (HILIC) was conducted using a Waters ACQUITY BEH amide column. The column oven temperature was maintained at 25 °C, and the autosampler was set at 10 °C. The injection volume was 5 μL. The flow rate was 0.4 mL min−1. Mobile phase A (pH 9.0) consisted of acetonitrile and water (1:9, v/v) with 20 mM ammonium acetate, and mobile phase B (pH 9.0) consisted of acetonitrile and water (9:1, v/v) with 20 mM ammonium acetate. The following linear gradient was used: 0–2.0 min with 95% B, 2.0–12.0 min with 95–50% B, 12.0–13.0 min with 50–50% B, 13.0–13.1 min with 50–95% B, 13.1–16.0 min with 95–95% B. The mass spectrometry detection parameters were set as follows: ESI source voltage in negative ion mode at −4.5 kV; vaporizer temperature at 500 °C; drying gas (N2) pressure at 50 psi; nebulizer gas (N2) pressure at 50 psi; curtain gas (N2) pressure at 35 psi; and a scan range of *m/z* 70–850 during HILIC analysis (Tian et al, 2022). Data acquisition and processing were performed via Analyst® TF 1.7.1 software (AB Sciex, Concord, ON, Canada).

## DHODH enzymatic activity assay

As previously described (Mao et al, 2021; Shi et al, 2022), the activity of DHODH was assessed spectrophotometrically at a temperature of 37 °C by monitoring the reduction in absorbance at 596 nm of reduced DCPIP (2,6-dichlorophenol-indophenol; Catalog #HY-D0018; MedChemExpress), which serves as an artificial electron acceptor, over a duration of 60 min. Briefly, the reaction was initiated by the addition of 20 mM DHO (Catalog #HY-W015495; MedChemExpress) to 1 ml of a standard reaction buffer, which was further supplemented with 50 μM DCPIP, 2 μg of rotenone (a complex I inhibitor; Catalog #HY-B1756; MedChemExpress), 2 μg of antimycin A (a complex III inhibitor; Catalog #MS0070; MedChemExpress), 10 μM ADA5 (a complex IV inhibitor; Catalog #HY-U00448; MedChemExpress), and 0.1 mg of whole-cell lysate. The change in absorbance per minute for the control samples was subsequently subtracted from the change observed in the experimental samples.

## In vivo xenograft mouse study

### Subcutaneous mouse model

Male athymic nu/nu mice, aged 6–8 weeks, as well as NSG (NOD-SCID IL2Rγnull) mice, were obtained from GemPharmatech (Jiangsu, China) or from in-house breeding facilities in Bern, respectively. All the mice were housed in standardized conditions (12 h inverted dark: light phase, 22 ± 2 °C, with food and water ad libitum). In the subcutaneous tumor models conducted in Nanjing, athymic nu/nu mice were administered an injection of $5 \times 10^6$ tumor cells, which were suspended in PBS mixed in a 1:1 ratio with B.D. Matrigel Basement Membrane Matrix (Cat. #356231; Corning). In the subcutaneous tumor models performed in Bern, MESO-1 cells were suspended in PBS mixed with B.D. Matrigel Basement Membrane Matrix (Cat. #356231; Corning) at a 1:1 ratio, followed by the injection of $4 \times 10^6$ cells into NSG mice. Upon tumors becoming palpable, the mice were randomly allocated to one of the following treatment groups: (1) the control group or (2) the DHODH inhibitor Brequinar (30 mg/kg, administered intraperitoneally three times weekly) for a duration of 4 weeks. For the combination treatment assessment, the mice were randomly assigned to one of the following groups: (1) the control group; (2) the cisplatin group (3.75 mg/kg, administered intraperitoneally once weekly); (3) the DHODH inhibitor Brequinar (15 mg/kg, administered intraperitoneally three times weekly for 4 weeks); and (4) a combination of the doses. Tumor size was measured using calipers, and tumor volume was calculated using the formula: $(a \times b^2)/2$, where "a" and "b" denote the length and width of the tumor, respectively. Throughout the experiments, the mice were euthanized at a predetermined endpoint, adhering to the Institutional Animal Care and Use Committee (IACUC) protocol. Mice were immediately euthanized if any tumor exceeded a volume of 1000 mm³, reached a diameter of 1.5 cm, or resulted in a 20% loss of body weight. At the end of the study, the mice were euthanized using $CO_2$, and the tumors were excised and processed for subsequent experiments. All protocols pertaining to the mouse experiments received approval from Nanjing Medical University (IACUC-1706007-3-2022) and the Canton of Bern (BE85/2023) in Switzerland. The sample size was based on previous mouse studies in conjunction with the ARRIVE recommendations on refinement and reduction of animal use in research.

## Orthotopic pleural mesothelioma mouse model

Male NCG (NOD/ShiLtJGpt Prkdcem26Cd52Il2rgem26Cd22/Gpt) mice, aged between 6 to 8 weeks, were obtained from GemPharmatech (Jiangsu, China) for the establishment of an orthotopic mouse model of pleural mesothelioma. Mice were housed in standardized conditions (12 h inverted dark: light phase, $22 \pm 2\,°C$, with food and water ad libitum). In summary, the mice were anesthetized using isoflurane and oxygen inhalation, followed by the intrapleural injection of $2 \times 10^6$ tumor cells transduced with the luciferase reporter vector HBLV-luc-BSD (Hanbio Biotechnology, China) in 100 µl of PBS. Tumor progression was monitored through bioluminescence imaging (BLI), and the survival of the mice was assessed daily. NCG mice with tumors were anesthetized with isoflurane and received an intraperitoneal injection of 150 mg/kg of D-luciferin (Catalog #LUCK 100; GOLDBIO). BLI measurements were conducted using an IVIS Spectrum (Caliper Life Sciences, USA) 10–15 min post-injection. During the imaging process, the mice were kept under isoflurane anesthesia administered via a nose cone. The BLI signal was quantified as total flux (photons per second), reflecting the average ventral and dorsal flux. The image radiance values were normalized using Living Image software (PerkinElmer, USA). All experimental protocols involving mice were approved by Nanjing Medical University (IACUC-1706007-3-2022). The sample size was based on previous mouse studies in conjunction with the ARRIVE recommendations on refinement and reduction of animal use in research.

## Patient-derived xenograft (PDX) mouse model

Pleural mesothelioma PDX models BE1156T and BE261T were established from mesothelioma patients (Table EV5) by implanting fresh tissues (5mmX5mm) subcutaneously into NSG (NOD-SCIDIL2Rgnull) mice. Expanded PDX tumors were harvested and enzymatically digested using collagenase I/II to obtain single-cell suspensions. These cells were resuspended in PBS and mixed in a 1:1 ratio with B.D. Matrigel Basement Membrane Matrix (Catalog #356231; Corning). When tumors reached 100–200 mm³, NSG mice were randomly assigned to one of two treatment groups: (1) Control group; (2) DHODH inhibitor Brequinar (30 mg/kg, i.p., three times per week). Tumor size was measured using callipers, and tumor volume was calculated using the formula: $(a \times b^2)/2$, where "a" and "b" represent the tumor length and width, respectively. Mice were euthanized when the largest tumor reached 1000 mm³. All animal experiments were conducted in compliance with protocols approved by the Ethics Committee of the Canton of Bern, Switzerland (license number BE85/2023). The sample size was based on previous mice studies in conjunction with the ARRIVE recommendations on refinement and reduction of animal use in research. All patients provided informed consent for the use of their clinical data and specimens in this study, and the experiments conformed to the principles set out in the WMA Declaration of Helsinki and the Department of Health and Human Services Belmont Report.

## Patient samples

For the construction of tissue microarrays (TMA), we conducted a retrospective collection of formalin-fixed, paraffin-embedded (FFPE) samples from 17 patients who underwent surgical intervention for mesothelioma at The First Affiliated Hospital of Nanjing Medical University between 2011 and 2020. Among these patients, eight had both tumor specimens and adjacent normal tissues available. Two pathologists evaluated the samples and selected two 2.0-mm diameter dots from each specimen for inclusion in the TMA. In addition, the Bern cohort consists of 98 PM patients who underwent consecutive surgical resections at Bern University Hospital between 2003 and 2017. In this cohort, two pathologists evaluated the samples and selected six to eight 2.0 mm diameter dots from each specimen for TMA inclusive. Detailed patient information for the two cohorts is available in Table EV9 and was previously described (Xu et al, 2024). The processing of patient tissue for TMAs were approved by the Ethical Committee of The First Affiliated Hospital of Nanjing Medical University (No.2022-SRFA-328) and the Ethics Commission of the Canton of Bern (KEK Bern 2016-01497). All patients provided informed consent for the use of their clinical data and specimens in this study, and the experiments conformed to the principles set out in the WMA Declaration of Helsinki and the Department of Health and Human Services Belmont Report.

## Immunohistochemistry and quantitative analysis

Utilizing a standardized protocol, surgically excised tumor tissues were subjected to formalin fixation and paraffin embedding, followed by sectioning and staining with hematoxylin-eosin (H&E). Subsequent immunohistochemical staining was executed using an automated BOND RX system (Leica Biosystems, Newcastle, UK). FFPE tissue blocks were sectioned to a thickness of 4 µm, deparaffinized, and rehydrated. Whole-slide images, represented by each punch dot, were captured and processed using Case Viewer 2.4 (3DHISTEC Ltd.). The immunohistochemical analysis of the Bern cohort was performed in a similar manner. Specifically, tissue microarrays were sectioned at 4 µm, deparaffinized, rehydrated, and subsequently stained with the relevant antibodies utilizing the automated BOND RX system (Leica Biosystems). Visualization was achieved through the Bond Polymer Refine Detection Kit (Catalog #DS9800; Leica Biosystems), following the manufacturer's instructions. Images were obtained using PANNORAMIC whole slide scanners. The antibodies used in IHC analysis are detailed in Table EV6. The H-SCORE was calculated using the formula H-SCORE = $\Sigma(pi \times i)$ = (percentage of weak intensity×1) + (percentage of moderate intensity×2) + (percentage of strong intensity×3).

## Statistical analysis

Data points for animal experiments were excluded before the end of the experiment if animals had to be culled to comply with the animal license or for husbandry-related reasons. For other experiments, data acquisition was blinded, and no data were excluded from analyses. The normality of the data was assessed using the Kolmogorov–Smirnov test. The data represent biological replicates ($n$) and are presented as the mean values ± SD. or mean values ± SEM, as indicated in the figure legends. Comparisons of mean values were conducted with unpaired two-tailed Student's $t$ tests, one-way or two-way ANOVA, as appropriate. For ANOVA, adjustments for multiple comparisons were made using Dunnett,

**The paper explained**

**Problem**

Pleural mesothelioma (PM) is a highly lethal cancer characterized by significant tumor heterogeneity. Despite substantial efforts that have been made to understand this heterogeneity at various biological levels, the metabolic underpinnings remain poorly defined. In particular, the molecular mechanisms linking genetic alterations to metabolic reprogramming are not well understood—representing a major gap in advancing precision oncology approaches for PM.

**Results**

Using an integrative multi-omics approach, we identify for the first time that the loss-of-function mutations in *NF2* define a unique subtype of PM marked by upregulated de novo pyrimidine synthesis. *NF2*-deficient PM cells exhibit a strong dependency on this pathway for continued growth both in vitro and in vivo. Mechanistically, we demonstrate that the NF2-YAP signaling axis drives this metabolic shift by transcriptionally activating key enzymes involved in de novo pyrimidine biosynthesis, including CAD and DHODH.

**Impact**

Our findings reveal a previously unrecognized metabolic vulnerability in *NF2*-deficient PM, positioning de novo pyrimidine synthesis as a promising synthetic lethal target. This work provides critical insight into the intersection of cancer genetics and metabolism, opening avenues for the development of targeted therapeutic strategies.

Tukey, or Sidak's corrections when necessary. Non-parametric statistical tests were employed for variables that did not conform to a normal distribution, as indicated in the figure legends. A *P* value < 0.05 was considered indicative of statistical significance. The sample size was estimated in this study. In the survival analysis, tumor samples from all datasets were categorized into two groups based on the optimal separation cut-off value, allowing for the generation of Kaplan–Meier survival curves using the "survminer" and "survival" R packages. The predictive values of categorical variables were evaluated using the log-rank test. The evaluation of drug combination synergy was conducted using the highest single-agent (HSA) model through the SynergyFinder web tool(Ianevski et al, 2017). All statistical analyses were performed using GraphPad Prism 10 (GraphPad Software, Inc.) or R software (version 3.4.3), unless noted differently in the Method Details.

## Data availability

The datasets generated in this research are accessible in the Gene Expression Omnibus (GEO) under the accession number GSE295651 for RNA sequencing, in ProteomeXchange under the accession number PXD063342 for mass spectrometry proteomics (https://www.ebi.ac.uk/pride/;), and in MetaboLights under the accession number MTBLS12461 for metabolomics (https://www.ebi.ac.uk/metabolights/MTBLS12461).

The source data of this paper are collected in the following database record: biostudies:S-SCDT-10_1038-S44321-025-00278-4.

## Peer review information

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

## Acknowledgements

The authors express their gratitude to the Core Facility of the First Affiliated Hospital of Nanjing Medical University for their assistance in the analysis of experimental samples. In addition, the findings presented in this publication are partially based on data obtained from the TCGA Research Network, GEO databases, and the European Bioinformatics Institute. The authors acknowledge the contributions of the affiliated institutions. The authors' sole responsibility was to interpret and report these data. This work was supported by grants from National Natural Science Foundation of China (#82404066 to D Xu, #82172889 to Y-Q Shu and #82173012 to R-W Peng); Natural Science Foundation of Jiangsu Province (#BK20241139 to D Xu); Swiss National Science Foundation (#320030-231251 to R-W Peng).

## Author contributions

**Duo Xu**: Conceptualization; Resources; Data curation; Software; Formal analysis; Funding acquisition; Validation; Investigation; Visualization; Methodology; Writing—original draft; Project administration; Writing—review and editing. **Yanyun Gao**: Conceptualization; Data curation; Formal analysis; Validation; Investigation; Visualization; Methodology; Writing—original draft; Writing—review and editing. **Shengchen Liu**: Conceptualization; Resources; Data curation; Software; Formal analysis; Validation; Investigation; Visualization; Methodology; Writing—original draft; Writing—review and editing. **Shiyuan Yin**: Validation; Investigation; Visualization; Methodology; Writing—review and editing. **Tong Hu**: Resources; Data curation; Software; Validation; Investigation; Methodology; Writing—review and editing. **Haibin Deng**: Resources; Formal analysis; Investigation; Methodology; Writing—review and editing. **Tuo Zhang**: Data curation; Validation; Visualization; Methodology; Writing—review and editing. **Balazs Hegedüs**: Resources; Methodology; Writing—review and editing. **Thomas M Marti**: Resources; Methodology; Writing—review and editing. **Patrick Dorn**: Resources; Methodology; Writing—review and editing. **Shun-Qing Liang**: Resources; Software; Methodology; Writing—review and editing. **Ralph A Schmid**: Resources; Writing—review and editing. **Ren-Wang Peng**: Conceptualization; Resources; Supervision; Funding acquisition; Methodology; Writing—original draft; Project administration; Writing—review and editing. **Yongqian Shu**: Conceptualization; Resources; Supervision; Funding acquisition; Writing—original draft; Project administration; Writing—review and editing.

Source data underlying figure panels in this paper may have individual authorship assigned. Where available, figure panel/source data authorship is listed in the following database record: biostudies:S-SCDT-10_1038-S44321-025-00278-4.

## Disclosure and competing interests statement

The authors declare no competing interests.

# Expanded View Figures

**Figure EV1.  *NF2* deficiency promotes tumorigenesis in PM.**

(**A**) Immunoblots of the indicated proteins in H2452 (top) and H28 (bottom) PM cell lines transfected with a scrambled control (sgCtrl) or *NF2*-targeting sgRNAs (sg*NF2*-1, sg*NF2*-2). Representative images from three independent experiments are shown. Quantitative data can be found in Appendix Fig. S5A,B. (**B**) Cell viability assay of H2452 (left) and H28 (right) cells transfected with a scrambled control (sgCtrl) or *NF2*-targeting sgRNAs (sg*NF2*-1, sg*NF2*-2) at the indicated time points. The data are presented as the mean ± SD ($n = 3$). Two-way ANOVA with multiple comparisons was used for statistical analysis. (**C**, **D**) Clonogenic assay of H2452 (left) and H28 (right) cells transfected with a scrambled control (sgCtrl) or *NF2*-targeting sgRNAs (sg*NF2*-1, sg*NF2*-2). After being continually cultured for 14 days, viable cells were stained with crystal violet dye (**C**). (**D**) Quantification of the clonogenic assay results. The data are presented as the mean ± SD ($n = 3$). One-way ANOVA with multiple comparisons was used for statistical analysis. (**E**) Representative bioluminescence imaging (BLI) image showing tumor growth in athymic nu/nu mice across the indicated groups (H2452 sgCtrl and sg*NF2*-1) within an orthotopic mesothelioma model. (**F**) Quantification of BLI signals (photons/s) for the indicated groups (H2452 sgCtrl and sg*NF2*-1) in an orthotopic mesothelioma model using athymic nu/nu mice ($n = 9$ per group). The data are presented as the mean ± SEM. ns: not significant. Two-way ANOVA with multiple comparisons was used for statistical analysis. (**G**) Kaplan–Meier curves showing survival rates in the specified groups (H2452 sgCtrl and sg*NF2*-1) within an orthotopic mesothelioma model using athymic nu/nu mice ($n = 9$ per group). The *P* value was calculated by the log-rank test. (**H**) Kaplan–Meier curves showing overall survival (OS) in the TCGA cohort of mesothelioma patients ($n = 87$), stratified by the genetic status of *NF2*. The *P* value was calculated using the log-rank test in R. (**I**) Kaplan–Meier curves showing OS based on the protein levels of NF2 in the independent Bern cohort of mesothelioma patients ($n = 82$). Patients were stratified into high (in black) and low (in red) groups according to the optimal cut-off value of individual expression across the cohort using the surv_cutpoint function in the R "maxstat" package. The *P* value was calculated using the log-rank test in R.

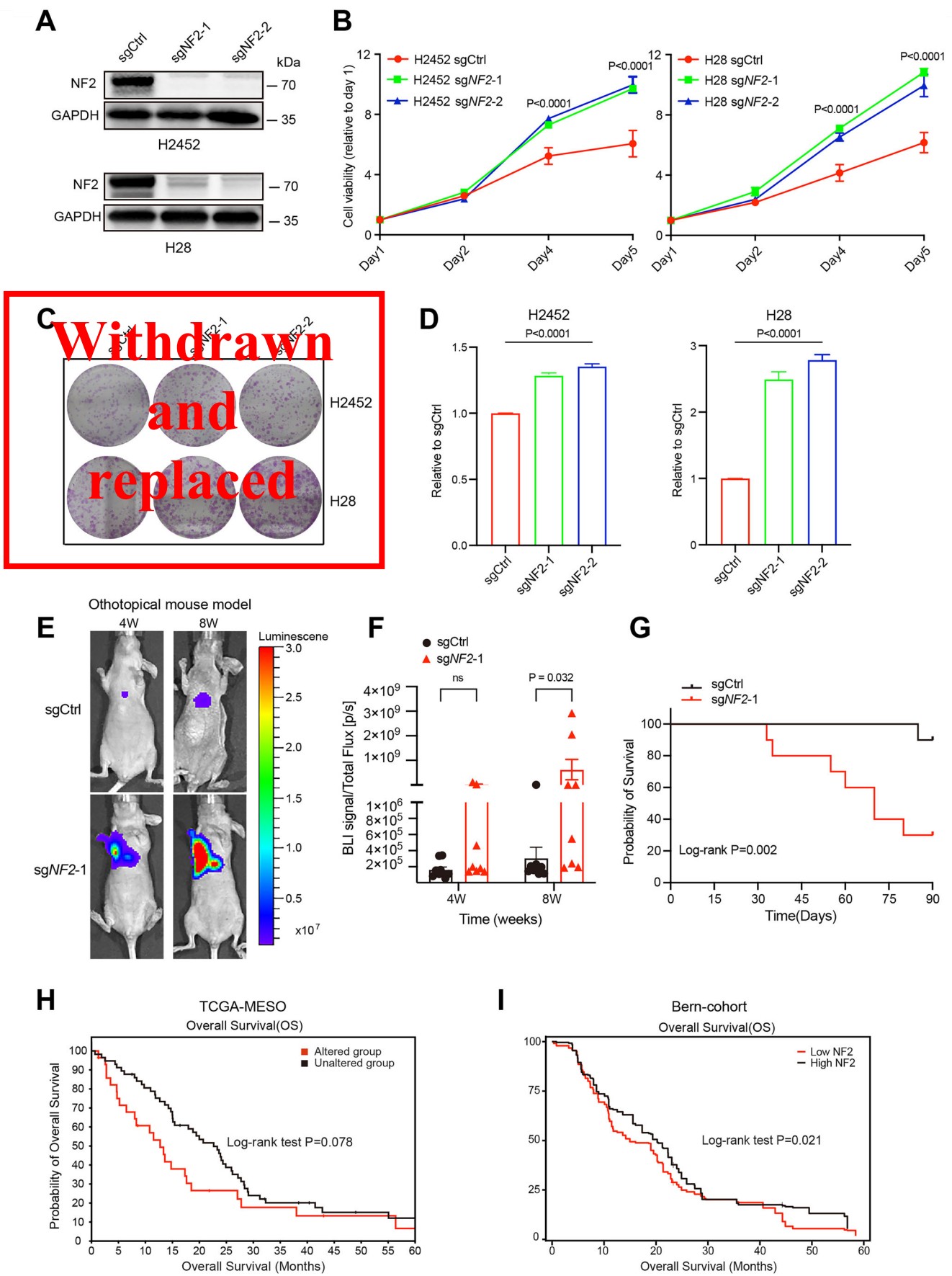

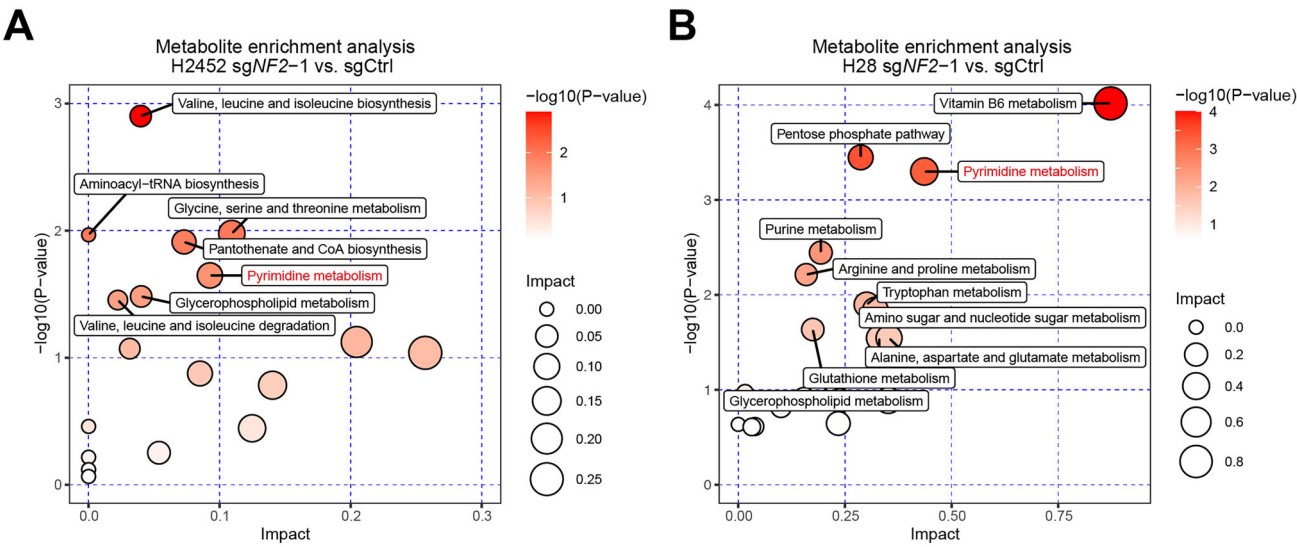

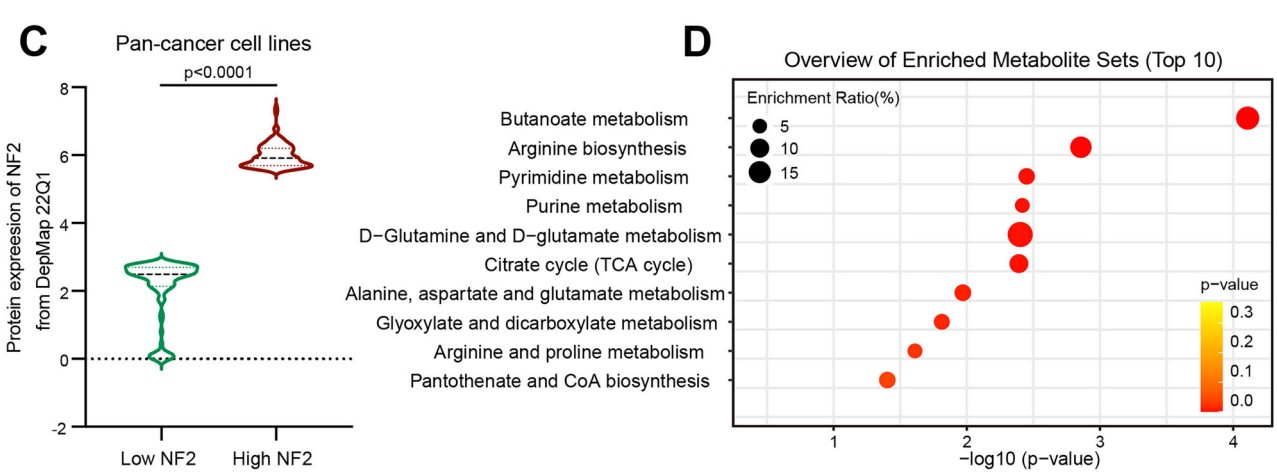

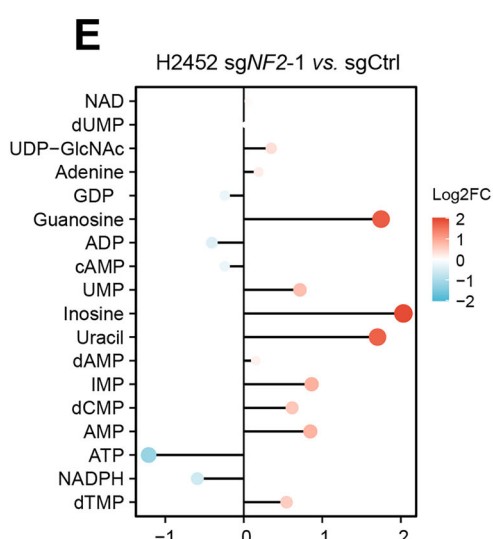

◀ **Figure EV2.  *NF2* deficiency drives metabolic alterations in PM.**

(A, B) Metabolite pathway enrichment analysis of differentially expressed metabolites with *P* value < 0.05 and a variable important in projection (VIP) > 1 in *NF2*-knockout (sg*NF2*-1) compared to control (sgCtrl) groups. The significantly altered metabolic pathways with *P* value < 0.05 are labeled in H2452 (**A**) and H28 (**B**). Pathway impact refers to the importance of altered metabolites in the respective metabolic pathway, as calculated by Metabo-Analyst. The analysis included six biological replicates for H2452, and three biological replicates for H28. Related to Fig. 1F. (**C**) The differential expression of NF2 between low-NF2 and high-NF2 groups across pan-cancer cell lines. The data were stratified according to the lowest and highest 100 NF2 protein expression levels in the ranking list, derived from the proteomic dataset available at the Cancer Dependency Map Data Portal. Protein expression is defined by the Z-score, which is calculated with reference to each protein as measured across the entire panel of cell lines. A two-tailed unpaired *t* test was used for comparisons. (**D**) Bubble diagram showing the top 10 enriched metabolite sets in the NF2-low group compared to the NF2-high group. (**E**) Lollipop chart depicting the differentially expressed metabolites (*P* value < 0.05) in the H2452 sg*NF2*-1 group compared to the sgCtrl group.

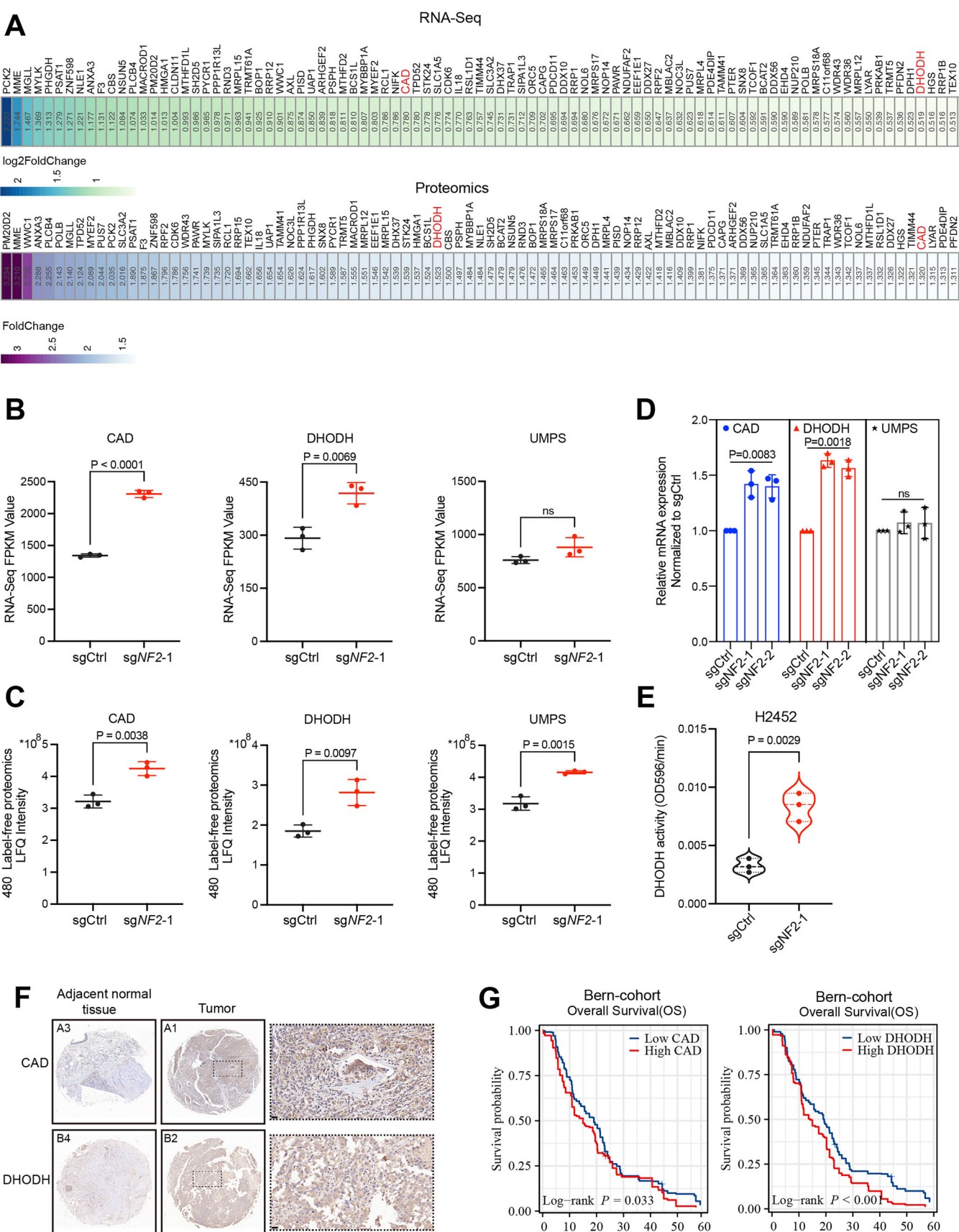

◀

**Figure EV3. Loss of *NF2* elevates enzyme expression within the de novo pyrimidine synthesis pathway.**

(A) Heatmap showing the 92 common candidates shared by both the differentially upregulated genes ($P$ value < 0.05 and fold change>1.4) and proteins ($P$ value < 0.05 and fold change>1.3) in the H2452 sg*NF2*-1 group compared to the sgCtrl group. Related to Fig. 2B. (B, C) The differences in FPKM values from RNA sequencing (B) and LFQ intensities from 480 label-free proteomes (C) of the indicated markers between the H2452 sg*NF2*-1 and sgCtrl groups. The data are presented as the mean ± SD ($n = 3$). A two-tailed unpaired $t$ test was used for statistical analysis. (D) The mRNA expression of the indicated markers in H2452 PM cell lines transfected with a scrambled control (sgCtrl) or *NF2*-targeting sgRNAs (sg*NF2*-1, sg*NF2*-2). The data are presented as the mean ± SD ($n = 3$). One-way ANOVA with multiple comparisons was used for statistical analysis. (E) The enzymatic activity of DHODH in H2452 PM cell lines transfected with a scrambled control (sgCtrl) or *NF2*-targeting sgRNA (sg*NF2*-1). The data are presented as the mean ± SD ($n = 3$). A two-tailed unpaired $t$ test was used for statistical analysis. (F) Representative images of immunohistochemistry (IHC) staining showing the expression of CAD and DHODH in paired normal tissue and tumor samples from the internal mesothelioma tissue microarray. Scale bar: 20 μm. (G) Kaplan–Meier curves showing OS based on the protein levels of CAD (left) and DHODH (right) in the independent Bern cohort of mesothelioma patients ($n = 82$). Patients were stratified into high (in black) and low (in red) according to the optimal cut-off value of individual expression across the cohort using the surv_cutpoint function in the R "maxstat" package. The $P$ value was calculated using the log-rank test in R.

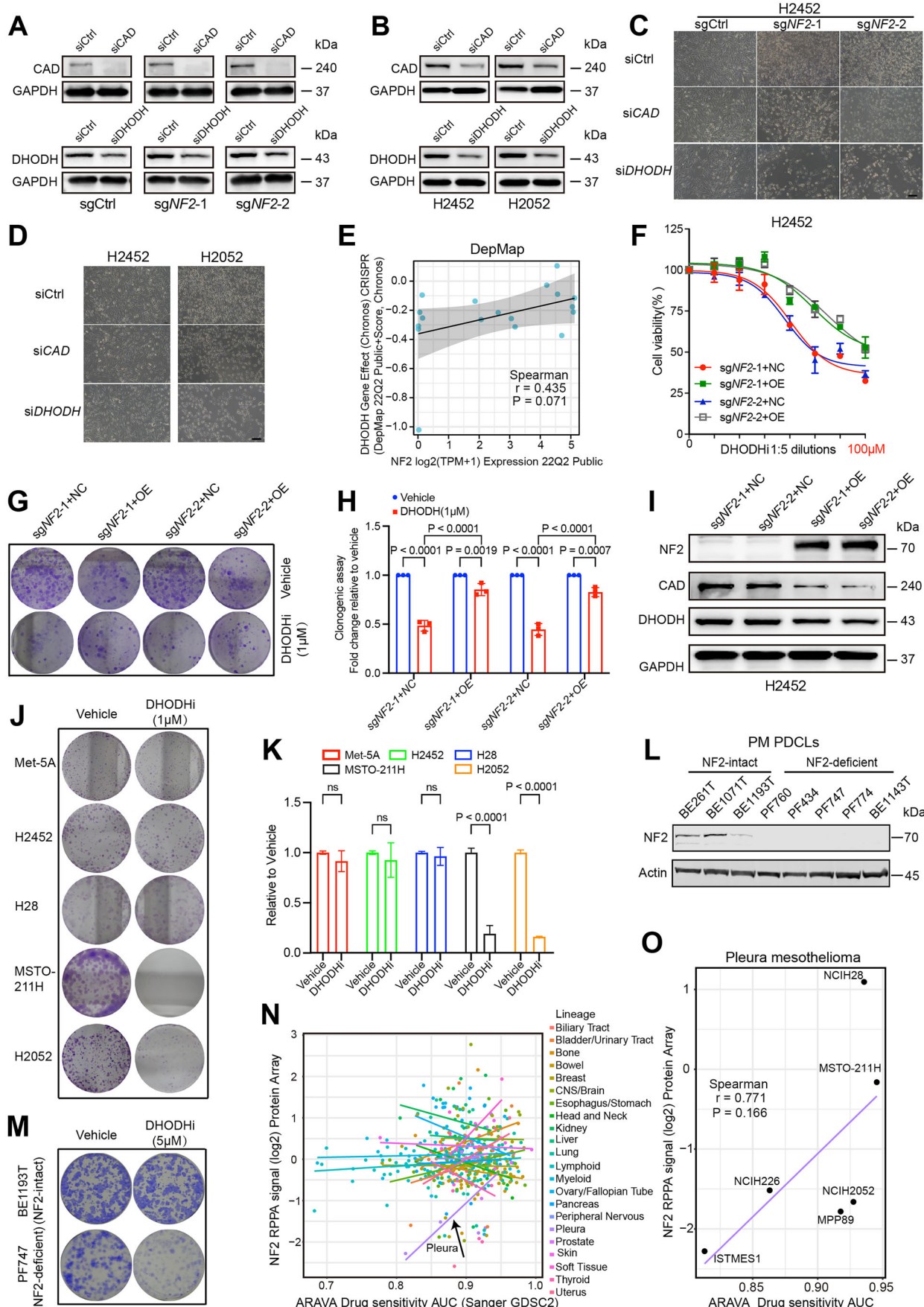

◀ **Figure EV4. Targeting de novo pyrimidine synthesis selectively induces cell death in *NF2*-deficient PM cells.**

(A) Immunoblots of the indicated proteins in H2452 sgCtrl, sg*NF2*-1, and sg*NF2*-2 PM cells after transfection with small interfering RNA (siRNA) targeting *CAD* or *DHODH* for 72 h. Representative images from three independent experiments are shown. Quantitative data can be found in Appendix Fig. S5H. (B) Immunoblots of the indicated proteins in H2452 (*NF2* wild-type) and H2052 (*NF2* mutant) cells after transfection with small interfering RNA (siRNA) targeting *CAD* or *DHODH* for 72 h. Representative images from three independent experiments are shown. Quantitative data can be found in Appendix Fig. S5I. (C) Representative images of the indicated cell populations after transfection with small interfering RNA (siRNA) targeting *CAD* or *DHODH* for 72 h. Scale bar: 100 µM. (D) Representative images of H2452 (*NF2* wild-type) and H2052 (*NF2* mutant) cells after transfection with small interfering RNA (siRNA) targeting *CAD* or *DHODH* for 72 h. Scale bar:100 µM. (E) Scatter plot showing the correlation between the transcript levels of NF2 and the genetic dependency score of DHODH in PM cell lines ($n = 18$). The data were obtained from the Dependency Map (22Q2). (F) H2452 sg*NF2*-1 and sg*NF2*-2 PM cells, transfected with a negative control (NC) or sgRNA-resistant NF2 cDNA (OE), were treated with increasing doses of the DHODH inhibitor Brequinar. Cell viability was measured 120 h post-treatment. The data are presented as the mean ± SD. Representative results from three independent experiments are shown. (G, H) Clonogenic assay of H2452 sg*NF2*-1 and sg*NF2*-2 PM cells transfected with a negative control (NC) or sgRNA-resistant NF2 cDNA (OE) and treated with vehicle or the DHODH inhibitor (1 µM) for 120 h. After a 14-day culture period, viable cells were stained with crystal violet. Representative images (G) and quantification (H) from three independent experiments are shown. The data are presented as the mean ± SD. Two-way ANOVA with multiple comparisons was used for statistical analysis. (I) Immunoblots of the indicated proteins in H2452 sg*NF2*-1 and sg*NF2*-2 PM cells transfected with an negative control (NC) or sgRNA-resistant NF2 cDNA (OE). Representative images from three independent experiments are shown. Quantitative data can be found in Appendix Fig. S5J. (J, K), Clonogenic assay of normal mesothelial cells (Met-5A) and PM cell lines (*NF2* wild-type: H2452, H28, MSTO-211H; *NF2* mutant: H2052) treated with vehicle or the indicated drugs for 96 h. After a 14-day culture period, the viable cells were stained with crystal violet. Representative images (J) and quantification (K) from three independent experiments are shown. The data are presented as the mean ± SD. Two-way ANOVA with multiple comparisons was used for statistical analysis. (L) Immunoblots of the indicated proteins in primary-derived cell lines (PDCLs) used for the cell viability assays shown in Appendix Fig. S5L. (M) Clonogenic assay of representative patient derived cell lines (PDCLs) BE1193T(NF2-intact) and PF747(NF2-deficient) treated with 1 µM DHODH inhibitor Brequinar for 96 h. After 14 days of cell growth, viable cells were fixed and stained with crystal violet dye. Representative images from three independent experiments are shown. (N, O) Scatter plot showing the correlation between NF2 protein levels and drug sensitivity to leflunomide (ARAVA; Sanger GDSC2) across a pan-cancer cohort of cell lines ($n = 1578$). Data from PM cell lines are presented separately in (O; $n = 6$). The data were generated from the DepMap project.

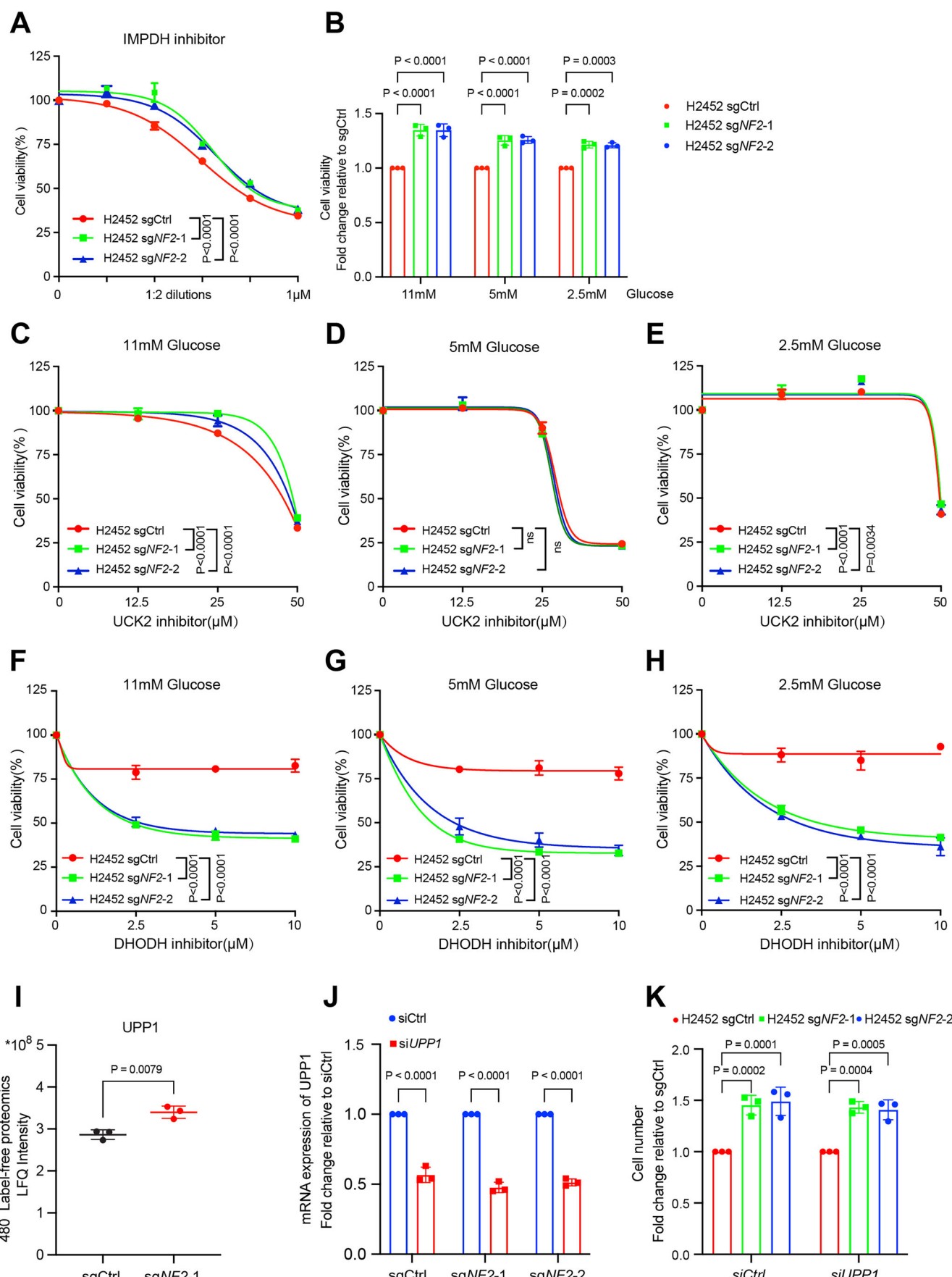

◀ **Figure EV5. *NF2*-deficient PM cells are independent of purine biosynthesis and pyrimidine salvage pathway.**

(A) Cell viability of H2452 *NF2* wild-type (sgCtrl) and *NF2* knockout (sg*NF2*-1, sg*NF2*-2) cells treated with various doses of the IMPDH inhibitor for 96 h. Representative data from three independent experiments are shown. The data are presented as the mean ± SD. Two-way ANOVA with multiple comparisons was used for statistical analysis. (B) Cell viability of H2452 *NF2* wild-type (sgCtrl) and *NF2* knockout (sg*NF2*-1, sg*NF2*-2) cells cultured with the indicated concentrations of glucose. The data are presented as the mean ± SD ($n = 3$). Two-way ANOVA with multiple comparisons was used for statistical analysis. (C–E) Cell viability of H2452 *NF2* wild-type (sgCtrl) and *NF2* knockout (sg*NF2*-1, sg*NF2*-2) cells treated with the indicated doses of the UCK2 inhibitor for 96 h at 10 mM (C), 5 mM (D) and 2.5 mM (E) glucose. Representative data from three independent experiments are shown. The data are presented as the mean ± SD. Two-way ANOVA with multiple comparisons was used for statistical analysis. (F–H) Cell viability of H2452 *NF2* wild-type (sgCtrl) and *NF2* knockout (sg*NF2*-1, sg*NF2*-2) cells treated with the indicated doses of the DHODH inhibitor for 96 h at 10 mM (F), 5 mM (G) and 2.5 mM (H) glucose. Representative data from three independent experiments were shown. The data are presented as the mean ± SD. Two-way ANOVA with multiple comparisons was used for statistical analysis. (I) The differences in LFQ intensities from 480 label-free proteomes of the indicated markers between the H2452 sg*NF2*-1 and sgCtrl groups. The data are presented as the mean ± SD ($n = 3$). A two-tailed unpaired *t* test was used for statistical analysis. (J, K) The mRNA expression of UPP1 (J) and cell viability (K) of the indicated cell populations transfected with small interfering RNA (siRNA) targeting the negative control or *UPP1* for 72 h. The data are presented as the mean ± SD ($n = 3$). Two-way ANOVA with multiple comparisons was used for statistical analysis.

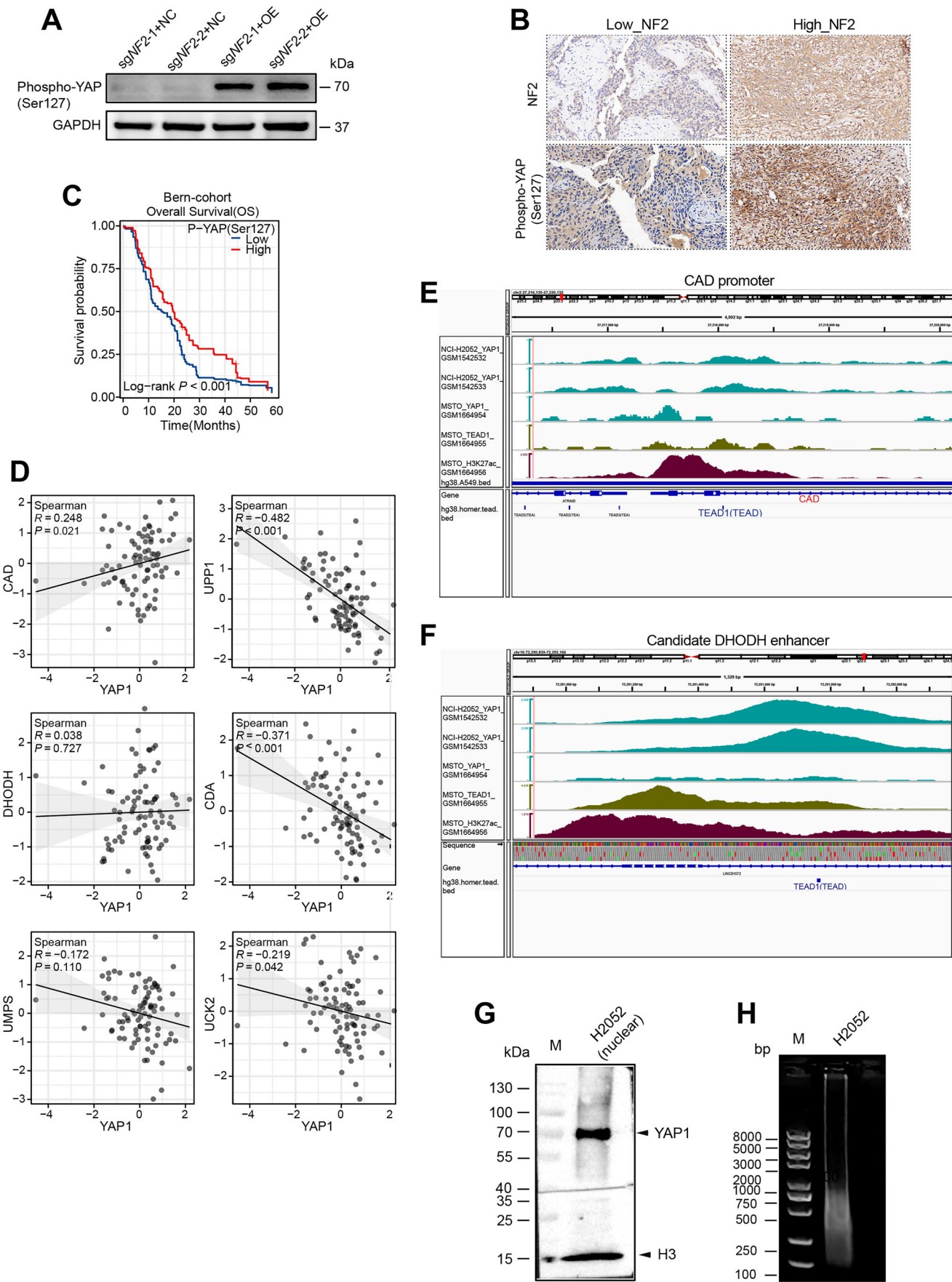

**Figure EV6.  YAP enhances de novo pyrimidine synthesis through the transcriptional regulation of key enzymes.**

(A) Immunoblots of the indicated proteins in H2452 sg*NF2*-1 and sg*NF2*-2 PM cells transfected with a negative control (NC) or sgRNA-resistant NF2 cDNA (OE). Representative images from three independent experiments are shown. Quantitative data can be found in Appendix Fig. S5K. (B) Representative IHC images of the indicated proteins in the low-NF2 and high-NF2 subgroups, stratified by the top and bottom quartiles of the H-score in our internal mesothelioma tissue microarray. The original overall magnification: ×200 (scale bar: 20 μm). (C) Kaplan–Meier curves showing OS based on the protein levels of P-YAP(Ser127) in the independent Bern cohort of mesothelioma patients ($n = 82$). Patients were stratified into high (in black) and low (in red) according to the optimal cut-off value of individual expression across the cohort using the surv_cutpoint function in the R "maxstat" package. The $P$ value was calculated using the log-rank test in R. (D) Spearman correlation coefficients between the mRNA expression of YAP1 and other markers related to pyrimidine metabolism in the TCGA-MESO cohort ($n = 87$). (E, F) Representative YAP1/TEAD1-binding peak regions identified in the ChIP-seq datasets. The ChIP-seq data for YAP1/TEAD1 in the mesothelioma cell lines NCI-H2052 and MSTO-211H were retrieved from the Cistrome Data Browser (CistromeDB). The Y-axis is presented without normalization across the ChIP-seq datasets. (G) Immunoblots of nuclear YAP1 in the H2052 PM cell line. (H) Chromatin break agarose gel electrophoresis results for the H2052 PM cell line. Source data are available online for this figure.

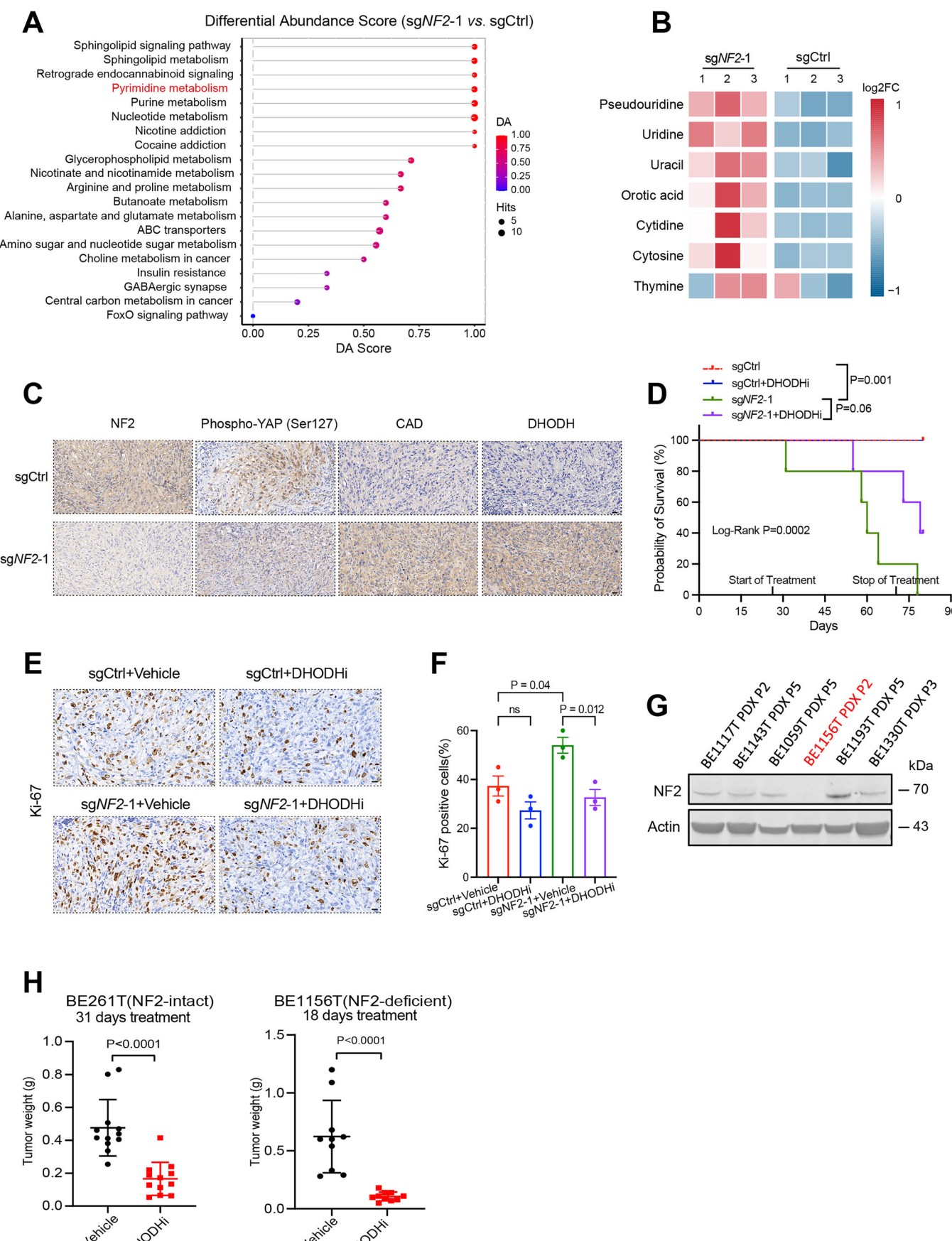

◄ **Figure EV7.  DHODH inhibition specifically reduces tumor growth in *NF2*-deficient PM mouse models.**

(A) Differential abundance score (DA) reflecting the overall metabolic alterations between H2452 *sgNF2*-1 and sgCtrl-resected orthotopic tumors. A score of 1 indicates an upregulated expression pattern of metabolites identified in this pathway, and a score of -1 indicates a downregulated expression pattern. The length of the line segment represents the absolute value of the DA score, and the size of the dot at the end of the line segment represents the number of metabolites in the pathway. The depth of the color of the line segment and dot is proportional to the DA score: the darker the red color, the more inclined the overall expression of the pathway is to be upregulated; conversely the darker the blue color, the more inclined the overall expression is to be downregulated. (B) Heatmap showing the differentially expressed metabolites involved in pyrimidine metabolism ($P$ value < 0.05 and VIP > 1) between H2452 *sgNF2*-1- and sgCtrl-resected orthotopic tumors ($n = 3$). (C) Representative IHC staining of the indicated proteins in resected H2452 sgCtrl or sg*NF2*-1 tumors. The original overall magnification is ×200 (scale bar: 20 μm). (D) Kaplan–Meier curves showing survival rates in the specified groups within an orthotopic mesothelioma model using NCG (NOD/ShiLtJGpt Prkdc[em26Cd52]Il2rg[em26Cd22]/Gpt) mice ($n = 5$ per group). The $P$ value was calculated using the log-rank test. (E, F) IHC analysis of Ki-67 in resected H2452 tumors expressing sgCtrl or sg*NF2*-1 treated with or without the DHODH inhibitor (Brequinar; 30 mg/kg). The original overall magnification is ×200 (scale bar: 20 μm). Representative images (E) were captured and processed using Case Viewer software. The quantification of Ki-67-positive cells is shown in (F). The data are presented as the mean ± SD. Two-way ANOVA with multiple comparisons was used for statistical analysis. (G) Immunoblots of the indicated proteins in patient-derived xenograft (PDX) mesothelioma tumors. Cell lysates were obtained from snap-frozen PDX tumor tissues. (H) The tumor weights of PDX BE261T (left; $n = 6$ per group) and BE1156T (right; $n = 5$ per group) with and without the DHODH inhibitor treatment (Brequinar, 30 mg/kg, three times weekly). A two-tailed unpaired $t$ test was used for statistical analysis.

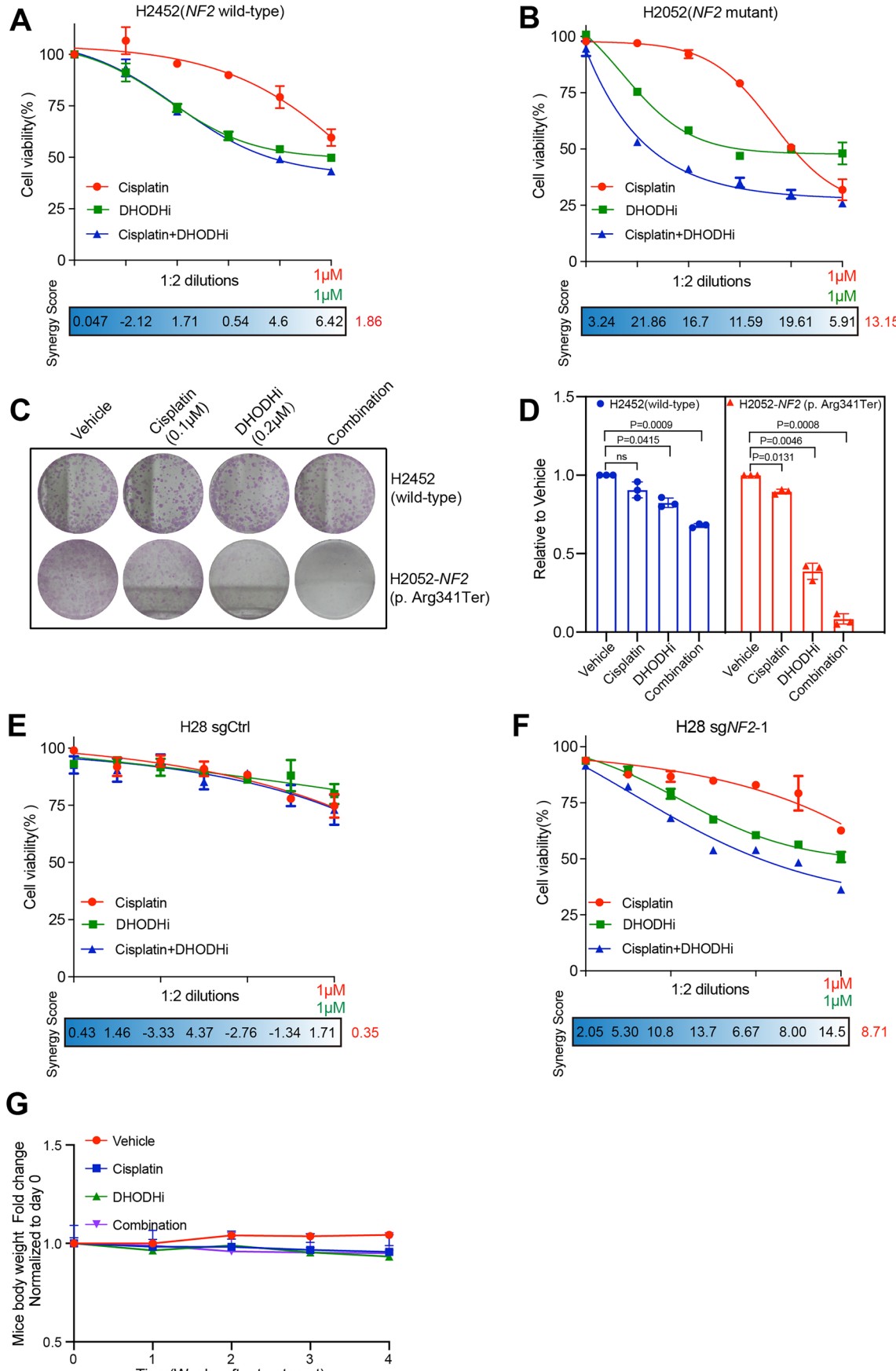

◄ **Figure EV8. DHODH inhibition exhibits a synergistic antitumour effect with cisplatin in PM.**

(A, B) Cell viability of wild-type (H2452; A) and *NF2*-mutant (H2052; B) PM cell lines treated with the indicated doses of drugs for 96 h. Representative data from three independent experiments are shown. The data are presented as the mean ± SD. The highest single agent (HSA) synergy scores at the independent combination doses were calculated via SynergyFinder +. A synergistic score <−10 indicates that the interaction between two drugs is likely antagonistic; a score ranging from −10–10 indicates that the interaction is likely additive; and a score>10 indicates that the interaction is likely synergistic. (C, D) Clonogenic assay of H2452 (*NF2* wild-type) and H2052 (*NF2* mutant) PM cells treated with vehicle or the indicated drugs for 96 h. After a 14-day culture period, the viable cells were stained with crystal violet. Representative images (C) and quantification (D) from three independent experiments are shown. The data are presented as the mean ± SD. Two-way ANOVA with multiple comparisons was used for statistical analysis. (E, F) Cell viability of wild-type (H28 sgCtrl; E) and *NF2*-deficient PM cell lines (H28 sg*NF2*-1; F) treated with the indicated doses of drugs for 96 h ($n = 3$). The data are presented as the mean ± SD. Representative results from three independent experiments are shown. (G) The body weights of mice during the indicated experiment. The data are presented as the mean ± SEM ($n = 3$ or 4 per group).

