## [Peer Review File · EMBO Molecular Medicine]

De Novo Pyrimidine Synthesis Is a Collateral Metabolic Vulnerability in NF2-deficient Mesothelioma

Duo Xu, Yanyun Gao, Shengchen Liu, Shiyuan Yin, Tong Hu, Haibin Deng, Tuo Zhang, Balazs Hegedüs, Thomas M. Marti, Patrick Dorn, Shunqing Liang, Ralph Schmid, Ren-Wang Peng, and Yongqian Shu

Corresponding authors: Yongqian Shu (shuyongqian@cscs.org.cn), Ren-Wang Peng (Renwang.Peng@insel.ch), Duo Xu (xuduo@jsph.org.cn)

Review Timeline:

Submission Date:	19th Mar 24
Editorial Decision:	18th Apr 24
Re-submission Date:	6th Mar 25
Editorial Decision:	11th Apr 25
Revision Received:	23rd May 25
Editorial Decision:	18th Jun 25
Revision Received:	7th Jul 25
Accepted:	9th Jul 25

Editor: Lise Roth

Transaction Report:

Dear Prof. Shu,

Thank you for the submission of your manuscript to EMBO Molecular Medicine. We have now received feedback from the three referees who agreed to evaluate your work.

As you will see from the enclosed reports, the referees acknowledge the potential interest of the findings, however, referees #2 and #3 also raise several major concerns on the study, and do not feel that the conclusions are sufficiently supported by the data at this point.

Given the nature of the referees' concerns and the amount of time and work that would be required to address them, and considering that at EMBO Press we encourage one round of revisions only in a reasonable time frame, I am afraid I see little choice but to return the manuscript to you at this point with the decision that we cannot offer to publish it.

Given the potential interest of the findings, we would, however, have no objection to consider a new manuscript on the same topic if at some time in the near future you obtained data that would considerably strengthen the message of the study and address the referees concerns in full. To be completely clear, however, I would like to stress that if you were to send a new manuscript this would be treated as a new submission rather than a revision and would be reviewed afresh, in particular with respect to the literature and the novelty of your findings at the time of resubmission. If you decide to follow this route, please make sure you nevertheless upload a letter of response to the referees' comments.

At this stage, though, I am sorry to have to disappoint you. I nevertheless hope that the referee comments will be helpful in your continued work in this area, and I thank you for considering EMBO Molecular Medicine.

Yours sincerely,

Lise Roth

***** Reviewer's comments *****

Referee #1 (Comments on Novelty/Model System for Author):

Page 13, para 2, line 1: "Mouse studies were conducted in accordance with Institutional Animal Care and Ethical Committee-approved animal guidelines and protocols."... Please indicate exact approval numbers and dates for all animal studies.

Referee #1 (Remarks for Author):

With interest I reviewed the above-referenced manuscript that identifies a synthetic lethality in NF2-deficient mesothelioma. The dataset is robust, the results are convincing, and the analyses and experiments done constitute a multifaceted and robust piece of evidence. The authors should address the three comments conveyed below:

RAS signaling promotes De Novo Pyrimidine Synthesis (Ref 25) and NF2 is a RAS inhibitor. Could NF2 promotes De Novo Pyrimidine Synthesis via aberrant RAS signaling?

Page 13, para 2, line 1: "Mouse studies were conducted in accordance with Institutional Animal Care and Ethical Committee-approved animal guidelines and protocols."... Please indicate exact approval numbers and dates for all animal studies.

Page 7, para 2, line 5: "Fisher's exact test comparisons for every two subtypes"... The hypergeometric test would be appropriate here (<https://systems.crump.ucla.edu/hypergeometric/index.php>). Please calculate and show results for cluster 1 and NF2.

Referee #2 (Comments on Novelty/Model System for Author):

Commercially available mesothelioma cell lines have been shown to diverge substantially from primary PDCL specifically as regards to their transcriptomic and metabolic phenotypes. As such, the omission of any PDCLs from the study is a major concern. The relevance of xenografts in immunocompromised mice for a cancer that is driven by inflammation and treated with immunotherapy is also highly questionable

Referee #2 (Remarks for Author):

The manuscript by Xu and colleagues describes a novel synthetic dependency of NF2-deficient Mesothelioma for the de novo pyrimidine synthesis pathway. The authors use an elegant multi-omic approach to identify elevated expression of CAD and DHODH in Mesothelioma cell lines deficient for or depleted of NF2 and suggest that loss of NF2 leads to YAP/TEAD-dependent upregulation of both enzymes. Moreover, they show a potential therapeutic vulnerability specific to NF2-deficient Mesothelioma cells using a small molecule inhibitor of DHODH. Overall there is much to the manuscript that shows merit, however, a number of deficiencies and several overstatements detract considerably from the manuscript and weaken the case for publication.

1) Of greatest concern, commercially available Mesothelioma cell lines, including some of those used herein, were previously shown to diverge considerably from primary patient-derived Mesothelioma lines, specifically resulting in alterations to metabolism that may exaggerate the metabolic dependencies and indeed clinical relevance of the present observations (Chernova et al. Cell Death & Differentiation, 2016). While it would be unreasonable to expect the entire study to be repeated from scratch, some key findings should be investigated in primary PDCLs, which are readily available through Mesobank.com or Netmeso.fr

Other major concerns:

2) Although the authors convincingly show that deletion of NF2 upregulates CAD and DHODH in cell culture, the IHC analysis of the TMA is less supportive (Figure 1 H-K). To begin with the TMA analysis is performed on very low numbers of samples (2 cores from 17 patients) and does not appear to be adequately powered to avoid a

Type I error. Additionally, while high NF2 does appear to correlate with low DHODH expression, no similar analysis is provided for CAD, despite data showing that CAD levels are higher in tumours than in adjacent normal.

3) The short-term cell viability assays shown in Figures 3C, 3D & Figure S8E have not been normalised properly and exaggerate the effects of the DHODH inhibitor used. Cell viability for each cell line, modified or not, should be set to 100% in the absence of drug and the drug effects normalised separately, relative to the untreated sample for each cell line. Moreover the "best fit" regression lines bear little relationship to the data points shown and should be omitted altogether. This latter point also applies to Figure 4E, 4F, Figure S8C and S8H.

Fortunately, the effects on longer term cell growth and spheroid formation are much more convincing and may suggest a delayed response rather than an immediate response.

4) The authors make an interesting point about low glucose forcing cells to use the pyrimidine salvage³ pathway which would plausibly provide a mechanisms of resistance to DHODH inhibition. Mesothelioma is notoriously poorly vascularised and thus low glucose would be expected in situ. This should be tested experimentally, as was done for the UCK2 inhibitor.

5) The level of apoptosis shown In Figure 4C and 4D is surprisingly low, especially given the very strong γ H2AX signal. This suggests the effect of the DHODH inhibitor may be primarily cytostatic. Note that γ H2AX signal increases during Mitosis. The authors should investigate the cell-cycle effects of the drug. Moreover, this analysis should be repeated in cell lines already deficient for NF2, not just following NF2 manipulation.

6) Similarly, the mouse xenograft studies should be expanded to include additional NF2 deficient cell lines (preferably primary PDCLs), not just following NF2 deletion in a single line.

7) The data claiming direct regulation of CAD and DHODH expression by YAP/TEAD are weak and are not supported by their examination of ChIP-SEQ data. In Figure S10 panel D, the Y axis scale for YAP binding to the CAD promoter in 2 cells lines ranges only from 0 to 1, as compared with 0-2555 for H3K27Ac. The same is true for TEAD binding. These axis labels are moreover only visible upon magnification to 400%.

In Panel E of the same figure, YAP and TEAD do bind the DHODH enhancer region quite well, however, their binding profiles do not overlap and are shown in different cells.

The effect of YAP depletion or inhibition on expression of CAD and DHODH shown in

Figure 5F is marginal and inconsistent.

8) The Ki67 staining in Figure 11E is misleading. Ki67 is a binary signal that either shows strong positive staining or none at all, not the "graded" staining suggested for the drug-treated sgNF2 sample. It would appear that this sample was not allowed to develop for long enough and should be repeated. The data should moreover be quantified.

Additional points that require attention:

1) In Figure S1, the authors compare gene expression in 328 Mesothelioma samples with 65 normal tissues. The authors should clarify if all (or indeed any) of these normal tissues are pleural.

2) References for the datasets used in the analysis performed in Figure 1 (and Figure S1-4) should be included in the materials and methods and cited accordingly

3) The Venn diagram shown in Fig 1D is confusing and could be omitted altogether.

The message is already effectively represented by panel 1E

4) P53 loss is known to have profound metabolic consequences and occurs in up to 20% or so of mesothelioma - it would be interesting to see how it aligns with the different clusters

5) Fig 1 panel F makes little sense. It is described as a heatmap in the legend but is not one in any sense that I am familiar with. As drawn, several of the labels are very difficult to make out. What is the significance of the different sized rectangles?

6) The authors applied different (arbitrary) cut-offs for fold change in their integrated analysis of transcriptomic and proteomic data. Justification for the choice of cut-offs should be provided.

7) The Y axes of data shown in Figure S7C, D and F are all tailored to exaggerate the effect of NF2 deletion. Please redraw these data using Zero values at the axis intercepts.

8) The pleural cell line data are very difficult to discern in Figure S8H. Again the regression line shown looks rather inflated based on the few pleural cell lines I can make out at high magnification.

9) The drug effect in the orthotopic mouse model (Figure 11D) is not significant, although claimed in the text to be "marked".

Referee #3 (Comments on Novelty/Model System for Author):

Identifies potential new vulnerability in a difficult cancer type. Identifies a new mechanism, but further evidence is needed to support the main claims. Some of the statistical analysis should be improved, but that is not a critical issue in this particular paper. They use a pre-clinical model, patient data are queried from available databases.

Referee #3 (Remarks for Author):

Title: "De Novo Pyrimidine Synthesis Is a Collateral Metabolic Vulnerability in Patients with Malignant Pleural Mesothelioma Harboring NF2 Deficiency"

General: The manuscript titled 'De Novo Pyrimidine Synthesis Is a Collateral Metabolic Vulnerability in Patients with Malignant Pleural Mesothelioma harboring NF2 Deficiency' by Xu Duo et al characterizes a subtype of malignant pleural mesothelioma (MPM), which is determined by the loss of NF2 tumor suppressor and a subsequent activation of YAP proto-oncogene, which in turn upregulates de novo pyrimidine synthesis pathway. The concept that de novo pyrimidine synthesis becomes a targetable vulnerability in certain condition is not entirely new, but the authors are the first to show this for MPM. In addition, they present a new finding that pyrimidine synthesis genes DHODH and CAD are directly transcriptionally regulated by YAP. While the authors provide an important insight into a subset of MPM and potential therapeutical targets, additional work and evidence is needed to reinforce authors' reasoning and conclusions. The specific points are laid out below.

Major points:

1. Authors refer extensively to the upregulation of de novo pyrimidine synthesis in NF2-deficient MPM cells, but this major aspect of the study has not been directly supported by evidence. The authors show that CAD and DHODH are upregulated (mRNA and protein level), but do not provide evidence that the de novo pyrimidine synthesis pathway is actually more active. Increased nucleotide levels in NF2-deficient cells (presented in Fig. 1) do not conclusively demonstrate an increased activity of the de novo pathway. Therefore, metabolic flux measurements, for example glutamine to UMP C13 tracing, must be provided to demonstrate an increased de novo pyrimidine pathway flux in NF2 deficient cells, and enzymatic activity should be measured for DHODH or CAD.
2. Following from point 1 above: Please provide evidence, using inhibitors or KOs,

that salvage does not contribute to increased nucleotide levels/increased proliferation observed in NF2-deficient cells. Fig S7 shows that UPP1 expression is significantly upregulated in NF2-deficient cells. Indeed, the statement: "Interestingly, the protein levels of uridine phosphorylase 1 (UPP1) and uridine-cytidine kinase 2 (UCK2), pivotal mediators of the pyrimidine salvage pathway, were not markedly different between the wild-type and NF2-deficient groups." is not supported by the figure S7F since there is a significant upregulation in UPP1 protein levels.

3. For the key experiments where CRISPR-induced NF2 deficient clones are used, the authors need to provide a reconstitution/addback control using sgRNA-resistant cDNA.

4. What happens with UMPS? It is not mentioned anywhere, information about its expression level in NF2-deficient cells needs to be provided.

5. Similarly, authors never discuss/assess TAZ, even though TAZ can complement YAP. What happens to TAZ in NF2-deficient cells and how TAZ reacts to YAP manipulations? Does the inhibitor used for YAP also inhibit TAZ?

6. Please provide stronger evidence that YAP regulates DHODH and CAD expression. Functional assay, for example luciferase reporters, are required to demonstrate expression from DHODH and CAD promoters is enhanced by YAP.

7. The authors should provide better reasoning of why they focused on NF2 in their manuscript. There are other oncogenes/tumor suppressor involved in the MPM etiology.

8. The Crispr-generated NF2-deficient clones come up in the text figures without any proper introduction, making it difficult to follow the story line. They should be properly introduced, demonstrating the level of the KO on their first appearance. Reconstitution control should be provided (see point 2 above).

9. For all the presented western blots, quantifications from at least 3 independent experiments should be provided, as some of the blots are quite unconvincing.

10. The curve fitting presented in this manuscript is inappropriate (see figures 3C, 4E, 4F, S8E, S9A, S9C-E.). Please redo these curve fits, consider using a wider concentration range of inhibitors.

11. Single cell RNAseq data are discussed in the Discussion section but are not included in the manuscript: "Intriguingly, our ongoing single-cell RNA sequencing (scRNA-seq) study revealed that malignant mesothelioma cells exhibit robust enrichment of nucleotide metabolism-related genes, among which DHODH is expressed at lower levels in immune cells, such as T and natural killer (NK) cells, further indicating that targeting DHODH might be an ideal therapeutic approach for

precisely regulating nucleotide synthesis in cancer cells without easily damaging the tumor immune microenvironment (TIME). Indeed, the metabolic interplay between cancer cells and immune cells and how this crosstalk impacts immune surveillance and antitumour immunity in MPM deserve further investigation." To allow assessment of these conclusions, please include these results and methods in the manuscript, and make the raw data available at a dedicated public repository. Alternatively, the authors are asked to remove this part of discussion.

12. "Based on the TCGAMESO dataset, MPM patients with genetic inactivation of NF2 had an extremely poor prognosis (Figure S5A, B)." - the difference on S5B is not significant, the statement is thus not true. Please correct.

Minor points:

13. Graphical abstract: lower right part refers to clinical analysis, implying work with patients. In fact, only existing clinical data were evaluated. Please correct.

14. Superiority statements and statements about 'significance' - unless supported by a proper statistical analysis - must be avoided. Generally, only the statistically significant results are discussed, therefore, to emphasize this further is confusing.

15. "Notably we found that both the transcript and protein levels of CAD and DHODH, the critical rate-limiting enzymes involved in the de novo pyrimidine biosynthesis pathway, were significantly elevated in NF2-deficient MPM cells compared to those in wild type group" - These enzymes are undoubtedly critical, but the question is whether they are also rate-limiting. In the author's model, no evidence of that is provided. Please rephrase.

16. Please, unify the general style of the manuscript. Sometimes spaces are used and other times they are missing

a. i.e. Figure 1.E "...MPM patients (n = 87)" vs Figure 1.D "...TCGA-MPM cohort (n=87)".

b. "p <0.05 was considered to indicate statistical significance".

The presented graphs mostly use p values, but other times stars are used to show significance (S5F, S7E, Figure 6A).

There are spaces between citation numbers and words.

Please keep the lettering of the figures from left to right row-wise (for example, figure 2H is mixed up).

17. The meaning of this statement is unclear: "NF2 status defines a unique subset of MPM characterized by abnormal de novo pyrimidine synthesis." Please indicate what abnormal means (upregulated/downregulated).

18. It is unclear what: "pyrimidine metabolism is the primary metabolic subtype of

MPM" means. Based on the results, it is not the only metabolic subtype.

19. Data retrieval and preprocessing - please state which DepMap datasets you used for the analysis. In the methods section "Comparison of genetic alteration frequencies among subtypes", please specify which datasets were used.

20. "MPM tumor tissues showed an apparent enrichment in a variety of metabolic processes compared to normal tissues, suggesting metabolic adaptation or reprogramming during tumorigenesis". This statement does not describe what the figures show. Figure S1 actually shows that there are more downregulated metabolic genes than upregulated, and Figure S2 also shows that more pathways are upregulated in the normal tissue than in the tumor.

21. Reference 32 does not say anything about NF2.

22. "Unlike the research hotspots in the field of cancer cell metabolism, pyrimidine biosynthesis is a less appreciated aspect; however, it is essential to produce large amounts of cellular building blocks to meet the needs of cell proliferation and has been recently documented to conquer ferroptosis for cell survival 25, 42." - The statement is overblown. Please cite PMID 30449682 which shows that pyrimidine synthesis is essential for tumor growth. What do you mean by "conquer ferroptosis"? Please rephrase.

Additional specific comments on individual figures (beyond the general ones discussed above):

Figure 1:

Figure 1A - the information from the legend should be in the methods. 1A also represents different datasets. How were these datasets integrated? Please describe it in methods. Does each bar represent one patient? Explain this in the legend.

Figure 1G - labels are missing in the top 3 graphs.

The statement: "In an effort to identify the oncogenic signalling crosstalk with pyrimidine metabolism, we observed that NF2 loss of function (mutations and/or homozygous deletions) was more frequent in Cluster 1 (73%) than in the other metabolic subgroups (27%) (Figure 1D, E; Table S15)" is not well supported by the figures. Please provide better visualization.

Figure 1E - how were the patients stratified into the metabolic subgroups since this dataset is presumably different from the one used in Figure 1A? If the number of mutations would be relative to the size of the cluster (number of patients) would cluster 1 still have the most NF2 mutations? All major mutated genes in mesothelioma should be analyzed.

Legend Figure 1E: "Genetic status of NF2, BAP1, and SLC12A1 among different metabolic subgroups in the TCGA cohort of MPM patients (n = 87). The data were downloaded from the (<https://www.cbioportal.org/>)."

Which data was downloaded? Are these the same datasets as those indicated in methods - "Comparison of genetic alteration frequencies among subtypes" from xenabrowser?

Figure 1D - it is unclear what the venn diagrams contain and represent. Please use better visualization. Why did the authors focus only on NF2 if SLC12A1 and BAP1 are also significant?

Figure 1F is difficult to interpret. A heat map of individual genes would be more appropriate. Please indicate the number of biological and technical replicates used to generate this data. Are the results consistent in other cell lines (H28)? Which H2452 sgNF2 cells were used (in the rest of the manuscript, there are two different sgRNAs)?

Figure 1G - Are there any other significantly different metabolites (unrelated to pyrimidine synthesis)?

Figure 2:

Figure 2B should be improved to better visualize these important results. Which genes from proteomics and transcriptomics were the most significantly different? Why are absolute values of the fold changes being compared? This way, the information whether the genes were upregulated or downregulated is lost. Also, how did the authors set the thresholds? 0.5 and 1.3 seems rather arbitrary. Also see the major point 8 above.

Remove unnecessary pathways in Figure 2C and S1B (i.e. African trypanosomiasis, pertussis). Please explain what is the "gene ratio" on x axis.

Figure 2G - To refer to these results from 3 cell lines as results from a 'panel' is inappropriate. 3 cell lines do not constitute a panel. Please rephrase.

Figure 3

Figure 3A - statistical comparison is done between two different groups, t-test is thus not appropriate. Please correct. Figure 3A proliferation assay is not included in the methods. Time course would be preferable.

Figure 3A, B - the silencing efficacy in these experiments should be documented on the protein level.

Figure 4

Figure 4A - uridine concentrations used are very high. Can supplementation with physiological concentrations of uridine rescue the phenotype? Please include control for 0 uM of DHODHi and for a Nf2 WT cell line - preferentially in H2452 shctrl.

Figure 4B Is the DNA damage and apoptosis upon DHODH inhibition recapitulated in other cell lines (H28, H2052) upon NF2 KO? This would allow to make more general conclusions about MPM.

Figure 4EF (and also SF9) Different cell lines are used, please show how H2452 WT compares to H2452 NF2 KO in one graph, not to H2052 in separate graphs. Curves do not fit the data as indicated above in the major points.

Figure 4H is missing x axis labels.

Figure 4G, H - please indicate what is the baseline for the viability. The y axis label is not descriptive enough.

Figure 5

Figure 5I: Why was the chip experiment done only in wt cells? If the author's hypothesis is true, the chip signal should be stronger in NF2 KO cells. Control genes, not only CAD and DHODH, should be included in this experiment.

In the related supplementary figure S10C: While this statement is true:

"Interestingly, further examination of the TCGA cohort of MPM, we found that the mRNA level of YAP is positively correlated with CAD but negative correlation with enzymes of the pyrimidine salvage pathway (UPP1, CDA, and UCK2) (Figure S10C).", there is no correlation with DHODH and a negative correlation with UMPS, two important enzymes of de novo pyrimidine synthesis pathway. How do the authors explain this discrepancy?

Figure 6

In the related text:

"Importantly, the DHODH inhibitor brequinar obviously retarded tumor growth in NF2-deficient group without apparent influence on wild-type group and markedly prolonged the survival of mice bearing orthotopically transplanted MPM tumors (Figure 6A-E; Figure S11D)." The difference in survivability between sgNF2 and sgNf2+DHODHi (S11D) is not statistically significant. Please correct the statement.

Supplementary figures:

Supplementary Figure 3A - the authors should include in the methods which genes were included in a "Gene signature of pyrimidine metabolism" and how it was

calculated. Please make larger spaces between the individual cancer types for clarity. The tumors, where data from normal tissue are not included, should be removed, as that is confusing and lacks reporting value.

Supplementary Figure 3B - please explain what "mean value of pyrimidine metabolic signatures" means.

Figure S4 In the related text: „we consistently showed that differentially expressed metabolites in human MPM cell lines belonging to Cluster 1 were strongly enriched in pyrimidine biosynthesis (Figure S4E, F; Table S13,14)." - the highlighted cell lines (blue), that belong to cluster 1, are actually not clustering. Please correct the statement and/or the conclusion from this data.

Supplementary Figure 4A is never referenced in the text, please remove.

Supplementary Figure 4F - there are two scales for the p value (shade and x axis), please use only one.

Supplementary Figure 5D - y axis depicts cell number, not proliferation rate, y axis title is unclear, please indicate what is being measured.

Supplementary Figure 6B - according to the legend, log₂FC is represented by both color and size of the dots. Please correct.

Supplementary Figure 6C - please clarify the y axis label, the protein expression needs to be relative to a reference.

Below are our point-by-point responses, with the reviewers' suggestions and comments in black and our responses in blue, and quoted text from the manuscript *in purple and italics*.

Referee #1 (Comments on Novelty/Model System for Author):

Page 13, para 2, line 1: "Mouse studies were conducted in accordance with Institutional Animal Care and Ethical Committee-approved animal guidelines and protocols."... Please indicate exact approval numbers and dates for all animal studies.

Thank you for pointing out that we did not describe this adequately. We now provide that information in the revised manuscript ("Materials and methods" section, pages 16&17).

All protocols for mouse experiments were approved by Nanjing Medical University (IACUC-1706007-3-2022) and the Canton of Bern (BE85/2023) in Switzerland.

Referee #1 (Remarks for Author):

With interest I reviewed the above-referenced manuscript that identifies a synthetic lethality in NF2-deficient mesothelioma. The dataset is robust, the results are convincing, and the analyses and experiments done constitute a multifaceted and robust piece of evidence. The authors should address the three comments conveyed below:

Thank you for the highly positive feedback on our study.

RAS signalling promotes De Novo Pyrimidine Synthesis (Ref 25) and NF2 is a RAS inhibitor. Could NF2 promotes De Novo Pyrimidine Synthesis via aberrant RAS signalling?

Given that RAS signalling promotes de novo pyrimidine synthesis by upregulating enzymes or pathways involved in this process (Reference 25) and that NF2 (neurofibromin 2, also known as Merlin) functions as a tumor suppressor and has been shown to negatively regulate RAS signalling, it's possible that loss of NF2 function could promote de novo pyrimidine synthesis through aberrant RAS signalling,

To validate this hypothesis, we performed new experiments in *NF2* wild-type H2452 mesothelioma cells. Our results showed that genetic knockdown (KD) of *KRAS* (siRNA) not only decreased the mRNA level of *KRAS*, but also of *CAD* and *DHODH* in H2452 cells expressing a control sgRNA (H2452 sgRNA), confirming that *KRAS* signalling promotes pyrimidine metabolism programming. However, in *NF2* depleted H2452 cells (H2452 sgNF2), we did not observe obvious change in the transcript levels of *CAD* and *DHODH* upon *KRAS* KD (See Figure, below).

Based on these results, we conclude that *NF2* deficiency-induced alterations in pyrimidine metabolism is independent of aberrant *KRAS* signalling in mesothelioma. We discussed this interesting finding in the revised manuscript (page 29) as follows: “*RAS* signalling has been reported to promote *de novo* pyrimidine synthesis, in addition *NF2* is a well-known *RAS* inhibitor (Wang, Cui et al., 2021). However, in the *NF2* deletion group, there were no obvious alterations in the transcriptional levels of *CAD* and *DHODH* after interfering with *KRAS* (data not shown), indicating that *NF2* deficiency-induced changes in pyrimidine metabolism are independent of *KRAS* signalling in mesothelioma.”.

Effect of *NF2* on *KRAS* signalling and pyrimidine metabolism in mesothelioma cells. (A-C) mRNA expression of *KRAS*, *CAD*, and *DHODH* in the indicated groups transfected with nontargeting control siRNAs or siRNAs targeting *KRAS* for 72h. The data are presented as the mean \pm S.D. (n=3). Multiple unpaired t tests/Two-way ANOVA were used for comparisons.

Page 13, para 2, line 1: "Mouse studies were conducted in accordance with Institutional Animal Care and Ethical Committee-approved animal guidelines and protocols."... Please indicate exact approval numbers and dates for all animal studies.

Thank you for pointing this out, and we now provide that information in the revised manuscript (pages 16&17) as follows: “All protocols for mouse experiments were approved by Nanjing Medical University (IACUC-1706007-3-2022) and the Canton of Bern in Switzerland (BE85/2023).”.

Page 7, para 2, line 5: "Fisher's exact test comparisons for every two subtypes"... The hypergeometric test would be appropriate here (<https://systems.crump.ucla.edu/hypergeometric/index.php>). Please calculate and show results for cluster 1 and NF2.

Thank you for this constructive suggestion. We performed a hypergeometric test and showed that the p value was 0.00088; please see new Supplementary Figure S4G. Moreover, we cited the above user-friendly web-based implementation in the methods section (page 7).

Referee #2 (Comments on Novelty/Model System for Author):

Commercially available mesothelioma cell lines have been shown to diverge substantially from primary PDCL specifically as regards to their transcriptomic and metabolic phenotypes. As such, the omission of any PDCLs from the study is a major concern. The relevance of xenografts in immunocompromised mice for a cancer that is driven by inflammation and treated with immunotherapy is also highly questionable.

We appreciate the reviewer's concerns regarding the use of commercially available mesothelioma cell lines and the relevance of xenograft models in our study. Thus, we have performed additional experiments by incorporating several PDCLs in this study. The results are shown in the new Figures 3, 4 & 6 in our revised manuscript. Notably, these new results confirmed that *NF2*-deficient PDCLs generally exhibited heightened sensitivity to the inhibition of *de novo* pyrimidine metabolism *in vitro* and *in vivo*, therefore further strengthening our findings and conclusion.

We also agree that immunocompromised xenograft models have limitations, particularly for cancers driven by inflammation and treated with immunotherapy. However, our primary focus in this study was to investigate tumor-intrinsic mechanisms related to *NF2* deficiency and metabolic alterations, rather than immune-related processes. The use of xenografts in immunocompromised mice allowed us to study these tumor cell-intrinsic effects in a controlled *in vivo* environment without the confounding influence of an intact immune system.

Nevertheless, our findings promise further investigations to address the relevance of our findings to immunotherapy and inflammation-driven processes. Therefore, we have included a statement in our revised manuscript that acknowledges the limitations of xenograft models in this context (Limitations of the study; page 31): "*This study utilized immunocompromised xenograft models to investigate tumor-intrinsic mechanisms; however, we acknowledge the*

inherent limitations of these models, particularly in relation to inflammation-driven cancers and their responses to immunotherapy. Future research should incorporate syngeneic or humanized mouse models to more accurately evaluate the interactions between tumor-intrinsic alterations and the immune microenvironment.”.

In summary, we highly have incorporated the new results from PDCLs into the revised manuscript, which further strengthens the findings of the present study.

Referee #2 (Remarks for Author):

The manuscript by Xu and colleagues describes a novel synthetic dependency of NF2-deficient Mesothelioma for the de novo pyrimidine synthesis pathway. The authors use an elegant multi-omic approach to identify elevated expression of CAD and DHODH in Mesothelioma cell lines deficient for or depleted of NF2 and suggest that loss of NF2 leads to YAP/TEAD-dependent upregulation of both enzymes. Moreover, they show a potential therapeutic vulnerability specific to NF2-deficient Mesothelioma cells using a small molecule inhibitor of DHODH. Overall, there is much to the manuscript that shows merit, however, a number of deficiencies and several overstatements detract considerably from the manuscript and weaken the case for publication.

We sincerely thank the reviewer for recognizing the merits of our manuscript, including the use of a multi-omics approach, the identification of a YAP/TEAD-dependent upregulation of CAD and DHODH, and the potential therapeutic vulnerability of NF2-deficient mesothelioma cells to DHODH inhibition. We also appreciate the constructive feedback regarding areas of deficiency and potential overstatements. Below, we address these concerns in detail:

1) Of greatest concern, commercially available Mesothelioma cell lines, including some of those used herein, were previously shown to diverge considerably from primary patient-derived Mesothelioma lines, specifically resulting in alterations to metabolism that may exaggerate the metabolic dependencies and indeed clinical relevance of the present observations (Chernova et al. Cell Death & Differentiation, 2016). While it would be unreasonable to expect the entire study to be repeated from scratch, some key findings should be investigated in primary PDCLs, which are readily available through Mesobank.com or Netmeso.fr

We appreciate the reviewer's important point regarding the divergence between commercially available mesothelioma cell lines and primary patient-derived cell

lines (PDCLs), as highlighted in the study by Chernova et al. (Cell Death & Differentiation, 2016). We agree that the use of PDCLs would enhance the translational relevance of our findings. We therefore performed new *in vitro* and *in vivo* experiments by incorporating PDCLs in this study (new Figures 3 and 6; pages 23&26), which consistently showed that *NF2*-deficient PDCLs, importantly, were more sensitive to the inhibition of *de novo* pyrimidine metabolism. In the meantime, DHODH inhibition exerted a lethal anti-tumor effect by triggering obvious DNA damage and apoptotic cell death in PDCLs (new Figure 4; page 24). The above findings further strengthen the clinical value of this study. Moreover, we have cited this reference (Chernova *et al.* Cell Death & Differentiation, 2016) in the manuscript.

Other major concerns:

2) Although the authors convincingly show that deletion of *NF2* upregulates CAD and DHODH in cell culture, the IHC analysis of the TMA is less supportive (Figure 1 H-K). To begin with the TMA analysis is performed on very low numbers of samples (2 cores from 17 patients) and does not appear to be adequately powered to avoid a Type I error. Additionally, while high *NF2* does appear to correlate with low DHODH expression, no similar analysis is provided for CAD, despite data showing that CAD levels are higher in tumours than in adjacent normal.

We thank the reviewer for your valuable critique of the limited numbers of samples (n=17, Nanjing cohort). We therefore performed new experiments by incorporating another tissue microarray cohort from 98 pleural mesothelioma patients that we have previously described (Xu, Gao et al., 2024). The results from our new analysis confirmed that mesothelioma patients with low *NF2* protein levels (*NF2*_{low}) displayed an increased tendency of CAD and DHODH (new Figure 2; page 23 in the revised manuscript).

Notably, PM patients with high expression of CAD and DHODH had a poorer prognosis (Figure S7G). However, no significant correlation was identified between NF2 and CAD expression in Nanjing cohort (Figure 2I, J), while in Bern cohort, patients with low NF2 levels (NF2_{low}) showed an increase in CAD expression (Figure 2L, M). Meanwhile, DHODH expression was notably elevated in NF2_{low} group across both cohorts (Figure 2K, N), reinforcing the link between NF2 loss and enhanced pyrimidine biosynthesis.

Taken together, we thank the reviewer for your valuable feedback, which has allowed us to present a more nuanced and comprehensive analysis of the clinical data. These revisions further enhance the rigor and translational impact of our manuscript.

3) The short-term cell viability assays shown in Figures 3C, 3D & Figure S8E have not been normalised properly and exaggerate the effects of the DHODH inhibitor used. Cell viability for each cell line, modified or not, should be set to 100% in the absence of drug and the drug effects normalised separately, relative to the untreated sample for each cell line. Moreover the "best fit" regression lines bear little relationship to the data points shown and should be omitted altogether. This latter point also applies to Figure 4E, 4F, Figure S8C and S8H.

Fortunately, the effects on longer term cell growth and spheroid formation are much more convincing and may suggest a delayed response rather than an immediate response.

We thank the reviewer for highlighting the issues with data normalization and presentation in the short-term cell viability assays. Retrospective analysis indicate that we indeed performed the data analysis by normalizing the data to the untreated sample for each cell line, and we now show 100% of the data points in the new figures. Moreover, we redid the "best fit" regression lines using log(inhibitor) vs. response-Variable slope (four parameters) to better cover the data points shown. The revised data have been incorporated into new Figures 3, S8 and 9 along with updated legends and corresponding descriptions in the Results section.

In addition, Figure S8C and S8H (new Figure S8E&O) show the Spearman coefficients between the two indicated markers, and the regression line represents the general trend compared with the actual slope. In brief, linear regression can find the best line to predict Y from X. Correlation quantifies the degree to which two variables are related. The correlations do not fit straight lines through the data points.

We also agree that the results from the clonogenic assay and spheroid formation experiments, which generally reflect the long-term drug response, showed sharper differences than did the results of the 120-hour cell viability assays and suggested a delayed response. In support of this, our new results show that DHODH inhibitors induced obvious cell cycle arrest in *NF2*-deficient mesothelioma cells (new Figure 4B, C).

In summary, we thank you for pointing out these important issues and for their constructive feedback. These revisions will improve the rigor and accuracy of our data presentation, ensuring that the conclusions drawn are both reliable and transparent.

4) The authors make an interesting point about low glucose forcing cells to use the pyrimidine salvage pathway which would plausibly provide a mechanism of resistance to DHODH inhibition. Mesothelioma is notoriously poorly vascularised and thus low glucose would be expected in situ. This should be tested experimentally, as was done for the UCK2 inhibitor.

We thank the reviewer for your insightful comment highlighting the potential role of the pyrimidine salvage pathway as a mechanism of resistance to DHODH inhibition under low-glucose conditions. We therefore performed new experiments to evaluate the drug response to DHODH inhibition under normal and low-glucose conditions in both wild-type and *NF2* deletion mesothelioma cells. Interestingly, glucose limitation did not affect the drug sensitivity to DHODH inhibition, especially after the deletion of *NF2*, further indicating that the survival of *NF2*-deleted mesothelioma cells might be independent of pyrimidine salvage pathways even under low-glucose conditions (new Figure S9F-H).

Recognizing the importance of this point, we have expanded the manuscript to address this aspect in greater depth by including an elaboration of these interesting findings in the revised manuscript (page 24).

Given that mesothelioma is poorly vascularized, resulting in limited glucose availability in situ (Ohta, Shridhar et al., 1999, Ostergaard, Tietze et al., 2013), we further assessed the sensitivity to DHODH inhibitors under glucose-limited conditions. Glucose restriction did not alter drug sensitivity, indicating that NF2-deficient PM cells maintain their reliance on pyrimidine biosynthesis even under metabolic stress (Figure S9F-H).

5) The level of apoptosis shown in Figure 4C and 4D is surprisingly low, especially given the very strong γ H2AX signal. This suggests the effect of the DHODH inhibitor may be primarily cytostatic. Note that H2AX signal increases during Mitosis. The authors should investigate the cell-cycle effects of the drug. Moreover, this analysis should be repeated in cell lines already deficient for *NF2*, not just following *NF2* manipulation.

We thank the reviewer for their insightful comment and appreciate the suggestion to investigate the cell-cycle effects of the drug and to repeat the analysis in cell lines that are inherently *NF2* deficient.

Our new experiments revealed that DHODH inhibition resulted in prominent cell cycle arrest in S-phase and substantial DNA damage in both commercial and primary mesothelioma cell lines with defective *NF2* (Figure 4B-E). To a certain

extent, under the DHODH inhibitor treatment, increased apoptotic cell death was observed in commercial cell lines following genetic knockout of *NF2* (Figure 4F). Intriguingly, we found that DHODH inhibition induced abundant apoptosis in PDCLs with defective *NF2*, especially in PF747 cells around 40% apoptotic cell death was induced after DHODH inhibitor treatment (Figure 4G).

Overall, our findings suggest a differential effect of DHODH inhibition on commercial cell lines and PDCLs, consistent with the previous observations that there is considerable divergence between established cell lines and primary patient-derived mesothelioma cells (Chernova et al. *Cell Death & Differentiation*, 2016), although the exact molecular alterations underlying the differential anti-tumor effects to DHODHi await further investigations. In the meantime, we have provided an elaboration of these interesting findings in the revised manuscript (page 24).

6) Similarly, the mouse xenograft studies should be expanded to include additional *NF2* deficient cell lines (preferably primary PDCLs), not just following *NF2* deletion in a single line.

Thank you for this comment. We therefore performed new *in vivo* experiments using PDCLs as well as commercial cell line that are inherently *NF2*-deficient. Our data showed that DHODH inhibition significantly inhibited the tumor growth in mesothelioma cells (MESO-1) that inherently carrying *NF2* deletions. Notably, we further validated that *NF2*-deficient primary-derived xenograft (PDX) mesothelioma displayed a stronger response to DHODH inhibition compared with *NF2*-intact PDX models (new Figure 6; page 26 in the manuscript). The above *in vivo* experiments further strengthening our findings and conclusion.

7) The data claiming direct regulation of *CAD* and DHODH expression by YAP/TEAD are weak and are not supported by their examination of ChIP-SEQ data. In Figure S10 panel D, the Y axis scale for YAP binding to the *CAD* promoter in 2 cells lines ranges only from 0 to 1, as compared with 0-2555 for H3K27Ac. The same is true for TEAD binding. These axis labels are moreover only visible upon magnification to 400%.

In Panel E of the same figure, YAP and TEAD do bind the DHODH enhancer region quite well, however, their binding profiles do not overlap and are shown in different cells.

The effect of YAP depletion or inhibition on expression of *CAD* and DHODH shown in Figure 5F is marginal and inconsistent.

Thank you for these insightful comments. We have carefully reviewed the data and analyses to address the concerns raised. Below, we outline the specific clarifications and improvements made to the manuscript:

The Y-axis values in the ChIP-seq datasets are not normalized and are intended for comparison within a dataset, not between two datasets. For example, the level of YAP1 binding in the CAD promoter region is comparable across datasets but cannot be directly compared with H3K27Ac due to differing scales. We have revised the figure legend to clarify this point, ensuring that the data interpretation aligns with its intended purpose.

The apparent lack of overlap in the TEAD1 and YAP1 peaks is primarily due to the narrow 1kb window size used for visualization. When zoomed out, the peaks overlap across the entire peak region, indicating binding of both YAP and TEAD1 to the same enhancer region. This slight misalignment in summit peaks is likely due to technical variations, such as sequencing library quality. Additionally, limited public data resources for ChIP-seq in mesothelioma restricted the availability of datasets for direct overlap comparison in NCI-H2052 cells.

Inspired by the findings from the public ChIP-seq datasets, we performed additional validation experiments to strengthen the claim that YAP regulates CAD and DHODH, including the ChIP-qPCR assay (Figure 5K) and a dual-luciferase reporter assay (new Figure 5L; page 26 in the manuscript). These analyses consistently demonstrated that YAP physically binds to the predicted promoter regions of CAD and DHODH, providing compelling evidence that YAP directly regulates CAD and DHODH

Finally, our new Western blot experiments confirmed that YAP inhibition reduced the expression of CAD and DHODH (new Figure 5H; Figure S13).

We trust that these updates address the concerns raised and provide more robust and consistent evidence to support the conclusions of our study. Thank you again for your constructive feedback.

8) The Ki67 staining in Figure 11E is misleading. Ki67 is a binary signal that shows either strong positive staining or none at all, not the "graded" staining suggested for the drug-treated sgNF2 sample. It would appear that this sample was not allowed to develop for long enough and should be repeated. The data should moreover be quantified.

Thank you for pointing this out and apologize for this misleading "graded" staining data. Here, we performed new IHC staining for Ki-67 and quantified Ki-

67-positive populations, revealing that DHODH inhibition specifically led to a considerable reduction in Ki-67-positive cells in *NF2*-deficient PM tumor samples (new Figure S11E, F).

Additional points that require attention:

1) In Figure S1, the authors compare gene expression in 328 Mesothelioma samples with 65 normal tissues. The authors should clarify if all (or indeed any) of these normal tissues are pleural.

We have provided detailed information about normal tissues (pleura: 22; lung: 43) in the figure legends and Supplementary Table 1.

2) References for the datasets used in the analysis performed in Figure 1 (and Figure S1-4) should be included in the materials and methods and cited accordingly

We removed these references to the “Materials and methods” section (page 5 in the revised manuscript).

3) The Venn diagram shown in Fig 1D is confusing and could be omitted altogether. The message is already effectively represented by panel 1E.

Thank you for the insightful comment, and we apologize for any confusing information in this figure.

In this section, our research aims are to identify metabolic subtype-specific mutated genes. Thus, we first computed the fraction of samples with genomic mutations for each subtype and compared genetic alteration frequencies between every two subtypes among the three identified metabolic groups (as shown in Figure 1D). Our data revealed that *NF2* is the cluster 1-specific mutated gene with the highest genetic alteration frequency (51%) among the three subgroups. Additionally, genetic alterations in *SLC12A1*, a newly identified oncogene (Teng, Guo et al., 2016), were found to be abundant in Cluster 2 (22%), and *BAP1* was more frequently detected in Cluster 3 (60%) (Figure 1D, E, S4G; Table S3).

We completely agree that Figure 1E precisely presents our data. We still think that Figure 1D can provide a more general information about our findings. Importantly, the results further highlighted that genetic alterations might influence cancer metabolic reprogramming and drive metabolic heterogeneity in pleural mesothelioma.

Please also see the comment and our response to Reviewer #3 (Additional specific comments on individual figures).

4) P53 loss is known to have profound metabolic consequences and occurs in up to 20% or so of mesothelioma - it would be interesting to see how it aligns with the different clusters

Thank you for your comment. Our analysis revealed no significant correlation between genetic alterations in *TP53* and metabolic subgroups, as shown in the figure below. We discussed this finding in the revised manuscript (page 28) as follows: “Although *TP53* loss is known to have profound metabolic consequences and occurs in up to 20% of mesothelioma cases (Hmeljak, Sanchez-Vega et al., 2018, Liu, Zhang et al., 2019), no direct association with metabolic subgroups was identified in our study (data now shown)”.

Genetic alterations in *TP53* among different metabolic subgroups. (A) Bar chart showing the frequency of *TP53* alterations among the three metabolic subgroups. (B) Heatmap showing the genetic status of *TP53* among different metabolic subgroups in the mesothelioma patients' cohort with each bar representing one tumor sample.

5) Fig 1 panel F makes little sense. It is described as a heatmap in the legend but is not one in any sense that I am familiar with. As drawn, several of the labels are very difficult to make out. What is the significance of the different sized rectangles?

Thank you for highlighting the issues with the original Figure 1F. In response, we replaced the original visualization with a revised figure (new Figure 1F) based on the enrichment analysis of differentially expressed metabolites following *NF2* deletion in two cell lines (H2452 and H28). The inclusion of H28 in the analysis was suggested by Reviewer #3, and the updated figure consistently highlights pyrimidine metabolism as a commonly altered metabolic process following *NF2* deletion across both cell lines. This addition enhances the robustness and reproducibility of our findings.

The revised figure provides a more intuitive and accessible representation of the data, and we have updated the corresponding figure legend and text in the manuscript to reflect these improvements (page 21).

6) The authors applied different (arbitrary) cut-offs for fold change in their integrated analysis of transcriptomic and proteomic data. Justification for the choice of cut-offs should be provided.

Thank you for this comment. In this study, we were interested in the genes/proteins whose expression was differentially upregulated upon *NF2* deletion, so a log₂-fold change > 0.5 was equivalent to a fold change > 1.4 for transcriptomics, and a fold change > 1.3 was selected as the cut-off for proteomics. To clarify, we have rephased all the settings according to the fold change format in the manuscript.

Moreover, to avoid bias, we also referenced valuable articles, showing that a log₂-fold change threshold of 0.5 was used to capture meaningful changes in gene expression (Dotsenko, Tewes et al., 2024, Schinke, Shi et al., 2022), and a fold change of 1.3 was a valuable threshold for proteomics (Lanfredi, Thome et al., 2021, Zhang, Ma et al., 2018). The above justification has been added to the indicated figure legends and methods section (pages 13&14 in the revised manuscript).

Please also see the comment and our response to Reviewer #3 (specific comments on Figure 2B).

7) The Y axes of data shown in Figure S7C, D and F are all tailored to exaggerate the effect of *NF2* deletion. Please redraw these data using Zero values at the axis intercepts.

Thank you for this suggestion, and these data were redrawn using zero values at the axis intercepts to avoid exaggerating the effect (new Figure S7B, C).

8) The pleural cell line data are very difficult to discern in Figure S8H. Again, the regression line shown looks rather inflated based on the few pleural cell lines I can make out at high magnification.

Thank you for pointing this out. Figure S8H (new Figure S8N) shows the correlation between *NF2* expression and drug sensitivity to the DHODH inhibitor leflunomide in the pan-cancer cohort. Yes, we completely agree that

the figure is difficult to discern based on the 22 cancer lineages. For clarity, we have separately displayed the data from mesothelioma cell lines in an independent panel (new Figure S8O), which shows that a positive correlation between the protein levels of NF2 and leflunomide typically exists in pleural mesothelioma across pan-cancer cell lines, indicating that the above association might be tissue context specific. In addition, the original data of this part are shown in Supplementary Table 21.

9) The drug effect in the orthotopic mouse model (Figure 11D) is not significant, although claimed in the text to be "marked".

Thank you for your attention to detail. We have removed this less stringent word from the sentence (page 26 in the revised manuscript).

Referee #3 (Comments on Novelty/Model System for Author):

Identifies potential new vulnerability in a difficult cancer type. Identifies a new mechanism, but further evidence is needed to support the main claims. Some of the statistical analysis should be improved, but that is not a critical issue in this particular paper. They use a pre-clinical model, patient data are queried from available databases.

Thank you for the coherent summary and positive assessments of our study.

Referee #3 (Remarks for Author):

Title: "De Novo Pyrimidine Synthesis Is a Collateral Metabolic Vulnerability in Patients with Malignant Pleural Mesothelioma Harboring NF2 Deficiency"

General: The manuscript titled 'De Novo Pyrimidine Synthesis Is a Collateral Metabolic Vulnerability in Patients with Malignant Pleural Mesothelioma harboring NF2 Deficiency' by Xu Duo et al characterizes a subtype of malignant pleural mesothelioma (MPM), which is determined by the loss of NF2 tumor suppressor and a subsequent activation of YAP proto-oncogene, which in turn upregulates de novo pyrimidine synthesis pathway. The concept that de novo pyrimidine synthesis becomes a targetable vulnerability in certain condition is not entirely new, but the authors are the first to show this for MPM. In addition, they present a new finding that pyrimidine synthesis genes DHODH and CAD are directly transcriptionally regulated by YAP. While the authors provide an important insight into a subset of MPM and potential therapeutic targets,

additional work and evidence is needed to reinforce authors' reasoning and conclusions. The specific points are laid out below.

We thank the reviewer for your thoughtful and encouraging comments regarding our work. We appreciate the acknowledgment of the novelty of our findings, including the characterization of *NF2*-deficient MPM as a distinct subtype and the demonstration of YAP-mediated transcriptional regulation of pyrimidine synthesis genes. We also value the feedback emphasizing the need for additional evidence to reinforce our reasoning and conclusions.

Major points:

1. Authors refer extensively to the upregulation of *de novo* pyrimidine synthesis in *NF2*-deficient MPM cells, but this major aspect of the study has not been directly supported by evidence. The authors show that CAD and DHODH are upregulated (mRNA and protein level), but do not provide evidence that the *de novo* pyrimidine synthesis pathway is actually more active. Increased nucleotide levels in *NF2*-deficient cells (presented in Fig. 1) do not conclusively demonstrate an increased activity of the *de novo* pathway. Therefore, metabolic flux measurements, for example glutamine to UMP C13 tracing, must be provided to demonstrate an increased *de novo* pyrimidine pathway flux in *NF2* deficient cells, and enzymatic activity should be measured for DHODH or CAD.

We thank the reviewer for their insightful comments and, as response, have performed new experiments as suggested.

First, we measured metabolic flux using ¹⁵N-glutamine stable isotope tracing assays according to a previous study (Pal, Kaplan et al., 2022), which revealed that ¹⁵N-glutamine (substrate of the *de novo* pyrimidine synthesis pathway)-derived ¹⁵N robustly labelled the UMP pool after the deletion of *NF2*(new Figure 1H).

Second, our measurement revealed that *NF2*-depleted mesothelioma cells showed increased enzymatic activity of DHODH compared to *NF2* wild-type cells (new Figure S7E).

Taken together, we thank the reviewer for pointing out this important limitation and for the proposed experiments, which provide additional evidence that has substantiated our claims about the *de novo* pyrimidine synthesis pathway in *NF2*-deficient MPM and improved the rigor of our study.

2. Following from point 1 above: Please provide evidence, using inhibitors or KOs, that salvage does not contribute to increased nucleotide levels/increased proliferation observed in *NF2*-deficient cells. Fig S7 shows that UPP1 expression is significantly upregulated in *NF2*-deficient cells. Indeed, the statement: "Interestingly, the protein levels of uridine phosphorylase 1 (UPP1) and uridine-cytidine kinase 2 (UCK2), pivotal mediators of the pyrimidine salvage pathway, were not markedly different between the wild-type and *NF2*-deficient groups." is not supported by the figure S7F since there is a significant upregulation in UPP1 protein levels.

We thank the reviewer for highlighting this important point regarding the potential contribution of the pyrimidine salvage pathway to the increased nucleotide levels/proliferation observed in *NF2*-deficient cells.

As response, we first performed the experiment to evaluate the cell proliferation in both wild-type and *NF2* deletion mesothelioma cells under normal and low-glucose setting, a condition known to force cells to rely on the salvage pathway (Skinner, Blanco-Fernandez et al., 2023). Notably, glucose limitation did not alter the increased proliferation rate of mesothelioma cells after deletion of *NF2*, suggesting that the survival benefit of mesothelioma cells upon *NF2* deletion is independent of pyrimidine salvage pathways (Figure S9B).

Second, genetic knockdown of *UPP1* did not affect the increased cell proliferation induced by *NF2* loss (new Figure S9I-K; page 24 in the manuscript). *Interestingly, despite the observation that NF2-deficient PM cells have increased protein levels of UPP1 (Figure S7), UPP1 KD did not affect the upregulated cell survival induced by NF2 deficiency.* This is consistent with our observations shown in the original submission that suppression of the pyrimidine salvage pathway with a UCK2 inhibitor led to a similar effect in *NF2*-deficient and wild-type cells under both normal and glucose deprivation "stress" conditions (Figure S9), further suggesting that the *NF2*-dependent survival gain is not dependent on the pyrimidine salvage pathway.

In summary, we thank the reviewer for your constructive feedback. By performing these additional experiments and revising the manuscript, we have provided compelling evidence that *NF2*-driven MPM tumorigenesis explicitly relies on *de novo* pyrimidine biosynthesis for cell survival. These findings provide a more nuanced understanding of pyrimidine metabolism in *NF2*-deficient cells.

3. For the key experiments where CRISPR-induced NF2 deficient clones are used, the authors need to provide a reconstitution/addback control using sgRNA-resistant cDNA.

Thank you for this insightful suggestion. In response, we performed new experiments incorporating sgRNA-resistant cDNA. Our results demonstrated that rescued expression of sgRNA-resistant *NF2* significantly reduced CAD/DHODH expression and upregulated YAP_pS127 expression. Importantly, it also restored cell viability under treatment with a DHODH inhibitor, effectively excluding the possibility of off-target effects of sg*NF2*. These findings have been incorporated into the revised manuscript as new Figures S8 and 10 (pages 23&25 in the revised manuscript).

4. What happens with UMPS? It is not mentioned anywhere, information about its expression level in NF2-deficient cells needs to be provided.

Thank you for this comment. Our transcriptomic and proteomic analyses revealed that while the mRNA expression of UMPS, another pivotal enzyme involved in *de novo* pyrimidine synthesis, did not change, its protein levels were significantly upregulated following *NF2* deletion. We have included these results in the revised manuscript (new Figure S7B-D) and described the findings in page 22. *While UMPS mRNA levels remained stable, its protein expression showed an increase following NF2 deletion (Figure S7B-D), supporting the idea that NF2 loss promotes the de novo pyrimidine synthesis pathway.*

5. Similarly, authors never discuss/assess TAZ, even though TAZ can complement YAP. What happens to TAZ in NF2-deficient cells and how TAZ reacts to YAP manipulations? Does the inhibitor used for YAP also inhibit TAZ?

Our transcriptomic and proteomic analyses revealed no significant alterations in TAZ expression following the deletion of *NF2*. To further investigate, we performed new qRT-PCR experiments to assess TAZ transcript levels under conditions of YAP knockdown and inhibition. These experiments demonstrated that TAZ transcript levels remained unchanged after genetic knockdown of YAP using small interfering RNA and were slightly decreased following treatment with a YAP inhibitor (YAP-TEAD Inhibitor 1 (Peptide 17)). This inhibitor, which disrupts the YAP-TEAD protein-protein interaction, has potential therapeutic utility in YAP-driven cancers.

These findings suggest that TAZ expression is not significantly affected by *NF2* deficiency or direct YAP manipulations, though it may exhibit a minor response to YAP inhibition. Relevant data are now presented as below and discussed in the revised section (page 30).

Alterations in the expression of TAZ after the deletion of *NF2*. (A-B) The difference in FPKM values from RNA-seq (A) and LFQ intensities from 480 label-free proteomes (B) of TAZ between the H2452 sg*NF2*-1 and sgCtrl groups. (C-D) mRNA expression of TAZ in H2452 *NF2*-KO cells transfected with the indicated siRNAs (C) or treated with YAP-TEAD inhibitor 1 (Peptide 17) for 48 h (D). The data are presented as the mean \pm S.D. A two-tailed unpaired t test was used for comparison.

6. Please provide stronger evidence that YAP regulates DHODH and CAD expression. Functional assay, for example luciferase reporters, are required to demonstrate expression from DHODH and CAD promoters is enhanced by YAP.

Thank you for this insightful suggestion. In response, we performed a dual-luciferase reporter assay to directly evaluate the effect of YAP on the promoters of DHODH and CAD. The results demonstrated that, PM cells harboring *NF2* mutations, which are associated with increased YAP activity (new Figure S10), exhibited robust increases in luciferase signals from the predicated binding regions of both CAD and DHODH (new Figure 5L). These findings further strengthen the conclusion drawn from the ChIP-qPCR assay (Figure 5K).

These additional functional data provide strong evidence supporting YAP-mediated transcriptional regulation of DHODH and CAD.

7. The authors should provide better reasoning of why they focused on *NF2* in their manuscript. There are other oncogenes/tumor suppressor involved in the MPM etiology.

Thank you for pointing this out. In the revised manuscript, we have included a more detailed explanation for our focus on *NF2* in the context of PM etiology and its unique, understudied biology. Specifically, we have highlighted the prevalence of *NF2* alterations in PM, their specific role in tumorigenesis, and

the rationale for investigating the metabolic consequences of NF2 deficiency (page 21 in the revised manuscript).

8. The Crispr-generated NF2-deficient clones come up in the text figures without any proper introduction, making it difficult to follow the story line. They should be properly introduced, demonstrating the level of the KO on their first appearance. Reconstitution control should be provided (see point 2 above).

Thank you for this comment. In response, we have revised the manuscript to properly introduce the CRISPR-generated NF2-deficient cells upon their first mention. Specifically:

We have included a description of CRISPR-generated *NF2*-deficient cells more clearly in the Results section where they first appear (page 21).

To validate the knockout efficiency, Western blotting was performed, showing robust abrogation of NF2 protein levels. These results have been included in the revised manuscript as new Figure S5A.

To establish the functional consequences of CRISPR-mediated NF2 knockout, we confirmed that the genetic abrogation of *NF2* robustly promoted the aggressiveness of PM both *in vitro* and *in vivo*. These results are presented in Figure S5 and discussed in the revised manuscript (page 21).

To address concerns regarding off-target effects of sg*NF2*, we performed a reconstitution experiment using an sgRNA-resistant NF2 cDNA. Our results demonstrated that the rescued expression of sgRNA-resistant *NF2* significantly blunted the alterations of CAD, DHODH and YAP_pS127 observed in *NF2*-deficient cells. Importantly, it also restored cell viability under DHODH inhibitor treatment, further excluding the possibility of off-target effects of sg*NF2*. These data are presented in new Figures S8 and 10 of the revised manuscript (pages 23&25).

Together, these findings support the role of *NF2* loss in driving tumor progression and metabolic reprogramming in PM.

9. For all the presented western blots, quantifications from at least 3 independent experiments should be provided, as some of the blots are quite unconvincing.

We have performed quantifications from at least 3 independent experiments for all presented western blots. These data are now included in the revised manuscript as new Supplementary Figure 13, along with detailed explanations in the figure legend. We trust that these quantifications provide greater clarity and robustness to our findings.

10. The curve fitting presented in this manuscript is inappropriate (see figures 3C, 4E, 4F, S8E, S9A, S9C-E.). Please redo these curve fits, consider using a wider concentration range of inhibitors.

Thank you for this insightful suggestion. In response, we have revised the curve fitting to improve its accuracy and appropriately represent the data. Specifically:

We redid the “best fit” regression lines using log(inhibitor) vs. response-variable slopes (four parameters) to better cover the data points shown.

In addition, for new functional experiments involving patient-derived cell lines (PDCLs), we used a wider concentration range of inhibitors, which clearly revealed that *NF2*-deficient cells were more sensitive to DHODH inhibitors than were *NF2*-intact cells (new Figure 3I).

We have also updated the corresponding figure legends to reflect the changes in analysis and methodology.

These revisions provide a more accurate representation of the data and strengthen the conclusions regarding the differential sensitivity of *NF2*-deficient cells to DHODH inhibitors.

11. Single cell RNAseq data are discussed in the Discussion section but are not included in the manuscript: "Intriguingly, our ongoing single-cell RNA sequencing (scRNA-seq) study revealed that malignant mesothelioma cells exhibit robust enrichment of nucleotide metabolism-related genes, among which DHODH is expressed at lower levels in immune cells, such as T and natural killer (NK) cells, further indicating that targeting DHODH might be an ideal therapeutic approach for precisely regulating nucleotide synthesis in cancer cells without easily damaging the tumor immune microenvironment (TIME). Indeed, the metabolic interplay between cancer cells and immune cells and how this crosstalk impacts immune surveillance and antitumour immunity in MPM deserve further investigation." To allow assessment of these conclusions, please include these results and methods in the manuscript, and

make the raw data available at a dedicated public repository. Alternatively, the authors are asked to remove this part of discussion.

Thank you for pointing this out. As the single-cell RNA sequencing (scRNA-seq) data are part of an ongoing study that is not yet "in press", we have decided to remove these sentences from the Discussion section of the manuscript. This ensures the focus remains on the presented data and avoids drawing conclusions based on results that are not yet publicly available. We will consider including these findings in a future publication once the study has been completed and the data have been thoroughly validated.

12. "Based on the TCGAMESO dataset, MPM patients with genetic inactivation of NF2 had an extremely poor prognosis (Figure S5A, B)." - the difference on S5B is not significant, the statement is thus not true. Please correct.

Thank you for raising this point. We have revised the statement to more accurately reflect the data, as follows: *"Importantly, PM patients with genetic inactivation of NF2 showed an overall poor prognosis, although the difference did not reach statistical significance with a p value of 0.078."*

In addition, to strengthen this analysis, we performed a new survival analysis based on our internal PM cohort. The results demonstrated that PM patients with low NF2 levels are significantly associated with a poor prognosis, reinforcing the clinical relevance of NF2. These findings are now presented in Figure S5I and discussed in the revised manuscript (page 21).

These updates ensure accuracy and provide additional supporting data for the prognostic impact of NF2 inactivation in PM patients.

Minor points:

13. Graphical abstract: lower right part refers to clinical analysis, implying work with patients. In fact, only existing clinical data were evaluated. Please correct.

The graphical abstract has been updated to avoid unclear information.

14. Superiority statements and statements about 'significance' - unless supported by a proper statistical analysis - must be avoided. Generally, only the statistically significant results are discussed, therefore, to emphasize this further is confusing.

Thank you for this comment. We have examined the full text thoroughly to avoid statistically significant statements.

15. "Notably we found that both the transcript and protein levels of CAD and DHODH, the critical rate-limiting enzymes involved in the de novo pyrimidine biosynthesis pathway, were significantly elevated in NF2-deficient MPM cells compared to those in wild18 type group"- These enzymes are undoubtedly critical, but the question is whether they are also rate-limiting. In the author's model, no evidence of that is provided. Please rephrase.

Thank you for pointing this out. We have rephrased the above sentences to avoid overstating the rate-limiting nature of these enzymes in our model, as follows: *"Importantly, CAD and DHODH, the classical rate-limiting enzymes for de novo pyrimidine biosynthesis, were significantly increased at both transcript and protein levels in NF2-deficient PM cells."*

To support this statement, we have added a new reference (Wang, Yang et al., 2019) to the revised manuscript (page 22). This citation establishes the widely accepted role of CAD and DHODH as rate-limiting enzymes in the *de novo* pyrimidine biosynthesis pathway, while acknowledging that our findings do not independently confirm this in the current model.

16. Please, unify the general style of the manuscript. Sometimes spaces are used and other times they are missing.

a. i.e. Figure 1.E "...MPM patients (n = 87)" vs Figure 1.D "...TCGA-MPM cohort (n=87)".

b. "p <0.05 was considered to indicate statistical significance".

The presented graphs mostly use p values, but other times stars are used to show significance (S5F, S7E, Figure 6A).

There are spaces between citation numbers and words.

Please keep the lettering of the figures from left to right row-wise (for example, figure 2H is mixed up).

Thank you for your attention to detail. We apologize for the above errors, and the full text as well as the figure layout have now been examined/modified to avoid the above errors.

17. The meaning of this statement is unclear: "NF2 status defines a unique subset of MPM characterized by abnormal de novo pyrimidine synthesis." Please indicate what abnormal means (upregulated/downregulated).

Thank you for pointing out the ambiguity in this statement. We have rephrased it for clarity as follows: *“NF2 deficiency defines a distinct subtype of PM characterized by elevated de novo pyrimidine synthesis.”* This revised statement more accurately describes the observed metabolic phenotype in NF2-deficient PM cells.

18. It is unclear what: "pyrimidine metabolism is the primary metabolic subtype of MPM" means. Based on the results, it is not the only metabolic subtype.

We apologize for the unclear statement. To address this, we have rephrased the sentence for clarity as follows: *“In this study, through an extensive multi-omics analysis, we reveal for the first time that upregulated pyrimidine metabolism defines a distinct metabolic subtype of PM.”* This revision acknowledges that pyrimidine metabolism is a key but not exclusive metabolic subtype in PM, providing a more accurate reflection of the results (page 5).

19. Data retrieval and preprocessing - please state which DepMap datasets you used for the analysis. In the methods section "Comparison of genetic alteration frequencies among subtypes", please specify which datasets were used.

We now provide the above information in the “Materials and methods” section (pages 6 and 7).

In addition, the transcriptomic (Expression Public 23Q4) and metabolomic (Metabolomics) datasets of the PM cell lines used in this study were downloaded from the Cancer Dependency Map Data Portal ([https://depmap.org/portal/download/custom/\(Tsherniak, Vazquez et al., 2017\)](https://depmap.org/portal/download/custom/(Tsherniak, Vazquez et al., 2017))).

Genetic alteration data, encompassing copy number variations and mutations, were sourced from the TCGA-MESO series via the UCSC Xena online data portal (<https://xenabrowser.net/datapages/>).

20. "MPM tumor tissues showed an apparent enrichment in a variety of metabolic processes compared to normal tissues, suggesting metabolic adaptation or reprogramming during tumorigenesis". This statement does not describe what the figures show. Figure S1 actually shows that there are more downregulated metabolic genes than upregulated, and Figure S2 also shows that more pathways are upregulated in the normal tissue than in the tumor.

Thank you for pointing this out. We agree that tumor metabolism is dysregulated, with some metabolic genes and processes being upregulated and others being downregulated in PM compared with normal tissues. To better align the text with the figures, we have rephrased the statement in the revised manuscript (page 19) as follows: *“Our results revealed a distinct pattern of metabolic gene alterations in PM, with 16 genes exhibiting increased expression and 30 genes showing decreased expression relative to normal tissues. Beyond the canonical molecular changes associated with PM, the tumors displayed enrichment in various metabolic processes (Figure S1A, B; Table S9, 10). Gene set enrichment analysis (GSEA) further identified 6 metabolic pathways that were enriched in tumors, alongside 13 pathways enriched in normal tissues. Notably, pyrimidine metabolism emerged as the top-ranked enriched metabolic pathway in PM, while fatty acid degradation was more prevalent in normal tissues (Figure S2A; Table S11).”*

This revised description accurately reflects the findings shown in Figures S1 and S2 and emphasizes the key observation of pyrimidine metabolism as a top-enriched pathway in PM tumors.

21. Reference 32 does not say anything about NF2.

Thank you for this comment and we apologize for this mistake. This reference has been deleted.

22. "Unlike the research hotspots in the field of cancer cell metabolism, pyrimidine biosynthesis is a less appreciated aspect; however, it is essential to produce large amounts of cellular building blocks to meet the needs of cell proliferation and has been recently documented to conquer ferroptosis for cell survival 25, 42." - The statement is overblown. Please cite PMID 30449682 which shows that pyrimidine synthesis is essential for tumor growth. What do you mean by "conquer ferroptosis"? Please rephrase.

Thank you for these insightful comments. We have revised the statement to better reflect the findings in the cited studies and incorporated the suggested reference (PMID 30449682). The rephrased statement now reads as follows (page 28): *“Pyrimidine biosynthesis is essential for tumor growth as it supports the production of large amounts of cellular building blocks required for cell proliferation (Bajzikova, Kovarova et al., 2019, Wang et al., 2021, Yang, Zhao et al., 2023). Intriguingly, a recent study revealed that DHODH-mediated pyrimidine synthesis plays a role in suppressing mitochondrial lipid peroxidation*

and ferroptosis, suggesting it may constitute to tumor survival through an additional defense mechanism (Mao, Liu et al., 2021).”.

This revision avoids overstatement, provides accurate context for the role of pyrimidine biosynthesis in tumor biology, and includes the suggested citation to strengthen the statement.

Additional specific comments on individual figures (beyond the general ones discussed above):

Figure 1:

Figure 1A - the information from the legend should be in the methods. 1A also represents different datasets. How were these datasets integrated? Please describe it in methods. Does each bar represent one patient? Explain this in the legend.

We have moved this information to the “Materials and Methods” section, and a detailed description has been incorporated to explain the integrated method (page 5): *“To mitigate batch effects, we utilized the "ComBat" function from the "sva" package, employing an empirical Bayesian framework to adjust and eliminate batch-related biases prior to merging the transcriptomic datasets (Leek, Johnson et al., 2012).”.*

Yes, each bar represents one patient. The figure legends have been updated with the above information. *Heatmap showing the gene set variation analysis (GSVA) enrichment scores for the three metabolic pathway-based clustering subgroups derived from the transcriptomic data of PM tumors, with each bar representing an individual patient sample.*

Figure 1G - labels are missing in the top 3 graphs.

We have added these labels (new Figure 1G).

The statement: "In an effort to identify the oncogenic signalling crosstalk with pyrimidine metabolism, we observed that NF2 loss of function (mutations and/or homozygous deletions) was more frequent in Cluster 1 (73%) than in the other metabolic subgroups (27%) (Figure 1D, E; Table S15)" is not well supported by the figures. Please provide better visualization.

Thank you for highlighting this issue. To address this concern, we have revised and clarified this section to better align with the data and improve its visualization. Specifically:

Our analysis aimed to identify metabolic subtype-specific mutated genes across the three metabolic subgroups identified in the study.

We computed the fraction of samples with genomic mutations for each subtype and compared genetic alteration frequencies between every two subtypes. As shown in Figure 1D), our data revealed:

- *NF2* is the most frequently mutated gene in cluster 1, with a genetic alteration frequency of 51%.
- *SLC12A1*, a newly identified oncogene, is predominantly altered in Cluster 2 (22%).
- *BAP1* mutations are most abundant in Cluster 3 (60%) (Figure 1D, E, S4G; Table S15).

To provide more clear information, we have provided another new panel in Figure 1E, which presents a more detailed and visually intuitive comparison of genetic alteration frequencies across the three subgroups.

Accordingly, we have revised the manuscript to align with these visual improvements and clarified the description of the findings. The updated text now reads: *“To explore the interplay between cancer genetics and tumor metabolism, we analyzed genetic alteration frequencies among the three metabolic subtypes using mutation and copy number variant (CNV) data from TCGA-MESO cohort. Our findings indicated that NF2 mutations/deletions were prevalent in Cluster 1, with the highest frequency of genetic alterations; SLC12A1, was frequently altered in Cluster 2, and BAP1 mutations were most common in Cluster 3 (Figure 1D, E, S4G; Table S3).”*.

We believe these updates address the concern and improve the clarity and visual representation of the findings (page 21 in the revised manuscript).

Figure 1E - how were the patients stratified into the metabolic subgroups since this dataset is presumably different from the one used in Figure 1A? If the number of mutations would be relative to the size of the cluster (number of patients) would cluster 1 still have the most *NF2* mutations? All major mutated genes in mesothelioma should be analyzed.

Thank you for this insightful comment, and we apologize for the unclear information in the text of our original submission. Below, we clarify the methods used for patient stratification, address the cluster-specific mutation frequencies, and elaborate on the comprehensive analysis of major mutated genes in mesothelioma:

- The data shown in Figure 1A were generated from publicly available transcriptomic datasets of mesothelioma, including the Cancer Genome Atlas (TCGA) MESO, the Gene Expression Omnibus GEO; GSE12345, GSE163720, GSE42977, and GSE51024) and the European Bioinformatics Institute (EMBL-EBI; E-MTAB-1109).
- Consensus clustering of the transcriptomic data was performed to identify the optimal number of stable metabolism-related subpopulations of PM patients. These subgroups were defined based on shared metabolic profiles and formed the basis for stratification in the study.
- The data shown in Figure 1E, however, were obtained specifically from TCGA-MESO cohort, which includes high-quality whole exome sequencing. This cohort provided the necessary genomic information to investigate genetic alterations across metabolic subtypes. Similar analytic strategy has been shown in this study (Blum, Meiller et al., 2019), which dissected the molecular features of MPM based on a combined cohort and then relied on the TCGA series to map the genetic alterations.
- To ensure consistency between the transcriptomic clustering used in Figure 1A and the genomic analysis in Figure 1E, we performed a distribution analysis comparing the stratified metabolic subtypes from individual datasets to the whole cohort. This analysis revealed that the stratified subtypes displayed a similar distribution pattern across datasets, supporting the robustness of the clustering approach (Figure S4B).
- Especially among TCGA-MESO cohort, cluster 1 still comprises the majority patients as 46%, almost the same compare with the whole cohort (47%). We acknowledge the importance of considering the size of each cluster when interpreting mutation frequencies. In cluster 1 from TCGA series, *NF2* mutations constituted 51% of the subgroup, which is the highest frequency among all clusters. This observation underscores the specificity of *NF2* mutations as a defining feature of cluster 1.

To address the broader context of mesothelioma genetics, we analysed all major mutated genes identified in mesothelioma patients from the TCGA-

MESO dataset. The frequencies of genetic alterations that showed significant differences among metabolic subtypes were highlighted in Figure 1D and E. The raw data for these analyses are provided in Table S3, ensuring transparency and reproducibility of the results.

We have clarified these points in the revised manuscript, ensuring that the relationship between transcriptomic and genomic data is clearly explained (page 20 and 21). The new description also highlights the inclusion of all major mutated genes in mesothelioma.

We hope this clarification resolves the concerns and provides a more comprehensive understanding of the methodology and results.

Legend Figure 1E: "Genetic status of NF2, BAP1, and SLC12A1 among different metabolic subgroups in the TCGA cohort of MPM patients (n = 87). The data were downloaded from the (<https://www.cbioportal.org/>)."
Which data was downloaded? Are these the same datasets as those indicated in methods - "Comparison of genetic alteration frequencies among subtypes" from xenabrowser?

The genetic mutation data were downloaded. The datasets are the same. To maintain consistency, we updated the figure legend as follows: "*The genetic mutation data were downloaded from the online data portal UCSC Xena*".

Figure 1D - it is unclear what the Venn diagrams contain and represent. Please use better visualization. Why did the authors focus only on NF2 if SLC12A1 and BAP1 are also significant?

Thank you for the insightful comment, and we apologize for any confusion caused by the presentation of Figure 1D. Below, we clarify the content of the Venn diagrams, improve their interpretation, and justify our focus on NF2 in this study:

- The Venn diagrams in Figure 1D illustrate metabolic subtype-specific mutated genes. These diagrams highlight the overlap and uniqueness of mutated genes across the three identified metabolic subtypes.
- Our primary research goal was to identify metabolic subtype-specific mutated genes. Through our analysis of the TCGA-MESO dataset, we found that: *NF2* is the most frequently mutated gene in Cluster 1 (51%), distinguishing this subgroup from the others; *SLC12A1*, a

newly identified oncogene (Teng et al., 2016), is predominantly altered in Cluster 2 (22%); *BAP1*, a well-known tumor suppressor in mesothelioma, is most frequently mutated in Cluster 3 (60%) (Figure 1D, E, S4G; Table S3).

- These findings suggest that genetic alterations influence cancer metabolic reprogramming and drive the metabolic heterogeneity observed in pleural mesothelioma.

We chose to focus on NF2 for the following reasons:

- Cluster 1 Composition and Prognosis: Cluster 1 represents the largest patient subgroup and is associated with the poorest prognosis among the three clusters.
- Clinical Gap: While *BAP1* is an important gene in mesothelioma biology, NF2 remains relatively underexplored, particularly in the context of its role in metabolic reprogramming.
- Therapeutic Relevance: Effective treatment strategies targeting *NF2*-deficient mesothelioma are lacking, and our study aims to address this gap by investigating the metabolic vulnerabilities associated with NF2 loss.

We have revised the manuscript to provide a concise explanation of why NF2 was chosen as the focus of this study. This is now elaborated on page 21.

Figure 1F is difficult to interpret. A heat map of individual genes would be more appropriate. Please indicate the number of biological and technical replicates used to generate this data. Are the results consistent in other cell lines (H28)? Which H2452 sgNF2 cells were used (in the rest of the manuscript, there are two different sgRNAs)?

Thank you for these valuable comments. We have addressed your concerns as follows:

We performed new untargeted metabolic profiling in H28 sgCtrl and sgNF2-1 cells, which consistently demonstrated that *NF2* deletion induced robust enrichment in pyrimidine metabolism (new Figure S6B). Importantly, the results in Figure 1F indicate that pyrimidine metabolism was the only consistently altered metabolic process in two independent *NF2*-deficient PM cell lines (H2452 and H28). This consistency across cell lines strengthens the conclusions of our study.

For the untargeted metabolite profiling experiments shown in Figure 1F, the following replicates were used:

Point-to-point responses (EMM-2024-19672)

- Six biological replicates for H2452 cells
- Three biological replicates for H28 cells.

For the experiments shown in Figure 1F, H2452 sgNF2-1 and H28 sgNF2-1 were used. The sgRNA sequences have been provided in Table S6.

These updates ensure that the data are clearly presented, reproducible, and consistent across models. The revisions to Figure 1F and its legend, along with the inclusion of results from additional cell lines, strengthen the reliability and robustness of our findings.

Figure 1G - Are there any other significantly different metabolites (unrelated to pyrimidine synthesis)?

The untargeted metabolic profiling shown in Figure 1F revealed that pyrimidine metabolism was the only commonly altered metabolic process shared between H2452 and H28 upon the deletion of NF2.

To further explore metabolite changes, we performed targeted metabolomic analysis to focus on specific metabolites of interest (Roberts, Souza et al., 2012). This analysis further revealed that NF2 depletion in H2452 cells led to a robust accumulation of metabolites involved in *de novo* pyrimidine synthesis, including UMP, dCMP, dTMP, uracil, and UDP-GlcNAc (Figure 1G, S6E; Table S18). In addition to pyrimidine-related metabolites, we identified other significantly altered metabolites among the 68 energy-related metabolites analysed. For instance, guanosine, inosine, ATP and NAPDH, also differed, as shown in Figure S6E. We have included a discussion of this finding in the manuscript to highlight additional metabolic changes observed in NF2-depleted cells (page 29): *“Although we observed an increased abundance of purine nucleotides including guanosine and inosine, following the deletion of NF2. Our results indicate that NF2-deficient PM cells exhibit a phenotype resistant to purine biosynthesis blockade, suggesting that the observed increase in purine nucleotides may be a secondary effect resulting from enhanced pyrimidine biosynthesis.”*

These updates ensure that the analysis captures broader metabolic alterations while emphasizing the primary focus on pyrimidine metabolism.

Figure 2:

Figure 2B should be improved to better visualize these important results. Which genes from proteomics and transcriptomics were the most significantly different?

Why are absolute values of the fold changes being compared? This way, the information whether the genes were upregulated or downregulated is lost. Also, how did the authors set the thresholds? 0.5 and 1.3 seems rather arbitrary. Also see the major point 8 above.

Thank you for these insightful suggestions. We have addressed these points in the revised manuscript as follows:

Improved Visualization and Interpretation: In addition to the general summary presented in Figure 2B, we have provided new data in Figure S7A to provide detailed information about the 92 common upregulated candidates identified through the combined transcriptomic and proteomic analysis. The top candidates identified in the transcriptome were PCK2, MME and MGLL, while PM20D2, MME and WWC1 were most prominent in the proteome (Figure 2B; S7A; Table S19, 20).

Clarification of Thresholds: Considering that loss of *NF2* can induce tumor-specific dependencies on other hyperactive genes/proteins within the tumor, making these candidates attractive synthetic lethal therapeutic targets, so in this study we focused on the above upregulated candidates after *NF2* deletion. Regarding the thresholds, we acknowledge the need for clear justification. A \log_2 -fold change >0.5 (equivalent to a fold change >1.4) was selected for transcriptomic data, while a fold change >1.3 was used for proteomic data. A \log_2 -fold change >0.5 was selected for transcriptomic data, while a fold change >1.3 was used for proteomic data. These thresholds were chosen based on prior studies demonstrating their utility in identifying meaningful changes (Dotsenko et al., 2024, Lanfredi et al., 2021, Schinke et al., 2022, Zhang et al., 2018). We have provided these references and a detailed explanation in the manuscript (page 22) to justify our approach. Please also see the comment and our response to Reviewer #2 (Additional points 6).

These updates enhance the clarity and robustness of Figure 2B and the accompanying analysis while providing stronger justifications for our methodological choices.

Remove unnecessary pathways in Figure 2C and S1B (i.e. African trypanosomiasis, pertusis). Please explain what is the "gene ratio" on x axis.

We have removed these unnecessary pathways in figures. Moreover, the gene ratio represents the number of dysregulated genes relative to the total number of annotated genes within the specified pathway. We have included this information in the corresponding figure legends.

Figure 2G - To refer to these results from 3 cell lines as results from a 'panel' is inappropriate. 3 cell lines do not constitute a panel. Please rephrase.

This has been rephrased (page 23 in the manuscript).

Figure 3

Figure 3A - statistical comparison is done between two different groups, t-test is thus not appropriate. Please correct. Figure 3A proliferation assay is not included in the methods. Time course would be preferable.

Thank you for your attention to detail. Here, two-way ANOVA with multiple comparisons was used for comparisons. Moreover, the proliferation assay has been incorporated into the “Materials and methods” section (page 8 in the revised manuscript). *Moreover, to determine cell proliferation, PM cells transfected with control or CAD/DHODH siRNAs were seeded in triplicate in 24-well plates at a density of 1×10^4 cells per well. After the indicated time points, the cells were trypsinized and counted via an automated cell counter.*

We performed new experiments to incorporate 72h(day3) and 120h(day5) and the results showed that genetic silencing of either *CAD* or *DHODH* via small interfering RNA (siRNA) impaired the viability of PM cells in a time-dependent manner, notably resulting in apparent growth inhibition in *NF2*-deletion PM cells (new Figure 3A, B; page 23).

Figure 3A, B - the silencing efficacy in these experiments should be documented on the protein level.

We performed western blotting to validate the silencing efficacy (new Figures S8A, B; S13).

Figure 4

Figure 4A - uridine concentrations used are very high. Can supplementation with physiological concentrations of uridine rescue the phenotype? Please include control for 0 uM of DHODHi and for a *Nf2* WT cell line - preferentially in H2452 shctrl.

Thank you for these insightful suggestions. In response, we performed new experiments using physiological concentrations of uridine (50µM) and a slightly elevated concentration (100µM, twofold greater than physiological). Both concentrations effectively rescued the phenotype, demonstrating that uridine

supplementation at physiological levels is sufficient to counteract the effects of DHODH inhibition.

Additionally, we included the following controls to strengthen the analysis: the vehicle control (0uM DHODHi) and H2452 sgCtrl (NF2 wild-type cell line). These results are now incorporated into new Figure 4A and discussed on page 24 of the revised manuscript.

Figure 4B Is the DNA damage and apoptosis upon DHODH inhibition recapitulated in other cell lines (H28, H2052) upon NF2 KO? This would allow to make more general conclusions about MPM.

Thank you for this comment. Here, we performed new experiments in which both *NF2*-deficient PDCLs and H28 cells with CRISPR-mediated *NF2* deletion were treated with DHODH inhibitors, followed by the detection of DNA damage and apoptosis (new Figure 4D-G; page 24). Our data showed that DHODH inhibition resulted in substantial DNA damage and apoptotic cell death in *NF2*-deficient but not in wild-type PM cells, further supporting our findings.

Figure 4EF (and also SF9) Different cell lines are used, please show how H2452 WT compares to H2452 NF2 KO in one graph, not to H2052 in separate graphs. Curves do not fit the data as indicated above in the major points.

Thank you for highlighting these points. In this study, we focused on the effects of the combination of cisplatin and DHODH inhibitor, and our aim was to evaluate whether there is a synergistic effect between these two drugs. Our findings are clinically relevant, and we performed experiments in the H2052 cell line, which is already deficient in *NF2*, not just following genetic manipulation. Here, we focused on the synergy score and showed that the combination of cisplatin and brequinar had a robust synergistic effect on the PM cell line harboring *NF2* mutations (mean synergy score: 13.15) but had a weaker effect (mean synergy score: 1.86) on the wild-type group. Generally, a score from -10 to 10 indicates that the interaction between two drugs is likely additive, but a score over 10 indicates that the interaction between two drugs is likely synergistic. As the combination index is calculated based on the independent cell line, it is optimal to show that in a separate graph. Moreover, we repeated this experiment in H2452 wild-type and *NF2*-KO cell lines, which produced similar results, as shown in the new Supplementary Figure 12E, F. Additionally, to improve the logic of this study, we moved these data to Figure S12 (page 27).

To address the concern about curve fitting, we reanalysed the data using log(inhibitor) vs. response-variable slopes (four parameters logistic regression) to better represent the data points. These improved fits are now reflected in the revised figures.

Figure 4H is missing x axis labels.

This has been added.

Figure 4G, H - please indicate what is the baseline for the viability. The y axis label is not descriptive enough.

Thank you for your attention to detail. This should be Figure 5G, H (now is new Figure 5I, J), right? The y axis represents normalization to the independent vehicle group without treatment with the DHODH inhibitor. We have modified the y axis as “Cell viability/set untreated group with DHODHi as 1”.

Figure 5

Figure 5I: Why was the chip experiment done only in wt cells? If the author's hypothesis is true, the chip signal should be stronger in NF2 KO cells. Control genes, not only CAD and DHODH, should be included in this experiment.

Thank you for this important observation. In response to your comment, we have extended our experiments and incorporated the following updates:

We performed additional CHIP–qPCR experiment in H2052 cells, which harbor an *NF2* mutation (p.Arg341Ter) and exhibit high YAP activity. The results revealed a significant positive signal in the predicated binding regions of CAD and DHODH when compared to the IgG control group. These findings are consistent with the hypothesis that CAD and DHODH are transcriptional targets for the YAP1 complex, as supported by CHIP-Seq data.

To further investigate whether NF2 loss enhances YAP-mediated regulation of these genes, we used a dual-luciferase reporter assay. This assay revealed that the luciferase signal for the CAD and DHODH predicted binding regions was indeed stronger in *NF2*-mutant cells (H2052, high YAP activity) than in *NF2* wild-type cells (H2452, low YAP expression (Figure 5L). These results support the hypothesis that NF2 loss amplifies YAP transcriptional activity.

In addition to CAD and DHODH, we incorporated control genes in the ChIP–qPCR experiments, including both positive *CYR61*, a well-established YAP target gene, which showed a significantly increased ChIP signal in the presence of YAP1, and negative (*ACTB*) genes, which showed no significant enrichment in the ChIP signal. These controls validate the specificity of the ChIP experiment and are now included as part of the revised data in new Figure 5K.

We trust these additional data and updates will address your concerns and provide a more robust validation of our findings.

In the related supplementary figure S10C: While this statement is true: "Interestingly, further examination of the TCGA cohort of MPM, we found that the mRNA level of YAP is positively correlated with CAD but negative correlation with enzymes of the pyrimidine salvage pathway (UPP1, CDA, and UCK2) (Figure S10C). ", there was no correlation with DHODH and a negative correlation with UMPS, two important enzymes of the de novo pyrimidine synthesis pathway. How do the authors explain this discrepancy?

Thank you for this insightful comment. We agree that the observed data based on patient samples exhibit some inconsistencies, which we have carefully considered. Below, we address these points in the setting of previous findings and our own results present in this study:

The lack of significant correlation between YAP1 and DHODH mRNA levels, as well as the negative correlation with UMPS, highlights a known limitation of relying solely on transcriptomic data. There can often be discrepancies between transcript levels and enzymatic activity due to post-transcriptional regulation, protein turnover, or context-specific cellular states.

Despite the lack of correlation at the mRNA level, our experimental results provide strong multilayer evidence that YAP1 directly regulates CAD and DHODH at the transcriptional level, as demonstrated by the ChIP–qPCR and luciferase reporter assays (Figure 5).

While we did not find evidence supporting a correlation between YAP1 and UMPS, this result aligns with our findings and was not claimed in the manuscript. UMPS may not be a primary target of YAP-mediated transcriptional regulation.

The significant negative correlation between YAP1 and enzymes of the pyrimidine salvage pathway (UPP1, CDA, and UCK2) is intriguing. This suggests a potential reprogramming of pyrimidine metabolism in YAP1-active

tumors, favouring the *de novo* synthesis pathway over salvage. This interesting observation merits further investigation in future studies.

We appreciate the opportunity to clarify these points and have incorporated the necessary explanations and interpretations into revised manuscript (page 25).

Figure 6

In the related text:

"Importantly, the DHODH inhibitor brequinar obviously retarded tumor growth in NF2-deficient group without apparent influence on wild-type group and markedly prolonged the survival of mice bearing orthotopically transplanted MPM tumors (Figure 6A-E; Figure S11D)." The difference in survivability between sgNF2 and sgNf2+DHODHi (S11D) is not statistically significant. Please correct the statement.

Thank you for pointing this out. We have corrected the statement as follows (page 26): *"This intervention resulted in a prolonged survival benefit in mice bearing orthotopically transplanted NF2-deficient PM tumors."*

Supplementary figures:

Supplementary Figure 3A - the authors should include in the methods which genes were included in a "Gene signature of pyrimidine metabolism" and how it was calculated. Please make larger spaces between the individual cancer types for clarity. The tumors, where data from normal tissue are not included, should be removed, as that is confusing and lacks reporting value.

Thank you for your constructive suggestions. We have incorporated the details into the "Materials and Methods" section as follows (page 6): *"Gene set variation analysis (GSVA) algorithm was used to calculate the gene set enrichment score for each metabolic pathway based on the transcriptomic data, which is a widely used state-of-the-art gene set projection method that reflects the activity level of biological processes (Hanzelmann, Castelo et al., 2013)."* To ensure rigor, we have modified the term "gene signature of pyrimidine metabolism" to "GSVA enrichment score of pyrimidine metabolism" in Figure S3A.

For clarity, we provided a revised Figure 3B by separately showing the comparisons of independent cancer types with normal samples. Additionally, we aimed to show the expression patterns of pyrimidine metabolism scores in tumor samples across various cancer cohorts, including some that lack normal controls, as shown in Figure S3C. Therefore, we would like to retain the original Figure S3A.

Supplementary Figure 3B - please explain what "mean value of pyrimidine metabolic signatures" means.

In this context, "mean value of pyrimidine metabolic signatures" refers to a numeric quantity representing the average of a group of numbers within the patient cohort, positioned between the highest and lowest values in that set.

Figure S4 In the related text: „we consistently showed that differentially expressed metabolites in human MPM cell lines belonging to Cluster 1 were strongly enriched in pyrimidine biosynthesis (Figure S4E, F; Table S13,14)." - the highlighted cell lines (blue), that belong to cluster 1, are actually not clustering. Please correct the statement and/or the conclusion from this data.

Thank you for this this important observation. We acknowledge that some of the cell lines originally highlighted as belonging to Cluster 1 did not cluster well, and we have reanalysed the data to address this issue.

We repeated the metabolic enrichment analysis, restricting the analysis cell lines that were exactly clustered into Cluster 1 (n=4). This revised analysis confirmed that differentially expressed metabolites in these cell lines were highly enriched in pyrimidine biosynthesis (Figure S4E, F), further supporting our findings while ensuring a consistent and accurate representation of the data.

We have also rephrased the related text to provide a clear and accurate message (page 20). The updated statement now reads: *“Analysis of metabolomics data from the Cancer Cell Line Encyclopaedia (CCLE) further corroborated that the differentially expressed metabolites in PM cell lines corresponding to Cluster 1 were predominantly enriched in pyrimidine metabolism.”*.

Supplementary Figure 4A is never referenced in the text, please remove.

Thank you for pointing this out, and we have modified the text to describe that figure (page 20 in the manuscript): *“Our analysis of global metabolic gene expression shifts revealed substantial variability in metabolic dysregulation among PM tumors, as evidenced by a greater distance in metabolic gene expression within tumors compared to normal tissues (Figure S4A).”*.

Supplementary Figure 4F - there are two scales for the p value (shade and x axis), please use only one.

We have modified this figure to remove the redundant p value.

Supplementary Figure 5D - y axis depicts cell number, not proliferation rate, y axis title is unclear, please indicate what is being measured.

Here, the cell viability was measured using the CCK-8 assay. The y-axis has been modified and shown in the revised Figure S5B.

Supplementary Figure 6B - according to the legend, log₂FC is represented by both color and size of the dots. Please correct.

This figure (now is new Figure S6E) has been modified.

Supplementary Figure 6C - please clarify the y axis label, the protein expression needs to be relative to a reference.

The data from the above figure were generated based on the proteomic dataset from the Cancer Dependency Map Data Portal (<https://depmap.org/portal/download/custom/>). The different analytical steps used for data acquisition and processing are briefly summarized below, and detailed information can be found in the corresponding publication (Goncalves, Poulos et al., 2022). *The mass spectrometry data provide relative rather than absolute protein quantitation. The Z score was calculated with reference to each protein as measured across the entire cell line panel. We have provided the above information in the related Figure legend.*

References

- Bajzikova M, Kovarova J, Coelho AR, Boukalova S, Oh S, Rohlenova K, Svec D, Hubackova S, Endaya B, Judasova K, Bezawork-Geleta A, Kluckova K, Chatre L, Zobalova R, Novakova A, Vanova K, Ezrova Z, Maghzal GJ, Magalhaes Novais S, Olsinova M et al. (2019) Reactivation of Dihydroorotate Dehydrogenase-Driven Pyrimidine Biosynthesis Restores Tumor Growth of Respiration-Deficient Cancer Cells. *Cell Metab* 29: 399-416 e10
- Blum Y, Meiller C, Quetel L, Elarouci N, Ayadi M, Tashtanbaeva D, Armenoult L, Montagne F, Tranchant R, Renier A, de Koning L, Copin MC, Hofman P, Hofman V, Porte H, Le Pimpec-Barthes F, Zucman-Rossi J, Jaurand MC, de Reynies A, Jean D (2019) Dissecting heterogeneity in malignant pleural mesothelioma through histo-molecular gradients for clinical applications. *Nat Commun* 10: 1333
- Dotsenko V, Tewes B, Hils M, Pasternack R, Isola J, Taavela J, Popp A, Sarin J, Huhtala H, Hiltunen P, Zimmermann T, Mohrbacher R, Greinwald R, Lundin KEA, Schuppan D, Maki M, Viiri K, Investigators CEC (2024) Transcriptomic analysis of intestine following administration of a transglutaminase 2 inhibitor to prevent gluten-induced intestinal damage in celiac disease. *Nat Immunol* 25: 1218-1230
- Goncalves E, Poulos RC, Cai Z, Barthorpe S, Manda SS, Lucas N, Beck A, Bucio-Noble D, Dausmann M, Hall C, Hecker M, Koh J, Lightfoot H, Mahboob S, Mali I, Morris J, Richardson L, Seneviratne AJ, Shepherd R, Sykes E et al. (2022) Pan-cancer proteomic map of 949 human cell lines. *Cancer Cell* 40: 835-849 e8
- Hanzelmann S, Castelo R, Guinney J (2013) GSVA: gene set variation analysis for microarray and RNA-seq data. *BMC Bioinformatics* 14: 7
- Hmeljak J, Sanchez-Vega F, Hoadley KA, Shih J, Stewart C, Heiman D, Tarpey P, Danilova L, Drill E, Gibb EA, Bowlby R, Kanchi R, Osmanbeyoglu HU, Sekido Y, Takeshita J, Newton Y, Graim K, Gupta M, Gay CM, Diao L et al. (2018) Integrative Molecular Characterization of Malignant Pleural Mesothelioma. *Cancer Discov* 8: 1548-1565
- Lanfredi GP, Thome CH, Ferreira GA, Silvestrini VC, Masson AP, Vargas AP, Grassi ML, Poersch A, Candido Dos Reis FJ, Faca VM (2021) Analysis of ovarian cancer cell secretome during epithelial to mesenchymal transition reveals a protein signature associated with advanced stages of ovarian tumors. *Biochim Biophys Acta Proteins Proteom* 1869: 140623
- Leek JT, Johnson WE, Parker HS, Jaffe AE, Storey JD (2012) The sva package for removing batch effects and other unwanted variation in high-throughput experiments. *Bioinformatics* 28: 882-3
- Liu J, Zhang C, Hu W, Feng Z (2019) Tumor suppressor p53 and metabolism. *J Mol Cell Biol* 11: 284-292
- Mao C, Liu X, Zhang Y, Lei G, Yan Y, Lee H, Koppula P, Wu S, Zhuang L, Fang B, Poyurovsky MV, Olszewski K, Gan B (2021) DHODH-mediated ferroptosis defence is a targetable vulnerability in cancer. *Nature* 593: 586-590
- Ohta Y, Shridhar V, Bright RK, Kalemkerian GP, Du W, Carbone M, Watanabe Y, Pass HI (1999) VEGF and VEGF type C play an important role in angiogenesis and lymphangiogenesis in human malignant mesothelioma tumours. *Br J Cancer* 81: 54-61

Ostergaard L, Tietze A, Nielsen T, Drasbek KR, Mouridsen K, Jespersen SN, Horsman MR (2013) The relationship between tumor blood flow, angiogenesis, tumor hypoxia, and aerobic glycolysis. *Cancer Res* 73: 5618-24

Pal S, Kaplan JP, Nguyen H, Stopka SA, Savani MR, Regan MS, Nguyen QD, Jones KL, Moreau LA, Peng J, Dipiazza MG, Perciaccante AJ, Zhu X, Hunsel BR, Liu KX, Alexandrescu S, Drissi R, Filbin MG, McBrayer SK, Agar NYR et al. (2022) A druggable addiction to de novo pyrimidine biosynthesis in diffuse midline glioma. *Cancer Cell* 40: 957-972 e10

Roberts LD, Souza AL, Gerszten RE, Clish CB (2012) Targeted metabolomics. *Curr Protoc Mol Biol* Chapter 30: Unit 30 2 1-24

Schinke H, Shi E, Lin Z, Quadt T, Kranz G, Zhou J, Wang H, Hess J, Heuer S, Belka C, Zitzelsberger H, Schumacher U, Genduso S, Riecken K, Gao Y, Wu Z, Reichel CA, Walz C, Canis M, Unger K et al. (2022) A transcriptomic map of EGFR-induced epithelial-to-mesenchymal transition identifies prognostic and therapeutic targets for head and neck cancer. *Mol Cancer* 21: 178

Skinner OS, Blanco-Fernandez J, Goodman RP, Kawakami A, Shen H, Kemeny LV, Joesch-Cohen L, Rees MG, Roth JA, Fisher DE, Mootha VK, Jourdain AA (2023) Salvage of ribose from uridine or RNA supports glycolysis in nutrient-limited conditions. *Nat Metab* 5: 765-776

Teng F, Guo M, Liu F, Wang C, Dong J, Zhang L, Zou Y, Chen R, Sun K, Fu H, Fu Z, Guo W, Ding G (2016) Treatment with an SLC12A1 antagonist inhibits tumorigenesis in a subset of hepatocellular carcinomas. *Oncotarget* 7: 53571-53582

Tsherniak A, Vazquez F, Montgomery PG, Weir BA, Kryukov G, Cowley GS, Gill S, Harrington WF, Pantel S, Krill-Burger JM, Meyers RM, Ali L, Goodale A, Lee Y, Jiang G, Hsiao J, Gerath WFJ, Howell S, Merkel E, Ghandi M et al. (2017) Defining a Cancer Dependency Map. *Cell* 170: 564-576 e16

Wang W, Cui J, Ma H, Lu W, Huang J (2021) Targeting Pyrimidine Metabolism in the Era of Precision Cancer Medicine. *Front Oncol* 11: 684961

Wang X, Yang K, Wu Q, Kim LJY, Morton AR, Gimple RC, Prager BC, Shi Y, Zhou W, Bhargava S, Zhu Z, Jiang L, Tao W, Qiu Z, Zhao L, Zhang G, Li X, Agnihotri S, Mischel PS, Mack SC et al. (2019) Targeting pyrimidine synthesis accentuates molecular therapy response in glioblastoma stem cells. *Sci Transl Med* 11

Xu D, Gao Y, Yang H, Spils M, Marti TM, Losmanova T, Su M, Wang W, Zhou Q, Dorn P, Shu Y, Peng RW (2024) BAP1 Deficiency Inflames the Tumor Immune Microenvironment and Is a Candidate Biomarker for Immunotherapy Response in Malignant Pleural Mesothelioma. *JTO Clin Res Rep* 5: 100672

Yang C, Zhao Y, Wang L, Guo Z, Ma L, Yang R, Wu Y, Li X, Niu J, Chu Q, Fu Y, Li B (2023) De novo pyrimidine biosynthetic complexes support cancer cell proliferation and ferroptosis defence. *Nat Cell Biol* 25: 836-847

Zhang Q, Ma C, Gearing M, Wang PG, Chin LS, Li L (2018) Integrated proteomics and network analysis identifies protein hubs and network alterations in Alzheimer's disease. *Acta Neuropathol Commun* 6: 19

11th Apr 2025

Dear Prof. Shu,

Thank you for submitting your manuscript to EMBO Molecular Medicine and please accept my apologies for the delay in getting back to you as one reviewer needed more time to complete their report. We have now received feedback from the three reviewers who reviewed your initial submission. As you can see from the reports below, the referees are overall satisfied with the revisions. However, reviewers #1 and #3 still have some minor concerns. We would therefore like to invite you to submit further revisions of your manuscript to address any remaining issues.

When submitting your revised manuscript, please carefully read the following instructions. We carry out an initial quality check on all revised manuscripts before they are resubmitted; failure to include the requested items will delay the evaluation of your revision.

We require:

1/ A .docx formatted version of the manuscript text (including legends for main figures, EV figures and tables). Please make sure that the changes are highlighted to be clearly visible.

2/ Please note that emails bounced for contributing author Wenqing Mou (wenqing.mou@whu.edu.cn), please adjust.

3/ Please remove 'data not shown' (p.29 and p.30). All data discussed in the manuscript should be presented in the main or EV figures.

4/ Funding should be merged with Acknowledgements. Please note that the information provided in the manuscript should match the information provided in the submission system (currently #82173012 is in the manuscript text but not in our system; #2021M701497 is in our system but not in the manuscript text).

5/ The references should be listed alphabetically, with 10 names before et al.

6/ The list of suppl. data should be removed from the manuscript text.

7/ The Methods section should be after the Discussion.

8/ Individual production quality figure files as .eps, .tif, .jpg (one file per figure). For guidance, download the 'Figure Guide PDF' (<https://www.embopress.org/page/journal/17574684/authorguide#figureformat>).

Please note: We replaced Supplementary Information with Expanded View (EV) Figures and Tables that are collapsible/expandable online. EV Figures should be cited as 'Figure EV1, Figure EV2' etc... in the text and their respective legends should be included in the main text after the legends of regular figures.

9/ Please ensure that all suppl. figures are called out in sequential order.

10/ All supplementary tables should be uploaded as separate files; Suppl. Tables 2, 3, 9, 10, 11, 12, 13, 14, 15, 17 should be renamed Dataset EV1 - Dataset EV 10; Suppl. Tables 1, 4, 5, 6, 7, 8, 16, 18, 21 should be renamed Table EV - Table EV9. Each corresponding legend should be in the dataset and table excel files and the list removed from the manuscript text.

11/ Please address the queries from our copy editors:

1. Please note that the exact p values are not provided in the legends of figures 2I, L; 3A, E, G; 4A, G; 6A, C, D, H

2. Please indicate the statistical test used for data analysis in the legend of figure 5A

3. Although 'n' is provided, please describe the nature of entity for 'n' in the legends of figures 1G, H; 3A, B, E, G; 4A, B, C, E, F, G; 5F, G, I, J, K, L

4. Please note that the error bars are not defined in the legends of figures 6E, F, G

12/ At EMBO Press we ask authors to provide source data for the main figures. Our source data coordinator will contact you to discuss which figure panels we would need source data for and will also provide you with helpful tips on how to upload and organize the files.

13/ Please provide a .docx formatted letter INCLUDING the reviewers' reports and your detailed point-by-point responses to their comments. As part of the EMBO Press transparent editorial process, the point-by-point response is part of the Review Process File (RPF), which will be published alongside your paper.

14/ A complete author checklist, which you can download from our author guidelines (<https://www.embopress.org/page/journal/17574684/authorguide#submissionofrevisions>). Please insert information in the checklist that is also reflected in the manuscript. The completed author checklist will also be part of the RPF.

15/ All Materials and Methods need to be described in the main text using our 'Structured Methods' format. According to this format, the Methods section includes a Reagents and Tools Table (listing key reagents, experimental models, software and relevant equipment and including their sources and relevant identifiers) followed by a Methods and Protocols section describing the methods, ideally using a step-by-step protocol format. The aim is to facilitate adoption of the methodologies across labs. Please download and fill our Reagents and Tools Table template (.docx), which you can find in our author guidelines: <https://www.embopress.org/page/journal/14693178/authorguide#structuredmethods>.

The tables with primers can be removed from the manuscript text and added to the reagents and tools table.

16/ It is mandatory to include a 'Data Availability' section after the Materials and Methods. Before submitting your revision, primary datasets produced in this study need to be deposited in an appropriate public database, and the accession numbers and database listed under 'Data Availability'. Please remember to provide a reviewer password if the datasets are not yet public (see <https://www.embopress.org/page/journal/17574684/authorguide#dataavailability>).

17/ The paper explained: EMBO Molecular Medicine articles are accompanied by a summary of the articles to emphasize the major findings in the paper and their medical implications for the non-specialist reader. Please provide a draft summary of your article highlighting

18/ Author contributions: CRedit has replaced the traditional author contributions section because it offers a systematic machine readable author contributions format that allows for more effective research assessment. Please remove the Authors Contributions from the manuscript and use the free text boxes beneath each contributing author's name in our system to add specific details on the author's contribution. More information is available in our guide to authors.

19/ Every published paper now includes a 'Synopsis' to further enhance discoverability. Synopses are displayed on the journal webpage and are freely accessible to all readers. They include a short stand first (maximum of 300 characters, including space) as well as 2-5 one-sentences bullet points that summarizes the paper. Please write the bullet points to summarize the key NEW findings. They should be designed to be complementary to the abstract - i.e. not repeat the same text. We encourage inclusion of key acronyms and quantitative information (maximum of 30 words / bullet point). Please use the passive voice. Please attach these in a separate file or send them by email, we will incorporate them accordingly.

Please also suggest a visual abstract to illustrate your article as a PNG file 550 px wide x 300-600 px high. A cropped portion of this image will serve as thumbnail for the table of content on our webpage.

20/ As part of the EMBO Publications transparent editorial process initiative (see our Editorial at <http://embomolmed.embopress.org/content/2/9/329>), EMBO Molecular Medicine will publish online a Review Process File (RPF) to accompany accepted manuscripts.

This file will be published in conjunction with your paper and will include the anonymous referee reports, your point-by-point response and all pertinent correspondence relating to the manuscript. Let us know whether you agree with the publication of the RPF and as here, if you want to remove or not any figures from it prior to publication.

I look forward to receiving your revised manuscript.

Yours sincerely,

Lise Roth

***** Reviewer's comments *****

Referee #1 (Comments on Novelty/Model System for Author):

The study is robust and multiple model systems are used.

Referee #1 (Remarks for Author):

In this revised manuscript by Duo Xu et al. entitled "De Novo Pyrimidine Synthesis Is a Collateral Metabolic Vulnerability in NF2-deficient Mesothelioma", the authors use multiple datasets and model systems to determine that NF2 loss in a proportion of MPM activates YAP to promote pyrimidine synthesis. The data are novel and robust and previous reviewer comments are adequately addressed. The only weakness of this manuscript is that human data from the on-site cohort are limited and largely the primary findings rely on published datasets.

To this end, Marazioti et al. described recently in this Journal that a subset of MPM display KRAS mutations/gain/activation (EMBO Mol Med. 2022 Feb 7;14(2):e13631. doi: 10.15252/emmm.202013631). This appears to be the same or a similar subset of patients with the one described in the present manuscript. Therefore, the authors should look for KRAS pathway activation in their data, especially in patients in cluster 1 without NF2 mutations.

Referee #2 (Comments on Novelty/Model System for Author):

The technical limitations of the study are specifically and appropriately addressed in the revised manuscript

Referee #2 (Remarks for Author):

The authors have done an admirable job of addressing all of my concerns raised at first review and I gladly congratulate them on their important and excellent study.

Referee #3 (Comments on Novelty/Model System for Author):

It somewhat unclear how some of the analysis have been done, so this should be described better.

Referee #3 (Remarks for Author):

The authors diligently address all the major points satisfactorily. However, there are a few additional issues that should be resolved to improve clarity/accuracy.

1. The inclusion of glutamine tracing is appreciated, and this provides a very strong support for increased de novo synthesis. However, please indicate which tracer was used (position of the ^{15}N label, manufacturer, catalogue number), gradient conditions for separation and the machine used for analysis.
2. Luciferase reporter assay: please indicate the precise sequences of promoter fragment used in these experiments in the methods or supplementary data.
3. Improved graphical abstract could not be found.
4. CAD+ DHODH being rate limiting for de novo synthesis - the paper they are citing does not show that. It would be better to just say these enzymes are 'critical' for the pathway to avoid this issue altogether.
5. Physiological uridine concentration is less than 50 μM in both plasma and TIF. IN fact, uracyl is close to 50 in TIF. It is thus

not appropriate to claim that 50 μ M uridine is a physiological concentration. However, as the point of Figure 4a is to show that uridine rescues at reasonable concentrations, they should remove the word 'physiological' to avoid misunderstandings.

6. Claims about significance not supported by data should be removed from the paper, not just for the instances pointed out by this reviewer, which were just given as examples. For example, Fig. 3I has no statistics despite the significance statement in the main text (Fig. 3J is not labeled BTW). This general issue should be corrected throughout.

7. Figure 1E: this new figure should be better stacked by cluster (not by gene) and indicate percentage of mutations in that cluster to correct for the cluster size

8. Page 28 Figure 1F

The analysis performed is unclear and it is not apparent how the authors arrived at their conclusion that pyrimidine biosynthesis is the only common upregulated pathway.

"We performed new untargeted metabolic profiling in H28 sgCtrl and sgNF2-1 cells, which consistently demonstrated that NF2 deletion induced robust enrichment in pyrimidine metabolism (new Figure S6B)."

The Figure 1F is presumably based on Figure S6A, B - I could not find how the enrichment analysis was performed, what 'Impact' on the x axis stands for (why does y axis use $-\ln p$ values instead of \log_{10} - minor point). Importantly, the legend for 1F states:

"The enrichment analysis was conducted using differentially expressed metabolites identified between NF2knockout (sgNF2-1) group and the scrambled control (sgCtrl) group, with a p value <0.05 and a variable importance in projection (VIP) >1 "

However, there is no VIP on S6B, nor do any of the values reach the threshold (if it is equivalent to Impact). Concerningly, it seems that Arginine and proline metabolism would be an overlap as well. The number of replicates provided in this response should be stated in the methods/legends.

9. Page 29 Figure 2

Criteria for selecting thresholds should be also mentioned in the manuscript.

9. Page 36 Figure S4

"We repeated the metabolic enrichment analysis, restricting the analysis cell lines that were exactly clustered into Cluster 1 ($n=4$). This revised analysis confirmed that differentially expressed metabolites in these cell lines were highly enriched in pyrimidine biosynthesis (Figure S4E, F), further supporting our findings while ensuring a consistent and accurate representation of the data."

In the text (page 20 of the manuscript):

"Analysis of metabolomics data from the Cancer Cell Line Encyclopaedia (CCLE)(Jiang, Song et al., 2023) further corroborated that the differentially expressed metabolites in PM cell lines corresponding to Cluster 1 were predominantly enriched in pyrimidine metabolism (Figure S4E, F; Table S14, 15)."

The blue cell lines still do not cluster into cluster 1, in fact they do not cluster at all, so why are they assigning those cell lines into cluster 1? They should be in cluster 2/3. This makes us concerned about how this clustering was performed. Why are the other 3 blue cell lines that do in fact cluster with cluster 1 excluded? This whole analysis is either very strange, or the authors are unable to get their point across.

10. Page 36 (Supplementary Figure 4F...) and page 37 (Supplementary figure 6B) - they are missing explanation of shading (S4F) and size of dots (S6B). Thresholds specified in the legend are higher and not what they claimed before:

"S4F, KEGG enrichment analysis of differentially expressed metabolites with a p value < 0.25 and fold change > 0.25 in PM cell lines classified as Cluster 1 ($n=4$) compared to other metabolic subtypes."

11. Page 37 (supplementary Figure 5D) - please indicate in figure legend, which assay was used to assess measure the viability - two different assay are described in methods.

12. The fits have been improved, but the fit for ctrl sg in Fig. 3C is linear and not actually representing the data points.

Below are our point-by-point responses, with the reviewers' suggestions and comments in black and our responses in blue, and quoted text from the manuscript *in purple and italics*.

To the copy editors:

Please note that the exact p values are not provided in the legends of figures 2I, L; 3A, E, G; 4A, G; 6A, C, D, H.

Thank you for this feedback. We have double-checked the data. The p-value was calculated using GraphPad, which provides an exact p-value but does not display values lower than 0.0001. We would like to confirm if this is acceptable.

Reviewer access details for the "Data Availability" section

GEO accession GSE295651:

Go to <https://www.ncbi.nlm.nih.gov/geo/query/acc.cgi?acc=GSE295651>; Enter token ynihwacwjvyfjwr into the box

ProteomeXchange accession PXD063342: Project accession: PXD063342; Token: ydPSmleVlxql

MetaboLights accession MTBLS12461:

Go <https://www.ebi.ac.uk/metabolights/reviewer9635a3e0-eda3-426e-82d3-b685ca72cf26> to

Referee #1 (Comments on Novelty/Model System for Author):

The study is robust and multiple model systems are used.

We sincerely thank the reviewer for recognizing the rigor of our study and the breadth of model systems we employed.

Referee #1 (Remarks for Author):

In this revised manuscript by Duo Xu et al. entitled "De Novo Pyrimidine Synthesis Is a Collateral Metabolic Vulnerability in NF2-deficient Mesothelioma", the authors use multiple datasets and model systems to determine that NF2 loss in a proportion of MPM activates YAP to promote

pyrimidine synthesis. The data are novel and robust and previous reviewer comments are adequately addressed.

We thank the reviewer for your clear and concise summary of our revised manuscript and for emphasizing the novelty and robustness of our findings. We also greatly appreciate the insightful earlier comments, which have significantly enhanced the quality of our work.

The only weakness of this manuscript is that human data from the on-site cohort are limited and largely the primary findings rely on published datasets. To this end, Marazioti et al. described recently in this Journal that a subset of MPM display *KRAS* mutations/gain/activation (EMBO Mol Med. 2022 Feb 7;14(2): e13631. doi: 10.15252/emmm.202013631). This appears to be the same or a similar subset of patients with the one described in the present manuscript. Therefore, the authors should look for *KRAS* pathway activation in their data, especially in patients in cluster 1 without *NF2* mutations.

Thank you for highlighting the limitations associated with the use of public datasets and for referring to the previous study that used the same TCGA cohort of MPM patients as we did in the present study.

The study published in EMBO Molecular Medicine (2022) indicates that *KRAS* alterations promote tumorigenesis in a subset of mesothelioma patients and displayed a non-significant repulsion with *NF2* mutations. We conducted a further evaluation of *KRAS* activation and its connection with pyrimidine metabolism activity within the same cohort. Our analysis revealed that the frequency of *KRAS* alterations is 27% in Cluster 1, 16% in Cluster 2, and 12% in Cluster 3 (as shown in panel A), suggesting that *KRAS* activation is enriched in Cluster 1. Notably, within Cluster 1, six patients with *KRAS* mutations were *NF2* wild-type, while five patients showed co-occurring mutations of both *NF2* and *KRAS* (as shown in panel B), indicating that *KRAS* and *NF2* mutations are not mutually exclusive.

Moreover, we analysed and compared pyrimidine metabolism activities across various genetic backgrounds in this MPM cohort. Our findings indicated that there were no notable differences in pyrimidine metabolism gene scores between patients with *KRAS* mutations who had wild-type *NF2* compared to those with *NF2* deficiencies. A similar result was found among patients with *NF2* deficiencies who had *KRAS* mutations and those with wild-type *KRAS* (as shown in panel C). This results from the patients is in line with our finding in the previous revision that knockdown of *KRAS* has no apparent effect on the *NF2*-driven increase of the expression of CAD and DHODH (as shown in panel

D and E).

Taken together, these results further strengthen our conclusion that the pyrimidine synthesis phenotype observed in *NF2*-mut MPM cells is independent of *KRAS* mutations. We have addressed this point in our updated manuscript, combined the figures mentioned into Appendix Figure 1, and correctly referenced the study on page 15.

A recent study indicated that *KRAS* alterations may promote tumorigenesis in a subset of mesothelioma patients and displayed a repulsive tendency with *NF2* mutations (Marazioti, Krontira et al., 2022). Our analysis revealed that *KRAS* and *NF2* mutations can coexist in a small group of patients in Cluster 1, representing 26% of those with *NF2* mutations. Importantly, we observed no significant differences in pyrimidine metabolism between patients with *NF2* deficiencies who had *KRAS* mutations and those with wild-type *KRAS*. This aligns with our finding that *KRAS* knockdown does not affect the *NF2*-driven increase in *CAD* and *DHODH* expression, suggesting the alterations in pyrimidine metabolism associated with *NF2* deficiency occur independently of *KRAS* signaling in mesothelioma (Appendix Fig S1). Given that *NF2* is a well-established inhibitor of RAS, and that RAS signaling has been shown to promote pyrimidine synthesis (Wang et al., 2021), future studies are warranted to determine whether the interplay between *NF2* and *KRAS* in regulating pyrimidine metabolism is context-dependent.

(A) The genetic alterations of *NF2* and *KRAS* among different metabolic subgroups in the TCGA-MESO cohort (n=87). The genetic status of *KRAS* is defined based on the study (Marazioti et al., 2022). (B) The count of patients exhibiting *KRAS* mutations or wild-type within Cluster 1, categorized by the absence (left panel) or presence (right panel) of *NF2* Point-to-point responses (EMM-2024-21983-V2)

alterations. (C) The difference of pyrimidine metabolism GSVA scores in the indicated groups. One-way ANOVA with multiple comparisons. (D, E) mRNA of KRAS, CAD, and DHODH in the indicated groups transfected with nontargeting control or siRNA targeting KRAS for 72h. The data are presented as the mean \pm S.D. (n=3). Two-way ANOVA with multiple comparisons.

Referee #2 (Comments on Novelty/Model System for Author):

The technical limitations of the study are specifically and appropriately addressed in the revised manuscript.

We thank the reviewer for acknowledging that we have specifically and appropriately addressed the technical limitations in the revised manuscript.

Referee #2 (Remarks for Author):

The authors have done an admirable job of addressing all of my concerns raised at first review and I gladly congratulate them on their important and excellent study.

We thank the reviewer for your generous appraisal of our revision and for acknowledging the care with which we addressed all concerns. We appreciate your encouragement and are pleased that you find our study both important and well presented.

Referee #3 (Comments on Novelty/Model System for Author):

It somewhat unclear how some of the analysis have been done, so this should be described better.

Thank you for noting that additional methodological detail was needed. In response, we have substantially expanded the Methods section and updated all relevant figure legends. We believe these additions render our analytical workflow fully transparent and thank you for helping us improve clarity and rigor.

Referee #3 (Remarks for Author):

The authors diligently address all the major points satisfactorily. However, there are a few additional issues that should be resolved to improve clarity/accuracy.

We thank the reviewer for recognizing the thoroughness of our prior revisions and for identifying these additional opportunities to enhance clarity and

accuracy. We have carefully considered each suggestion and revised the manuscript accordingly to address these points.

1. The inclusion of glutamine tracing is appreciated, and this provides a very strong support for increased de novo synthesis. However, please indicate which tracer was used (position of the ^{15}N label, manufacturer, catalogue number), gradient conditions for separation and the machine used for analysis.

We thank the reviewer for this comment.

In this experiment, L-Glutamine- ^{15}N (Catalog No. HY-N0390S; Medchem Express) was used as a tracer and attached please find the position of ^{15}N label.

The gradient condition for separation is as follows: “The column oven temperature was maintained at 25°C, and autosampler was set at 10 °C. The injection volume was 5 μL . Flow rate was 0.4 mL min⁻¹. Mobile phase A was a mixture of acetonitrile and water in a volume ratio of 1:9, containing 20 mM ammonium acetate, and mobile phase B was acetonitrile: water (9:1, v/v) with 20 mM ammonium acetate. The following linear gradient was used: 0 – 2.0 min with 95% B, 2.0 – 12.0 min with 95% – 50% B, 12.0 – 13.0 min with 50% – 50% B, 13.0 – 13.1 min with 50% – 95% B, 13.1 – 16.0 min with 95%-95% B.” Additionally, LC-MS analysis was performed on an Agilent 1290 II UPLC coupled to Sciex 5600+ quadrupole-TOF MS”.

We have now incorporated the detailed information into the “Reagent and Resources” table and “Materials and methods” section (page 20 and 27 in the revised manuscript).

2. Luciferase reporter assay: please indicate the precise sequences of promoter fragment used in these experiments in the methods or supplementary data.

We have incorporated this information into the Table EV8.

3. Improved graphical abstract could not be found.

We apologize for this oversight and have submitted it as a Cover Art/Synopsis Image.

4. CAD+ DHODH being rate limiting for de novo synthesis - the paper they are citing does not show that. It would be better to just say these enzymes are 'critical' for the pathway to avoid this issue altogether.

We thank the reviewer for this valuable comment. We have changed the term “rate limiting” to “critical” in the revised manuscript (page 5 and 8).

5. Physiological uridine concentration is less than 50 uM in both plasma and TIF. IN fact, uracyl is close to 50 in TIF. It is thus not appropriate to claim that 50 uM uridine is a physiological concentration. However, as the point of Figure 4a is to show that uridine rescues at reasonable concentrations, they should remove the word 'physiological' to avoid misunderstandings.

Thank the reviewer for pointing out this. We have removed the word “physiological” from the original sentences (page 10 in the revised manuscript).

6. Claims about significance not supported by data should be removed from the paper, not just for the instances pointed out by this reviewer, which were just given as examples. For example, Fig. 3I has no statistics despite the significance statement in the main text (Fig. 3J is not labeled BTW). This general issue should be corrected throughout.

We appreciate your feedback. We have carefully reviewed the entire text to ensure that there are no statements claiming statistical significance without proper statistical evidence.

7. Figure 1E: this new figure should be better stacked by cluster (not by gene) and indicate percentage of mutations in that cluster to correct for the cluster size.

Thank you for this comment. We have updated the figure for improved clarity.

8. Page 28 Figure 1F

The analysis performed is unclear and it is not apparent how the authors arrived at their conclusion that pyrimidine biosynthesis is the only common upregulated pathway.

"We performed new untargeted metabolic profiling in H28 sgCtrl and sgNF2-1 cells, which consistently demonstrated that NF2 deletion induced robust enrichment in pyrimidine metabolism (new Figure S6B)."

The Figure 1F is presumably based on Figure S6A, B - I could not find how the enrichment analysis was performed, what 'Impact' on the x axis stands for (why does y axis use $-\ln p$ values instead of \log_{10} - minor point). Importantly, the legend for 1F states: "The enrichment analysis was conducted using differentially expressed metabolites identified between NF2knockout (sgNF2-1) group and the scrambled control (sgCtrl) group, with a p value <0.05 and a variable importance in projection (VIP) >1 ". However, there is no VIP on S6B, nor do any of the values reach the threshold (if it is equivalent to Impact). Concerningly, it seems that Arginine and proline metabolism would be an overlap as well. The number of replicates provided in this response should be stated in the methods/legends.

Thank you for highlighting the need for greater clarity around our pathway-level analysis. In the revised manuscript (Methods and Figure legends, page 27, 39, 48), we have added a comprehensive description of our enrichment workflow, which is summarized below:

1. Differential metabolite selection

Metabolites differing between *NF2*-knockout (sg*NF2-1*) and scrambled control (sgCtrl) were called at $p < 0.05$ and $VIP > 1$.

2. Metabolite set enrichment analysis (MSEA)

We submitted the list of differentially expressed metabolites (obtained above) to MetaboAnalyst(<https://www.metaboanalyst.ca/docs/Format.xhtml>) for MSEA. In this analysis, the 'impact' is a numerical value that reflects how central or important the altered metabolites are in the context of a particular metabolic pathway. This value is based on network topology, which evaluates the position of each metabolite within the pathway. The more "central" a perturbed metabolite is (e.g., a hub or bottleneck), the higher the impact score for that pathway. Additionally, we have updated the Y-axis to display $-\log_{10}(p\text{-value})$ for consistency and readability.

3. Treatment of "Arginine and proline metabolism"

We have double-checked the 'Arginine and proline metabolism' in the original data. Although this pathway is significant in H28 sg*NF2-1* ($p < 0.05$), it does

not reach significance in H2452 sg*NF2-1* ($p = 0.16$). To prevent confusion, we have now updated Figure S6A and B by only labelling pathways meeting $p < 0.05$ in each cell line (6 pathways in H2452; 10 in H28).

4. Identification of the common pathway

After these updates, pyrimidine metabolism remains the only pathway significantly enriched in both H2452 and H28 sg*NF2-1* cells. This exclusivity is now clearly depicted in Figure 1F and its legend.

We trust these revisions make our analytical approach transparent and substantiate why pyrimidine biosynthesis uniquely emerges as a shared metabolic vulnerability in *NF2*-deficient mesothelioma.

9. Page 29 Figure 2

Criteria for selecting thresholds should be also mentioned in the manuscript.

We have incorporated this information into the Methods and Figure legends in the revised manuscript (page 25, 26, 39 and 48).

9. Page 36 Figure S4

"We repeated the metabolic enrichment analysis, restricting the analysis cell lines that were exactly clustered into Cluster 1 ($n=4$). This revised analysis confirmed that differentially expressed metabolites in these cell lines were highly enriched in pyrimidine biosynthesis (Figure S4E, F), further supporting our findings while ensuring a consistent and accurate representation of the data."

In the text (page 20 of the manuscript):

"Analysis of metabolomics data from the Cancer Cell Line Encyclopaedia (CCLE)(Jiang, Song et al., 2023) further corroborated that the differentially expressed metabolites in PM cell lines corresponding to Cluster 1 were predominantly enriched in pyrimidine metabolism (Figure S4E, F; Table S14, 15)."

The blue cell lines still do not cluster into cluster 1, in fact they do not cluster at all, so why are they assigning those cell lines into cluster 1? They should be in cluster 2/3. This makes us concerned about how this clustering was performed. Why are the other 3 blue cell lines that do in fact cluster with cluster 1 excluded? This whole analysis is either very strange, or the authors are unable to get their point across.

Thank you for raising this important point, and we apologize for the confusion in our previous description. We have clarified our approach in the Methods (page 19) and streamlined the Results (page 6) as follows:

1. Clustering methodology

To classify mesothelioma cell lines by our tumor-derived metabolic subtypes, we used the R package pamr as follows. First, we merged expression data from our tumor cohort (Figure 1A; n=328) with CCLE mesothelioma cell lines (n=19) and normalized each dataset independently. Next, we applied GSVA to compute enrichment scores for every metabolic pathway in each sample. Finally, we trained a nearest shrunken centroids classifier on the tumor GSVA profiles using pamr.train and assigned each cell line to a subtype with pamr.predict, consistent with the approach of Gong et al. (2021).

2. Explanation of clustering shifts between the clustering in cell lines and tumor samples

In principle, cell-line clustering should mirror the tumor-derived subtypes. However, it is recognized that long-term culture introduces metabolic shifts that cause some lines to diverge from their original profiles (Chernova, Sun et al., 2016). Indeed, Gong *et al.* (2021; Figure S6A) have observed similar deviations between breast cancer cell lines and patient samples. To maintain consistency and accuracy, we therefore restricted our downstream analyses to only those CCLE lines whose pathway-based classification exactly matched the corresponding tumor cluster.

Figure S6

3. Rationale for excluding three “blue” lines

For downstream metabolic enrichment, we therefore retained only those cell lines that were unambiguously assigned to Cluster 1. Of these, four lines also have CCLE metabolomic profiles available, and so our follow-up analyses were limited to those four Cluster 1 cell lines.

We trust that these revisions provide greater clarity regarding our clustering strategy, the observed shifts, and the rationale behind our cell line selection. While we acknowledge the limitations of this analysis, we have additional robust evidence—including metabolomics profiling and metabolic flux analysis—that supports our conclusions. However, we remain open to omitting the cell line data if the reviewer believes it would strengthen the overall focus of the manuscript.

***Page 6 Result:** We further stratified mesothelioma cell lines from the Cancer Cell Line Encyclopaedia (CCLE)(Jiang, Song et al., 2023) into three metabolic subgroups using the nearest shrunken centroids method (Gong, Ji et al., 2021, Tibshirani, Hastie et al., 2002). Our analysis revealed that metabolites differentially expressed in PM cell lines associated with Cluster 1-corroborated by publicly available metabolomics datasets-were predominantly enriched in pyrimidine metabolism (Figure EV4E, F; Dataset EV6, 7).*

***Page 19 Methods:** Furthermore, we extended the metabolic pathway-based subtypes to mesothelioma cell lines utilizing the R package "pamr." The expression data from the tumor samples (n=328) were integrated with the mesothelioma cell line data from the Cancer Cell Line Encyclopedia (CCLE; n=19) and subsequently normalized within each dataset. Prior to the implementation of the "pamr.train" and "pamr.predict" functions, we computed the enrichment score for each metabolic pathway in each sample using the GSVA method. Subsequently, the cell lines were categorized into the metabolic pathway-based subtypes of the tumor samples based on the outcomes of the "pamr" predictive analysis. Transcriptomic (Expression Public 23Q4) and metabolomic (Metabolomics) datasets for the PM cell lines used in this research were obtained from the Cancer Dependency Map Data Portal (Tsherniak, Vazquez et al., 2017).*

10. Page 36 (Supplementary Figure 4F...) and page 37 (Supplementary figure 6B) - they are missing explanation of shading (S4F) and size of dots (S6B). Thresholds specified in the legend are higher and not what they claimed before:

"S4F, KEGG enrichment analysis of differentially expressed metabolites with a

pvalue < 0.25 and fold change > 0.25 in PM cell lines classified as Cluster 1 (n=4) compared to other metabolic subtypes."

Thank you for your feedback.

In our initial submission, the shading of S4F indicated the p-value, while the X-axis also represented the p-value. We have now revised the figure to remove the shading and keep the dots in the same colour to prevent any confusion.

In S6A and B, the size of the dots indicates the impact of the enriched metabolic pathway (please see the comments above).

Furthermore, we reviewed the Figure legends about Figure S4F and verified that we did assert the same thresholds in our initial submission.

11. Page 37 (supplementary Figure 5D) - please indicate in figure legend, which assay was used to assess measure the viability - two different assay are described in methods.

Thank you for your comment. Figure S5D presents the quantified results of the clonogenic assay depicted in Figure S5C (page 47 in the revised manuscript).

12. The fits have been improved, but the fit for ctrl sg in Fig. 3C is linear and not actually representing the data points

Thank the reviewer for this comment. We have re-checked the data and generated the best-fit curve using GraphPad Prism [log(inhibitor) vs. response-variable slope (four parameters)], which more accurately represents the data points presented.

References

- Chernova T, Sun XM, Powley IR, Galavotti S, Grosso S, Murphy FA, Miles GJ, Cresswell L, Antonov AV, Bennett J et al. (2016) Molecular profiling reveals primary mesothelioma cell lines recapitulate human disease. *Cell Death Differ* 23: 1152-64
- Gong Y, Ji P, Yang YS, Xie S, Yu TJ, Xiao Y, Jin ML, Ma D, Guo LW, Pei YC et al. (2021) Metabolic-Pathway-Based Subtyping of Triple-Negative Breast Cancer Reveals Potential Therapeutic Targets. *Cell Metab* 33: 51-64 e9

Jiang M, Song Y, Liu H, Jin Y, Li R, Zhu X (2023) DHODH Inhibition Exerts Synergistic Therapeutic Effect with Cisplatin to Induce Ferroptosis in Cervical Cancer through Regulating mTOR Pathway. *Cancers (Basel)* 15

Marazioti A, Krontira AC, Behrend SJ, Giotopoulou GA, Ntaliarda G, Blanquart C, Bayram H, Iliopoulou M, Vreka M, Trassl L et al. (2022) KRAS signaling in malignant pleural mesothelioma. *EMBO Mol Med* 14: e13631

Tibshirani R, Hastie T, Narasimhan B, Chu G (2002) Diagnosis of multiple cancer types by shrunken centroids of gene expression. *Proc Natl Acad Sci U S A* 99: 6567-72

Tsherniak A, Vazquez F, Montgomery PG, Weir BA, Kryukov G, Cowley GS, Gill S, Harrington WF, Pantel S, Krill-Burger JM et al. (2017) Defining a Cancer Dependency Map. *Cell* 170: 564-576 e16

Wang W, Cui J, Ma H, Lu W, Huang J (2021) Targeting Pyrimidine Metabolism in the Era of Precision Cancer Medicine. *Front Oncol* 11: 684961

18th Jun 2025

Dear Prof. Shu,

Thank you for submitting your revised study, which was reviewed by referees #1 and #3. As you will see below, they are satisfied with the revisions, and I will therefore be able to accept your manuscript once the following editorial concerns are addressed:

1/ Please address the minor comments from referee #3.

2/ Manuscript text:

- Please remove the blue font text, and only indicate in track changes mode any new modification in the text.

- Methods:

o Patient-derived material: please include the full statement confirming that the experiments conformed to the principles set out in the WMA Declaration of Helsinki and the Department of Health and Human Services Belmont Report.

o Statistics: please provide a statement on inclusion/exclusion criteria.

- Data availability: thank you for providing reviewer token. Please note that the data must be publicly available before acceptance of the manuscript.

- Please rename 'Disclosure of potential conflicts of interest' to 'Disclosure statement and competing interests'.

- Our journal encourages inclusion of *data citations in the reference list* to directly cite datasets that were re-used and obtained from public databases. Data citations in the article text are distinct from normal bibliographical citations and should directly link to the database records from which the data can be accessed. In the main text, data citations are formatted as follows: "Data ref: Smith et al, 2001" or "Data ref: NCBI Sequence Read Archive PRJNA342805, 2017". In the Reference list, data citations must be labeled with "[DATASET]". A data reference must provide the database name, accession number/identifiers and a resolvable link to the landing page from which the data can be accessed at the end of the reference. Further instructions are available at .

3/ Figures:

The manuscript currently contains six main figures and thirteen EV figures. To balance the manuscript, it is suggested that some of the EV figures be merged together or moved to the Appendix.

4/ Checklist:

Please also fill in the section Experimental study design and statistics/Inclusion & exclusion criteria.

5/ Synopsis: I have cropped a small portion of your image to serve as a thumbnail for the table of content on our webpage (attached). Please let me know if you agree, or provide an alternative image (115x70 pixels). Please note that changes to the synopsis text or image during proofing are usually not allowed.

6/ As part of the EMBO Publications transparent editorial process initiative (see our Editorial at <http://embomolmed.embopress.org/content/2/9/329>), EMBO Molecular Medicine will publish online a Review Process File (RPF) to accompany accepted manuscripts. We note that you agree with the publication of the RPF.

I look forward to receiving your revised manuscript.

Yours sincerely,

Lise Roth

***** Reviewer's comments *****

Referee #1 (Comments on Novelty/Model System for Author):

The novelty of the manuscript. the model systems used and the analyses performed are robust and of high impact.

Referee #1 (Remarks for Author):

The authors have done a great job in addressing this reviewer's previous comments. Warm congrats.

Referee #3 (Remarks for Author):

The authors addressed all the points raised. Still, before publication the authors should attend to the following.

The clustering of cell lines as seen in Figure EV4 E,F is now better explained in the response letter. This explanation should be included also in the manuscript.

S4F shows proline and arginine metabolism as a top hit, not pyrimidine synthesis, please reflect it in the text, pyrimidine synthesis is one of the top hits, and the word 'predominantly' is thus not optimal.

Furthermore, the authors should state the pH used for the LC gradient prior to MS analysis in the glutamine tracing experiment. With these changes included, I recommend this paper for publishing.

Below are our point-by-point responses, with the reviewers' suggestions and comments in black and our responses in blue, and quoted text from the manuscript *in purple and italics*.

To the editors:

We thank you for your feedback and guidance. In response, we have revised the manuscript to fully comply with the journal's formatting and submission requirements. The datasets have been made publicly accessible since July 15th. Additionally, we have moved EV Figures 1-4 and 13 to the Appendix, now labelled as Appendix Figures 1-5, resulting in a total of 8 EV figures. We are also happy to confirm our agreement to use the cropped image you provided as the thumbnail for the table of contents.

Referee #1 (Comments on Novelty/Model System for Author):

The novelty of the manuscript. the model systems used and the analyses performed are robust and of high impact.

Referee #1 (Remarks for Author):

The authors have done a great job in addressing this reviewer's previous comments. Warm congrats.

We sincerely appreciate the reviewer's thoughtful and encouraging feedback. We are grateful for the recognition of the manuscript's novelty, the robustness of the model systems, and the impact of our analyses. We are pleased that our revisions have successfully addressed the previous comments, and we have further refined the presentation of our results and clarified key methodological aspects to enhance the overall rigor, transparency, and clarity of the work.

Referee #3 (Remarks for Author):

The authors addressed all the points raised.

We thank the reviewer for acknowledging that we have adequately addressed all the previously raised points. We appreciate the opportunity to incorporate additional details in the revised manuscript to further strengthen and clarify our findings.

Still, before publication the authors should attend to the following.

The clustering of cell lines as seen in Figure EV4 E,F is now better explained in the response letter. This explanation should be included also in the manuscript. S4F shows proline and arginine metabolism as a top hit, not pyrimidine synthesis, please reflect it in

the text, pyrimidine synthesis is one of the top hits, and the word 'predominantly' is thus not optimal.

We thank the reviewer for this valuable comment. We have now incorporated the detailed information of the clustering strategy into the “Results” section (page 6 in the revised manuscript). We have also revised the text to accurately reflect that, while pyrimidine synthesis is among the top enriched pathways, arginine and proline metabolism appears as the most prominent hit (Appendix Figure S4F). Accordingly, we have removed the term “predominantly” to avoid overstating the enrichment of pyrimidine metabolism.

We further stratified mesothelioma cell lines from the Cancer Cell Line Encyclopaedia (CCLE)(Jiang, Song et al., 2023) into three metabolic subgroups using the nearest shrunken centroids method (Gong, Ji et al., 2021, Tibshirani, Hastie et al., 2002). In principle, cell line clustering is expected to mirror tumor-derived subtypes; however, it is well recognized that long-term in vitro culture can introduce metabolic shifts, causing some cell lines to diverge from their original profiles(Chernova, Sun et al., 2016). To ensure biological relevance and consistency, we therefore restricted our downstream analyses to those CCLE lines whose pathway-based classification aligned precisely with the corresponding tumor clusters (Appendix Figure S4E, F; Dataset EV6, 7).

Our analysis revealed that metabolites differentially expressed in PM cell lines associated with Cluster 1—corroborated by publicly available metabolomics datasets—were significantly enriched in arginine and proline metabolism, as well as pyrimidine metabolism (Appendix Figure S4E, F; Datasets EV6, EV7).

Furthermore, the authors should state the pH used for the LC gradient prior to MS analysis in the glutamine tracing experiment.

We thank you for this helpful comment. We have now included the pH information in the “Methods” section of the revised manuscript (page 28).

Specifically, we now state:

Mobile phase A (pH 9.0) consisted of acetonitrile and water (1:9, v/v) with 20 mM ammonium acetate, and mobile phase B (pH 9.0) consisted of acetonitrile and water (9:1, v/v) with 20 mM ammonium acetate.

With these changes included, I recommend this paper for publishing.

We sincerely appreciate the reviewer’s positive recommendation for publication and are grateful for the constructive feedback that has contributed to strengthening the manuscript.

9th Jul 2025

Dear Prof. Shu,

Thank you for submitting your revised files. I am pleased to inform you that your manuscript is accepted for publication and is now being sent to our publisher to be included in the next available issue of EMBO Molecular Medicine.

Please note that the deposited datasets must be publicly available before online publication of your manuscript.

Yours sincerely,

Lise Roth
